# Suppression of transcriptional drift extends *C. elegans* lifespan by postponing the onset of mortality

Sunitha Rangaraju[1,2,3,4], Gregory M Solis[1,2,3,4], Ryan C Thompson[2], Rafael L Gomez-Amaro[1,2,3,4], Leo Kurian[5], Sandra E Encalada[2,3,4], Alexander B Niculescu III[6], Daniel R Salomon[2], Michael Petrascheck[1,2,3,4]*

[1]Department of Chemical Physiology, The Scripps Research Institute, La Jolla, United States; [2]Department of Molecular and Experimental Medicine, The Scripps Research Institute, La Jolla, United States; [3]Dorris Neuroscience Center, The Scripps Research Institute, La Jolla, United States; [4]Department of Molecular and Cellular Neuroscience, The Scripps Research Institute, La Jolla, United States; [5]Center for Molecular Medicine, University of Cologne, Cologne, Germany; [6]Department of Psychiatry, Indiana University School of Medicine, Indianapolis, United States

**Abstract** Longevity mechanisms increase lifespan by counteracting the effects of aging. However, whether longevity mechanisms counteract the effects of aging continually throughout life, or whether they act during specific periods of life, preventing changes that precede mortality is unclear. Here, we uncover *transcriptional drift*, a phenomenon that describes how aging causes genes within functional groups to change expression in opposing directions. These changes cause a transcriptome-wide loss in mRNA stoichiometry and loss of co-expression patterns in aging animals, as compared to young adults. Using *Caenorhabditis elegans* as a model, we show that extending lifespan by inhibiting serotonergic signals by the antidepressant mianserin attenuates transcriptional drift, allowing the preservation of a younger transcriptome into an older age. Our data are consistent with a model in which inhibition of serotonergic signals slows age-dependent physiological decline and the associated rise in mortality levels exclusively in young adults, thereby postponing the onset of major mortality.

*For correspondence: pscheck@scripps.edu

**Competing interests:** The author declares that no competing interests exist.

## Introduction

The most widely used standard to measure aging of an organism is the quantification of lifespan (*Partridge and Gems, 2007*). Lifespan relates to aging, as the latter causes the degeneration of tissues and organs, thereby increasing mortality due to systemic functional tissue failure (*Balch et al., 2008*; *Bishop et al., 2010*; *David et al., 2010*; *Haigis and Sweet-Cordero, 2011*; *Taylor and Dillin, 2011*; *Gladyshev, 2013*; *Burkewitz et al., 2015*; *Currais, 2015*). Several genetic and pharmacological strategies have been shown to prolong the lifespan of various organisms, including *C. elegans* (*Kenyon et al., 1993*; *Kaeberlein et al., 1999*; *Curran and Ruvkun, 2007*; *Evason et al., 2008*; *Onken and Driscoll, 2010*; *Alavez et al., 2011*; *Chin et al., 2014*; *Ye et al., 2014*; *Tatum et al., 2015*). Mutations in *age-1* or *daf-2,* for example, slow degenerative processes occurring throughout life, thereby constantly lowering mortality rates (*Johnson, 1990*; *Kenyon et al., 1993*; *Taylor et al., 2014*). Age-associated degenerative processes such as a decline in proteostatic capacity are not necessarily restricted to older organisms but can also be observed in young adults (*Labbadia and*

**eLife digest** All organisms age, leading to gradual declines in the body's systems and eventually death. How certain genetic mutations and drugs delay the effects of aging and promote survival to an older age is a question many researchers are exploring. One way this problem is investigated is by looking at how the activity – or expression – of different genes changes during aging.

Scientists interested in understanding aging and longevity often study a simple worm called *Caenorhabditis elegans*. This worm normally lives for about three weeks, and young *C. elegans* are able to produce offspring within days of hatching. This accelerated life cycle allows scientists to observe the entire lifespan of the worms. Over time, experiments have shown that DNA damage, changes in behavior and changes to gene expression are all markers of aging in the worms.

Now, Rangaraju et al. describe how changes in gene expression patterns that begin early in the lives of *C. elegans* shorten their lifespan. Specifically, in groups of genes that work together, some genes increase expression, while others decrease expression with age. This phenomenon is called "transcriptional drift" and leads to an age-associated loss of coordination among groups of genes that help orchestrate specific tasks.

Rangaraju et al. show that an antidepressant called mianserin prevents transcriptional drift in many of *C. elegans*' genes: young worms treated with the drug resist the effects of aging on the transcriptome and maintain coordinated patterns of gene expression for longer. Maintaining coordinated patterns of gene expression postpones the onset of age-related bodily declines and extends the life of treated worms by extending the duration of young adulthood and postponing the onset of age-associated death. The drug also appears to protect against stress-induced changes in gene expression. This suggests that some of the age-related shifts in gene expression occur when cells fail to recover normal gene expression patterns after a stressful event.

Questions that remain to be investigated in future studies are whether other longevity mechanisms also extend lifespan by preserving coordinated gene expression patterns, and whether other longevity mechanisms act by extending specific periods of life.

---

*Morimoto, 2015a*; *2015b*). This raises the possibility of degenerative processes that occur only in young adults and thus specifically contribute to the rise of mortality during young adulthood. Any longevity mechanisms preventing such a degenerative process would specifically slow mortality rates during the period of young adulthood, effectively prolonging its duration to postpone the onset of major age-associated mortality around midlife (*Bartke, 2015*). However, to identify such mechanisms would require mortality-independent metrics of age-associated change, as age-associated mortality rates during young adulthood are difficult to determine by demographic analysis against the back drop of non-aging-related death events (*Partridge and Gems, 2007*; *Beltran-Sancheza et al., 2012*).

In *C. elegans*, mortality-independent metrics of aging include age-associated decline of various behaviors or physiological parameters such as movement or stress resistance (*Huang et al., 2004*; *Bansal et al., 2015*). Molecular markers of aging include sets of genes whose expression change with age, such as micro-RNAs, electron transport chain (ETC) components, or genes involved in posttranslational modifications such as methylation (*Budovskaya et al., 2008*; *de Magalhaes et al., 2009*; *Pincus et al., 2011*; *Horvath et al., 2015*). However, aging also increases DNA damage, affects nuclear architecture, chromatin complexes, chromatin modifications, and the transcriptional machinery (*Mostoslavsky et al., 2006*; *Scaffidi and Misteli, 2006*; *Feser et al., 2010*; *Greer et al., 2011*; *Maures et al., 2011*; *Fushan et al., 2015*). Therefore, an emerging alternative approach to measure specific gene expression changes with age is to quantify the progressive imbalance in gene expression patterns as a function of age. Two such approaches, one measuring transcriptional noise, the cell-to-cell variation in gene expression, and the other measuring decreasing correlation in the expression of genetic modules, showed a loss of co-expression patterns with age (*Bahar et al., 2006*; *Southworth et al., 2009*). These studies suggest that age-associated changes can be measured independently from mortality by tracking the loss of gene expression patterns that are observed in young animals.

In the present study, we set out to investigate the mechanisms by which the atypical antidepressant mianserin extends lifespan by recording the transcriptional dynamics of mianserin-treated and untreated *C. elegans* across different ages. These studies revealed that aging causes *transcriptional drift*, an evolutionarily conserved phenomenon in which the expression of genes change in opposing directions within functional groups. These changes cause a transcriptome-wide loss in mRNA stoichiometry and loss of co-expression patterns in aging animals, as compared to young adults. Mianserin treatment reduced age-associated transcriptional drift across ~80% of the transcriptome, preserving many characteristics of transcriptomes of younger animals. We used transcriptional drift along with mortality analysis as metrics to monitor aging and find that mianserin treatment extended lifespan by exclusively slowing age-associated changes in young adults, thereby postponing the onset of mortality.

## Results

### Aging causes a loss of co-expression patterns observed in young adults

To better understand how aging changes gene expression patterns in a eukaryotic organism, and how these changes are affected by longevity, we measured gene expression changes in mianserin-treated or untreated *C. elegans* by RNA-sequencing (RNA-seq; *Figure 1a*). Cohort #1 was a time series to study how gene expression patterns change over time in control (water) animals or in animals treated with mianserin on day 1 of adulthood (24 hr after L4 stage). Cohort #2 was designed to study dosage effects of increasing concentrations of mianserin with aging, and cohort #3 was designed to study the effects of delayed mianserin-treatment of worms treated at day 3 or 5 of adulthood (*Figure 1a*). Lifespan of a sub-population of each cohort was simultaneously assessed to ensure the effect of mianserin.

Comparison of gene expression profiles of age-matched mianserin-treated and untreated controls, showed that approximately 3,000–6,000 genes changed with age in response to mianserin treatment (FDR<0.1, *Figure 1—source data 1*) (*Robinson and Oshlack, 2010*; *Kim et al., 2013*; *Lawrence et al., 2013*). We separated genes into sets that showed increased or decreased expression in response to mianserin, to conduct gene-set enrichment analysis. This revealed hundreds of gene ontologies (GO) that changed in response to mianserin (*Figure 1—source data 2*) (*Ashburner et al., 2000*; *Mi et al., 2005*). We observed that many GOs were enriched for both, genes that increased as well as decreased as a consequence of aging. This observation complicated any interpretation on whether pathways were activated or inhibited in response to mianserin, and how the associated function (GO) relates to mianserin-induced lifespan extension (*Figure 1b*).

We observed a similar scenario by conducting gene-set enrichment analysis for gene expression changes in response to age in untreated animals. As seen with mianserin, many GO annotations were enriched for both up- as well as downregulated genes at any given age (*Figure 1c*; *Figure 1—source data 3*, *4*), making it difficult to interpret whether those pathways are being activated or inhibited with age. We generated 50 representative pie charts out of the 249 GO annotations that contained genes that increased or decreased in expression by day 10 due to aging. These charts suggested that as animals age and become older, genes change expression in opposing directions, disrupting relative mRNA ratios within the GO, when compared to young adults (*Figure 1—figure supplement 1*). Thus, aging changed the stoichiometric relationship between mRNAs belonging to the same functional group (GO). In many cases, the fractions of genes that increased, decreased or did not change in expression showed no consistent pattern, nor provided any insight into the pathway activity (*Figure 1—figure supplement 1*).

Because the expression patterns observed in many GOs were difficult to interpret in terms of functional change, we turned to investigate expression changes in the superoxide detoxification pathway, a well-defined cellular function that declines with age (*Ashburner et al., 2000*; *Mi et al., 2005*; *Kumsta et al., 2011*; *Bansal et al., 2015*; *Rangaraju et al., 2015a*). As expected from our previous studies (*Rangaraju et al., 2015a*), the expression levels of some superoxide detoxification genes were higher in mianserin-treated animals compared to age-matched controls (*Figure 1d*). Exceptions were the expression levels of *sod-4* and *sod-5,* which were lowered upon mianserin treatment (*Figure 1d*). However, plotting expression changes of superoxide detoxification genes as a function of age (*Figure 1e*, left panel) revealed again a scenario in which genes changed in opposing

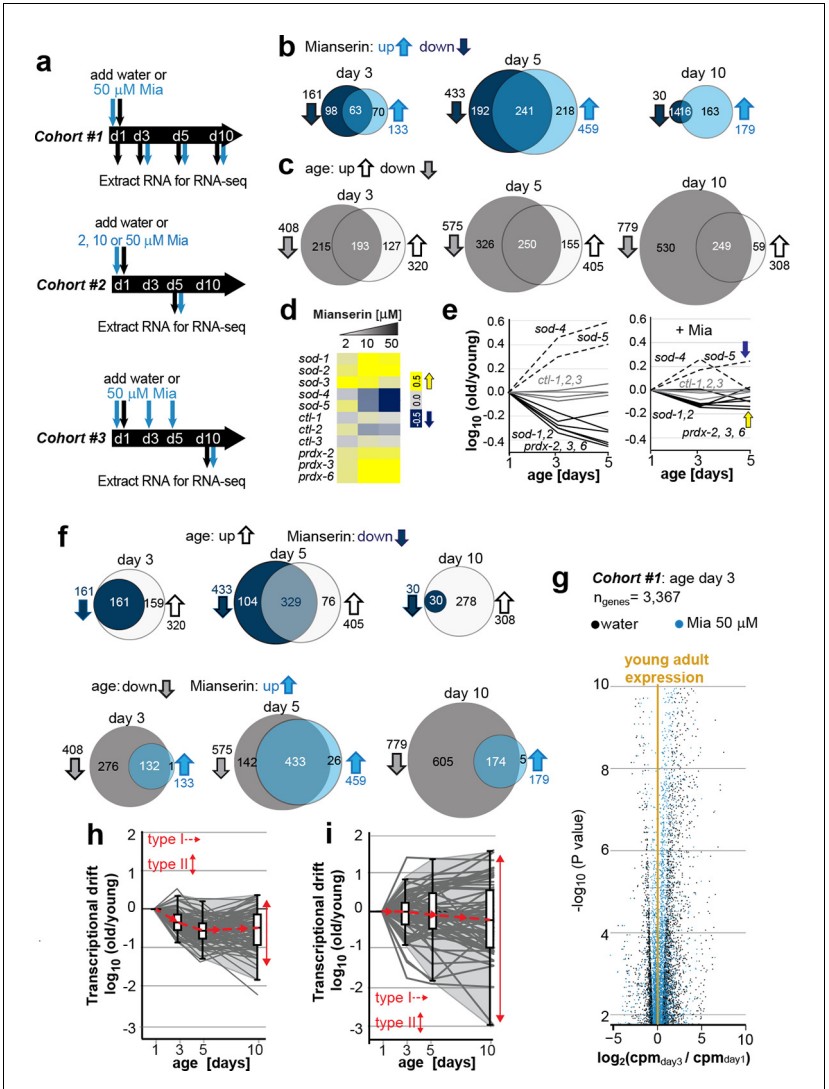

**Figure 1.** Transcriptional drift-variance increases with age. (**a**) Schematic of RNA-seq experiment. In cohort #1, water or mianserin was added on day 1 of adulthood and RNA samples were harvested on day 1 (water only), day 3 (d3), day 5 (d5) and day 10 (d10). In cohort #2, animals were treated with water or increasing concentrations of mianserin (2, 10 or 50 μM) on day 1 (d1) and RNA was harvested on day 5 (d5) for RNA-seq. In cohort #3, water or 50 μM mianserin was added on day 1, day 3, and day 5, and RNA was harvested on day 10 (d10) for RNA-seq. (**b**) Venn diagrams of the number of GOs enriched for genes that decrease expression with mianserin (down, dark blue circle) increase expression with mianserin (up, light blue circle) or are enriched for both (intersection). (**c**) Venn diagrams of the number of GOs enriched for genes that decrease expression with age (down, gray circle) increase expression with age (up, white circle) or are enriched for both (intersection). (**d**) Heat map depicting $\log_2$ changes in gene expression for oxidative stress genes elicited by increasing concentrations of mianserin (yellow, increased expression; blue, decreased expression) (**e**) Mianserin decreases expression of redox genes that increase with age and increases expression of genes that decrease with age. (**f**) Mianserin reverts age-associated changes on the level of GOs. Venn diagrams of the number of GOs enriched for genes that decrease expression with mianserin (down, dark blue circle) and increase with age (up, white circle) or vice versa (down with age, gray circle; up with mianserin, light blue circle). (**g**) Mianserin reverts age-associated changes on the level of individual genes. Volcano plot shows the negative $\log_{10}$ of P-values as a function of $\log_2$ fold changes of 3,367 genes that significantly change expression from day 1 to day 3 in samples of water-treated control animals (black) or samples from age-matched mianserin-treated animals (50 μM, blue). As animals age, gene expression levels change ("drift") away from levels observed in young adults (yellow line). Mianserin treatment attenuates age-associated gene expression changes preserving expression levels as seen in young adults. (**h**) Drift-plot shows log fold change (old/young) as a function of age for each gene involved in oxidative phosphorylation (gray lines. KEGG: cel 04142). Superimposed

*Figure 1 continued on next page*

*Figure 1 continued*

are Tukey-style box-plots to graph the increases in drift-variance across the entire pathway. Gene expression changes are classified into type I, which describes activation or repression of the entire pathway and into type II, which describes changes among genes relative to each other (drift-variances), see red arrows. (i) Drift-plot for lysosomal genes (KEGG: cel 00190). See *Figure 1—source data 1–5*, *Figure 1—figure supplement 1* and *Table 1* for additional information on data-sets. Also see Methods section for transcriptional drift calculation in each figure panel.

The following source data and figure supplement are available for figure 1:

**Source data 1.** RNA-seq gene expression data.

**Source data 2.** Gene ontologies changing in response to mianserin treatment.

**Source data 3.** Gene ontologies changing in response to age.

**Source data 4.** Differentially expressed genes in response to age.

**Source data 5.** Differentially expressed genes in response to mianserin treatment.

**Figure supplement 1.** This figure relates to *Figure 1* in main text.

---

directions as seen in the pie charts for many GOs before (*Figure 1—figure supplement 1*). Some mRNAs including those of *sod-4, -5* increased with age, while some decreased (*sod-1, -2, prdx-2, 3, 6*) and some did not change (*ctl-1, 2, 3*), leading to an overall 5-10-fold change in stoichiometric balance among superoxide detoxification-associated mRNAs by day 5 (*Figure 1e*, left panel). More interestingly, if the expression of an *sod* increased with age, mianserin treatment prevented the increase and if the expression of an *sod* decreased with age, mianserin prevented the decrease (*Figure 1e*, right panel). Thus, when we took the mRNA expression levels of young animals into account, the emerging picture suggested that mianserin treatment attenuated age-associated gene expression changes.

We therefore asked whether the complex gene-set enrichment patterns observed comparing mianserin-treated and untreated samples (*Figure 1b,c*) could be explained by mianserin preventing expression changes due to age. Indeed, many GO annotations that increased expression with age were decreased by mianserin treatment and vice versa (*Figure 1f*). This attenuation of age-associated changes by mianserin treatment was even more pronounced for individual genes (*Figure 1g*). Analyzing cohort #1 showed a significant change in expression levels of 3,367 genes, as the animals aged from day 1 to day 3, and a change in 5,947 genes from day 1 to day 10 (FDR < 0.1) (*Figure 1g*, significant genes only). Mianserin treatment reduced these age-associated expression changes in over 90% of cases. Including all age-associated expression changes for the 19,196 different transcripts present in our data-set, we found that mianserin treatment attenuated age-associated changes in transcription in 15,095 out of 19,169 genes (80%, binomial $P < 10^{-100}$). Thus, most of the changes observed between mianserin-treated and untreated animals are due to mianserin preventing transcriptional changes with age.

When we excluded all genes that changed due to age and were attenuated by mianserin, we obtained a much smaller gene-set consisting of mianserin-induced changes that was enriched for GOs related to stress, xenobiotic and immune-responses, as well as genes associated with aging and the determination of lifespan (*Table 1*, *Figure 1—source data 5*). These GOs have been previously shown to be regulated by serotonin in *C. elegans* with the exception of the xenobiotic response (*Zhang et al., 2005*; *Petrascheck et al., 2007*; *Rangaraju et al., 2015a*). Thus, accounting for age-associated transcriptional changes dramatically simplified a seemingly very complex gene-expression pattern (*Figure 1b,c*). It revealed that mianserin affected expression of a small set of physiological functions that are known to be regulated by serotonin and have been shown to be required for mianserin-induced lifespan extension or for aging in general (*Garsin et al., 2003*; *Rangaraju, et al., 2015*; *Petrascheck, et al., 2007*) (*Table 1*; *Figure 1f*; *Figure 1—source data 5*).

**Table 1.** GO annotations enriched for genes upregulated by mianserin during all ages, assessed by RNA-seq (day 3, 5 and 10).

| GO | Enriched P-value |
| --- | --- |
| response to stimulus | 4.47E-08 |
| response to stress | 5.83E-05 |
| response to xenobiotic stimulus | 3.25E-07 |
| defense response | 4.66E-05 |
| innate immune response | 1.56E-02 |
| immune response | 1.62E-02 |
| immune system process | 1.62E-02 |
| aging | 6.63E-05 |
| multicellular organismal aging | 6.63E-05 |
| determination of adult lifespan | 6.63E-05 |

Note: No process was specifically downregulated for all three ages.

Based on these observations, we classified gene expression changes for groups of genes into two types. Type I changes describe whether the overall expression across an entire functional group/pathway increases or decreases i.e. whether the pathway is up or down regulated with age. Type II changes describe the relative changes in gene expression among genes *within* functional groups with respect to each other. We named the type II change *transcriptional drift*. As animals age, genes within functional groups change expression levels in opposing directions resulting in the disruption of the co-expression patterns seen in young adults.

To analyze the effects of aging on transcriptional drift (type II), we designed graphs that plot the log-fold changes (log [old/young reference $_{day1}$]) in gene expression as a function of age. Such a plot can be constructed for whole transcriptomes as well as for any functional subset of genes, for example, genes involved in oxidative phosphorylation or lysosome biology (*Figure 1h,i*). In young adults, the log-fold change is 0 and values close to 0 therefore suggest gene expression as seen in young adults (*Figure 1h,i*). To quantify transcriptional drift changes with age (type II), we calculated the variance of the log-fold change for genes involved in each pathway. For the purpose of this study, we will refer to this variance as drift-variance (see Materials and methods). If gene expression ratios within a pathway stay constant with age, drift-variance will stay small. If a majority of genes within a pathway change expression in opposing directions or if the rates by which they change differ dramatically, drift-variance will increase. Note that "transcriptional drift" is different from "transcriptional noise" in that the former analyzes variance *among* genes *within the same biological replicates,* whereas the latter analyzes *variance of the same genes among biological replicates.* Hence, how far the aging transcriptome deviates away from the transcriptome seen in young adults can be graphed in a Tukey-style box plot, which plots the drift-variance as a function of age (*Figure 1h,i*). We will refer to these plots as drift-plots (*Figure 1h*; *Figure 2—figure supplement 1a–d*).

## Longevity mechanisms attenuate transcriptional drift-variance

We constructed drift-plots for all 19,196 genes in the data of cohort #1, which revealed a dramatic increase in drift-variance with age, showing a progressive loss of mRNA stoichiometries and co-expression patterns observed in young-adults (*Figure 2a*, shaded region encompassing the whiskers of Tukey-plot). This effect was also seen in other publicly available data-sets of aging *C. elegans* transcriptomes and drift-variance continued to increase with age at least until day 20 (*Figure 2—figure supplement 1e*). Mianserin treatment attenuated the effect of aging across the whole transcriptome and preserved the co-expression patterns observed in young-adults into later age. To test whether transcriptional drift is driven by a small subset of mRNAs or a transcriptome-wide phenomenon, we randomly divided the transcriptome into subsamples of ~1,000 genes. Each subsample showed identical increases in drift-variance with age, confirming a transcriptome-wide effect (*Figure 2—figure supplement 1f*).

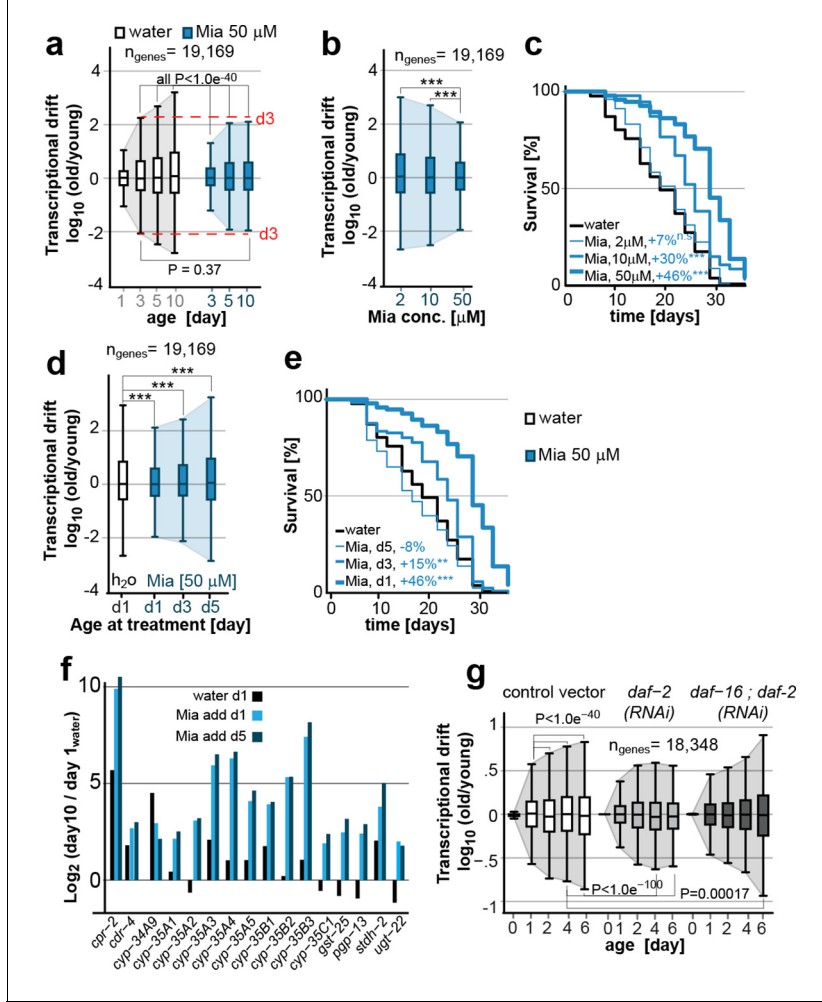

**Figure 2.** Transcriptional drift-variance is attenuated by two longevity paradigms. (**a**) Drift-plots show that mianserin attenuates increasing drift-variance with age. Note that drift-variance in 10-day-old mianserin-treated animals is the same as in untreated 3-day-old control animals (dotted red line). (**b**) Drift-plots show that increasing concentrations of mianserin cause drift-variance to decrease. Drift-variance was measured on day 5 by RNA-seq. (**c**) Corresponding to b, lifespan curves show that increasing concentrations of mianserin leads to a dose-dependent increase in survival. (**d**) Drift-plots show that initiating mianserin treatment at later ages reduces (d3) or abolishes (d5) its effect on transcriptional drift. Drift-variance was measured on day 10 by RNA-seq. (**e**) Corresponding to d, lifespan curves show initiating mianserin treatment at later ages reduces (d3) or abolishes (d5) its effect on lifespan. (**f**) Log-fold change of xenobiotic gene expression on day 10 when mianserin was added on day 1 or day 5, compared to control animals treated with water on day 1. Adding mianserin on day 1 or day 5 leads to comparable changes. (**g**) Drift-plots show *daf-2* RNAi attenuates increasing drift-variance with age in a manner dependent on *daf-16*. Left: vector control, middle: *daf-2* RNAi, right: *daf-16/daf-2* RNAi. P-values for transcriptional drift plots are calculated by robust Levene's test, which compare variances and not mean values. \*\*\*P<0.001. All error bars show drift-variance. See *Figure 2—figure supplement 1–2* for additional information on calculating drift-variance and *Table 2*. Also, see Methods section for transcriptional drift calculation in each figure panel.

The following figure supplements are available for figure 2:

**Figure supplement 1.** This figure relates to *Figure 1c, d* and *Figure 5a,b* in main text.

**Figure supplement 2.** Egg RNA does not affect drift-variance.

We previously showed, that the effect of mianserin to extend lifespan is dose-dependent (*Petrascheck et al., 2007*). To explore a possible quantitative relationship between longevity and drift-variance, we generated drift-plots for transcriptomes of animals treated with increasing doses of mianserin (*Figure 1a*, cohort #2). Increasing doses of mianserin progressively increased longevity and decreased drift-variance as measured in 5-day-old animals (*Figure 2b,c*; *Table 2*). Thus,

**Table 2.** Survival data for lifespan of RNA-seq experimental cohorts.

| Strain | Treatment | Treatment added on [day] | Conc. [μM] | Change in lifespan [%] Expt.1/ Expt.2/ Expt.3 | P-value Expt.1/ Expt.2/ Expt.3 | Mean Lifespan [days] Expt.1/ Expt.2/ Expt.3 | Number of animals Expt.1/ Expt.2/ Expt.3 |
|---|---|---|---|---|---|---|---|
| N2 | Water | d1 | 0 | | | 19.33/ 17.2/ 20.45 | 132/ 149/ 130 |
| N2 | Mia | d1 | 2 | +7/ +12/ -4 | 0.20/ 0.04/ 0.25 | 20.64/ 19.23/ 19.67 | 125/ 133/ 151 |
| N2 | Mia | d1 | 10 | +30/ +16/ +6 | 2.5E-07/ 3.7E-03/ 0.55 | 25.09/ 19.92/ 21.74 | 94/ 138/ 136 |
| N2 | Mia | d1 | 50 | +46/ +39/ +25 | 1.1E-19/ 1.9E-15/ 2.8E-08 | 28.25/ 23.92/ 25.63 | 95/ 131/ 125 |
| N2 | Mia | d3 | 50 | +15/ +14/ +1 | 2.0E-03/ 9.3E-04/ 0.29 | 22.23/ 19.69/ 20.75 | 121/ 134/ 152 |
| N2 | Mia | d5 | 50 | -8/ +8/ -2 | 0.18/ 0.06/ 0.84 | 17.79/ 18.52/ 20.13 | 123/ 151/ 139 |

Summary of all lifespan experiments performed in parallel for cohorts 1 and 2 of the RNA-seq studies in *Figure 2c,e*. The treatments, water or mianserin, at the indicated concentrations (conc.) were added on indicated day (D) of adulthood and lifespan (days) was scored until 95% of animals were dead in all tested conditions. All values (Change in lifespan [%], P-values) were calculated for the pairwise comparison between mianserin-treated and water-treated animals of the same condition, in 3 independent experiments (expts.). Statistical analysis was performed using the Mantel–Haenszel version of the log-rank test. Mean lifespan [days] and number of animals in each experiment are indicated.

remarkably, by varying the dose of a single molecule, it was possible to control the degree to which aging drives the loss of transcriptional co-expression away from patterns observed in young adults. These results suggested a quantitative relationship between mianserin-induced longevity and its effect on drift-variance.

Our previous studies had also shown that mianserin does not extend lifespan when added to 5-day-old post-reproductive adult animals (*Petrascheck et al., 2007*). Thus, we next tested whether mianserin attenuates transcriptional drift-variance independently of longevity by treating older animals. Mianserin did not attenuate transcriptional drift-variance when added on day 5 (*Figure 2d*). Adding mianserin on day 3 of adulthood caused a small extension of lifespan and a corresponding small attenuation of drift-variance, further supporting a quantitative relationship between suppression of drift-variance and extension of lifespan (*Figure 1a*, cohort #3, *Figure 2d,e*; *Table 2*). However, mianserin fully induced the xenobiotic response by up to 1,000-fold irrespective of whether added on day 1 or day 5 (*Figure 2f*). Therefore, the lack of an effect of mianserin when added to day 5 adults cannot be attributed to reduced drug uptake. Taken together, these results show that mianserin does not attenuate drift-variance when it does not extend lifespan.

We next asked whether the attenuation of drift-variance is unique to mianserin or whether it is observed in other lifespan-extension paradigms (*Figure 2g*). We asked whether reduced insulin signaling also attenuates drift-variance by analyzing the previously published gene expression data-sets of long-lived *C. elegans daf-2* RNAi-treated and vector control animals (*Murphy et al., 2003*). Analyses of drift-variance for these data-sets showed that treatment with *daf-2* RNAi attenuated drift-variance (*Figure 2g*). Moreover, mianserin and *daf-2* RNAi attenuated age-associated drift of overlapping sets of genes. Of the 6,958 genes for which expression levels were detected at all ages in both data-sets, 58% (4,078 genes, binomial P= 6.3e-47) were attenuated by both longevity-extending mechanisms. This overlap is consistent with experiments showing that these two longevity mechanisms partially overlap, potentially explaining why mianserin only causes a +11% lifespan extension in *daf-2(e1370)* mutant animals instead of 31% seen in the parallel wild-type experiments (*Petrascheck et al., 2007*). Thus, lifespan extension by mianserin or *daf-2* RNAi attenuates transcriptional drift in overlapping sets of genes.

Conversely, suppressing longevity by *daf-16(RNAi)* prevented the attenuation of drift-variance by *daf-2(RNAi)* and increased it beyond what was seen in control animals (*Figure 2g*). Thus, the activation of DAF-16 target genes leads to the attenuation of transcriptional drift in thousands of genes across the transcriptome. Taken together, these results show that drift-variances increase with age in *C. elegans* and are attenuated in two different longevity paradigms (*Figure 2a,g*).

From a technical perspective, the comparison between the mianserin data and the Murphy data (*Murphy et al., 2003*) also shows that the phenomenon of transcriptional drift is robust enough not to be influenced by the presence of eggs in the animals or the method of sterilization, as our study

used FUDR and the *Murphy et al. (2003)* study used sterile mutants (*Figure 2a,g*; *Figure 2—figure supplement 2*).

## Attenuating drift-variances in redox-pathways preserves homeostatic capacity

The results above suggested that preserving low drift-variance in transcriptomes preserves longevity. We therefore asked whether attenuating drift-variance in specific pathways preserves homeostatic capacity, the ability of pathways to appropriately respond to a stimulus or stress. Throughout life, organisms respond to stimuli by activating or repressing transcriptional programs, an ability that is lost with age. We hypothesized that one way by which regulatory ability may be lost could be due to a failure to return to their precise steady-state transcriptional levels after stimulation. This would give rise to increases in drift-variance (*Figure 3a*), as seen in the drift plots for oxidative phosphorylation or lysosome biology (*Figure 1h,i*). In this model, slight initial deviations in gene expression levels would be compounded over time resulting in imbalanced stoichiometries between pathway components resulting in functional decline with age (*Figure 3a*).

Our previous studies showed that mianserin protected *C. elegans* from oxidative stress by a neuronal mechanism that modulated peripheral stress response genes (NEUROX) (*Rangaraju et al., 2015a*). We therefore constructed drift plots for redox-associated pathways that showed that mianserin indeed increased the overall expression of oxidative stress response genes (type I) relative to age-matched controls but also attenuated transcriptional drift (type II) (*Figure 3b*; *Table 3*).

We therefore asked whether mianserin treatment increased resistance to oxidative stress by either directly activating the oxidative stress response or whether attenuating transcriptional drift would preserve homeostatic capacity into older age (*Rahman et al., 2013*). Animals were treated with water or mianserin on day 1 of adulthood, followed by treatment with the reactive oxygen species (ROS) generator paraquat on day 3, 5, or 10 (*Figure 3c*). On day 3 of adulthood, no difference in stress resistance between mianserin-treated and untreated animals was observed. As animals grew older (day 5 and day 10), mianserin treatment greatly improved stress resistance (*Figure 3c*; *Table 4*). Again, as with lifespan, delaying the start of mianserin treatment to day 3 and day 5 progressively reduced its protective effect on stress resistance, this time measured in animals subjected to paraquat on day 10 of adulthood (*Figure 3d*; *Table 5*). Thus, mianserin treatment specifically improves stress resistance in older (day 5 and day 10) but not in younger (day 3) animals consistent with a model in which it preserves the homeostatic capacity of redox function.

To further distinguish between a model in which mianserin directly activates an oxidative stress response from one that preserves the homeostatic capacity by attenuating drift-variance, we asked whether mianserin enhanced (direct activation) or attenuated (preserving capacity) genes that change in response to oxidative stress (*Figure 3e*). Oliveira et al. identified 252 genes that were upregulated and 88 genes that were downregulated in young *C. elegans* in response to oxidative stress, and can therefore be considered an experimentally determined oxidative stress signature (*Oliveira et al., 2009*). We hypothesized that a direct activation of the oxidative stress response by mianserin would mimic the increase in expression of the 252 genes and the decrease in the expression of the 88 genes as seen in response to oxidative stress. However, we observed an attenuation rather than an activation of the oxidative stress signatures, consistent with preserving homeostatic capacity rather than a direct activation. Genes that increased in response to oxidative stress (252) showed a lower expression while genes that decreased (88) in response to oxidative stress showed a higher expression in age-matched mianserin-treated animals (*Figure 3e*). Consistent with the functional data, differences in the oxidative stress signature were only observed in older animals (day 5, 10), but not in younger day 3 animals. These results are consistent with a model in which mianserin treatment preserves the redox system from age-associated decline, thus improving redox capacity in older age.

## Mianserin requires the serotonin receptor SER-5 to preserve low drift-variances

In mammals, mianserin antagonizes serotonergic signals sent by 5-HT2A/C receptors (*Gillman, 2006*). We next asked whether preservation of redox capacity and reducing drift-variance in redox pathways by mianserin depends on serotonergic signaling. To identify the serotonergic

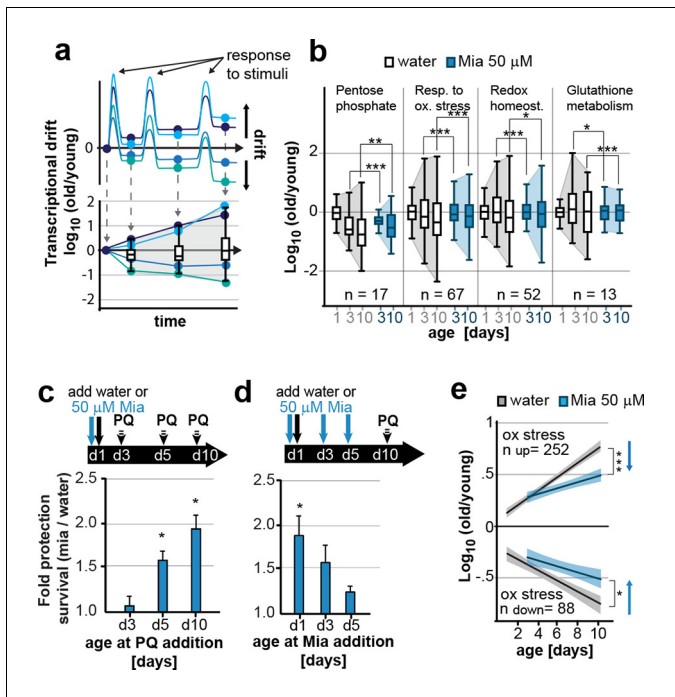

**Figure 3.** Preserving low drift-variances in redox pathways preserves redox capacity into old age. (a) Model for the occurrence of transcriptional drift with age. Genes belonging to the same pathway appropriately respond to a stimulus but subsequently fail to return to steady-state levels. Repeated stimuli compound this effect leading to increases in transcriptional drift. If multiple genes within a pathway have propensity to drift in one or the other direction drift-variance increases with age. (b) Drift-plots show increases in drift-variance in multiple KEGG or GO annotations associated with redox processes. P-values compare variance, not mean, n: No. of genes in each category. *P<0.05, **P<0.01, ***P<0.001, Levene's test. Error bars; drift-variance (c) Fold increase in survival of N2 wild-type (wt) mianserin treated vs. untreated animals when challenged with paraquat at different ages. The protective effect of mianserin increases with age. *P<0.05, *t-test*, Error bars: S.E.M. (d) Fold increase in survival of wt (N2) treated vs. untreated animals when challenged with paraquat on day 10. Delaying mianserin treatment into later life reduces its protective effect. *P<0.05, *t-test*, Error bars: S.E.M. (e) Linear regression of log fold-changes in gene expression with age for genes previously shown to change upon oxidative stress. Genes upregulated in response to oxidative stress (n=252) increase with age, and genes downregulated in response to oxidative stress decrease (n=88) with age. Mianserin attenuates age-associated expression changes in oxidative stress genes in the direction indicated by blue arrows. Shading: 95% confidence interval. ***P<0.001, Wilcoxon rank-sum test. See *Tables 3–5* for detailed statistics and Methods section for transcriptional drift calculation in each figure panel.

receptor, we treated multiple mutants, each deficient in signaling by a single G-protein coupled receptor (GPCR) with mianserin on day 1, followed by increasing concentrations of paraquat on day 5 to induce oxidative stress (*Figure 4a,b*; *Table 6*). Mianserin was unable to protect multiple *ser-5* mutant alleles (*ok3087, tm2647, tm2654*) from oxidative stress (*Figure 4a,b*; *Figure 4—figure supplement 1a*; *Table 6*). In addition, seven structurally distinct serotonergic antagonists/inverse agonists also protect from oxidative stress in a *ser-5* dependent manner (*Figure 4—figure supplement 1b*; *Table 7*). Furthermore, mianserin did not protect animals unable to synthesize serotonin (*tph-1 (mg280)*) (*Figure 4a*; *Table 6*) (*Sze et al., 2000*).

We next asked whether SER-5 was also required for mianserin to preserve low transcriptional drift-variances in redox-related genes. We measured redox gene expression levels by qRT-PCR in wild-type 5-day-old N2 and *ser-5(ok3087)* animals that were treated with mianserin or water on day 1 (*Figure 5a,b*; *Figure 1—figure supplement 1a*). In N2 samples, mianserin increased the expression of stress response genes that drift down with age (*sod-1, sod-2, prdx-2, -3, -6*) and decreased the expression of stress response genes that drift up with age (*sod-4, sod-5,* all *hsp-16s*), an effect that was not observed in *ser-5(ok3087)* mutants. In contrast, SER-3 and SER-4, two receptors we previously showed to be required for lifespan extension by mianserin, were dispensable for stress

protection (*Figure 4a,b*) (*Petrascheck et al., 2007*), as well as for the attenuation of drift-variance in redox-associated genes (*Figure 4—figure supplement 1c*). Thus, in wild-type animals, mianserin treatment preserved low drift-variances in redox-related genes into older age (day 5), in a *ser-5* dependent manner (*Figure 5a,b*).

Importantly, *ser-5* mutants were specifically defective in their response to mianserin, but showed no defect in their response to oxidative stress. Young (day 1) wild-type N2 animals and *ser-5 (ok3087)* mutants showed a nearly identical response to oxidative stress (*Figure 5c*). The age-specific effects of *ser-5* could not be attributed to expression changes, as *ser-5* expression remained constant from day 1 to day 10 in our RNA-seq experiment.

To test the hypothesis that mianserin preserved the homeostatic capacity of the redox system, as suggested by *Figure 3e*, we asked whether the treatment with mianserin on day 1 of adulthood led to an enhanced redox gene expression in response to the stressor paraquat in older animals (day 5). We therefore challenged older mianserin-treated or control animals (day 5) with paraquat for 8 hr and measured redox-gene expression by qRT-PCR (*Figure 5d*). Mianserin treatment led to an enhanced transcription of redox genes in response to paraquat as compared to age-matched control animals. The enhanced response was *ser-5* dependent (*Figure 5d*). Thus, SER-5 is required for mianserin to attenuate age-associated increases in drift-variance in redox genes, and to preserve the homeostatic capacity of the redox system into older age.

Furthermore, lifespan-extension by mianserin was strongly reduced or abrogated in *ser-5, snt-1* and *unc-26* mutant animals (*Figure 5e, f*; *Figure 5—figure supplement 1a*; *Table 8*). Seven additional serotonergic antagonists/inverse agonists also extended lifespan in a manner that was partially or fully dependent on *ser-5* (*Figure 5—figure supplement 1b*). Thus, these results show that inhibiting serotonergic signals via SER-5 extends lifespan, attenuates age-associated drift-variance in the redox system and preserves the homeostatic capacity of the redox system.

## Mianserin prolongs lifespan by slowing age-associated change in young adults

We next asked whether drift-variance could be used as a metric to monitor age-associated change in young adults. Comparing drift-variances between mianserin-treated and untreated animals, we noticed that by day 10, mianserin-treated animals exhibited a drift-variance slightly lower than that of 3-day-old control animal (P=0.37). This suggested that mianserin-treated animals showed a ~7–8 day delay in age-associated transcriptional change compared to age-matched controls (*Figure 2a*).

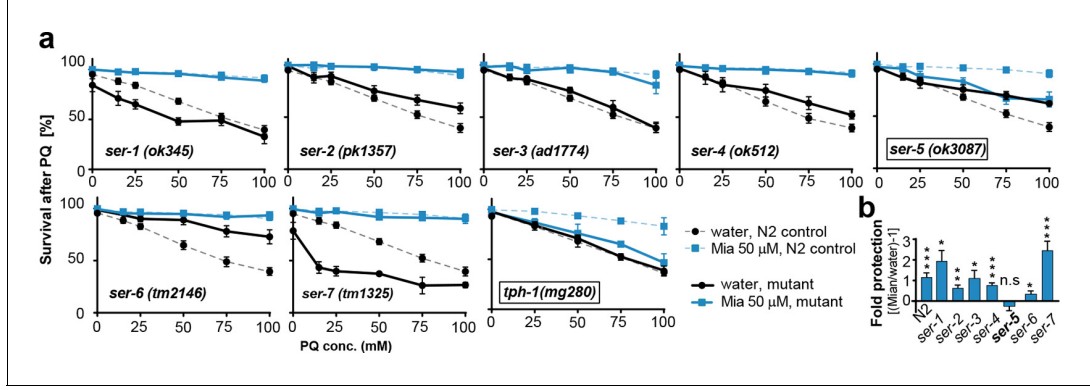

**Figure 4.** Preserving redox capacity into old age requires the serotonin receptor SER-5. (a) Survival of wt (dotted lines) or serotonin receptor mutants and serotonin synthesis mutant (bold lines) treated with water (black) or mianserin (blue) on day 1, followed by increasing concentrations of paraquat on day 5. (b) Bar graph shows fold protection as a ratio of survival of mianserin-treated vs. water-treated GPCR mutant animals ((Mia/water)-1). *P<0.05, **P<0.01, ***P<0.001, n.s., not significant, *t-test*; Error bars: S.E.M. See *Figure 4—figure supplement 1*, and *Tables 6* and *7* for detailed statistics.

The following figure supplement is available for figure 4:

**Figure supplement 1.** This figure relates to *Figure 4a* in main text.

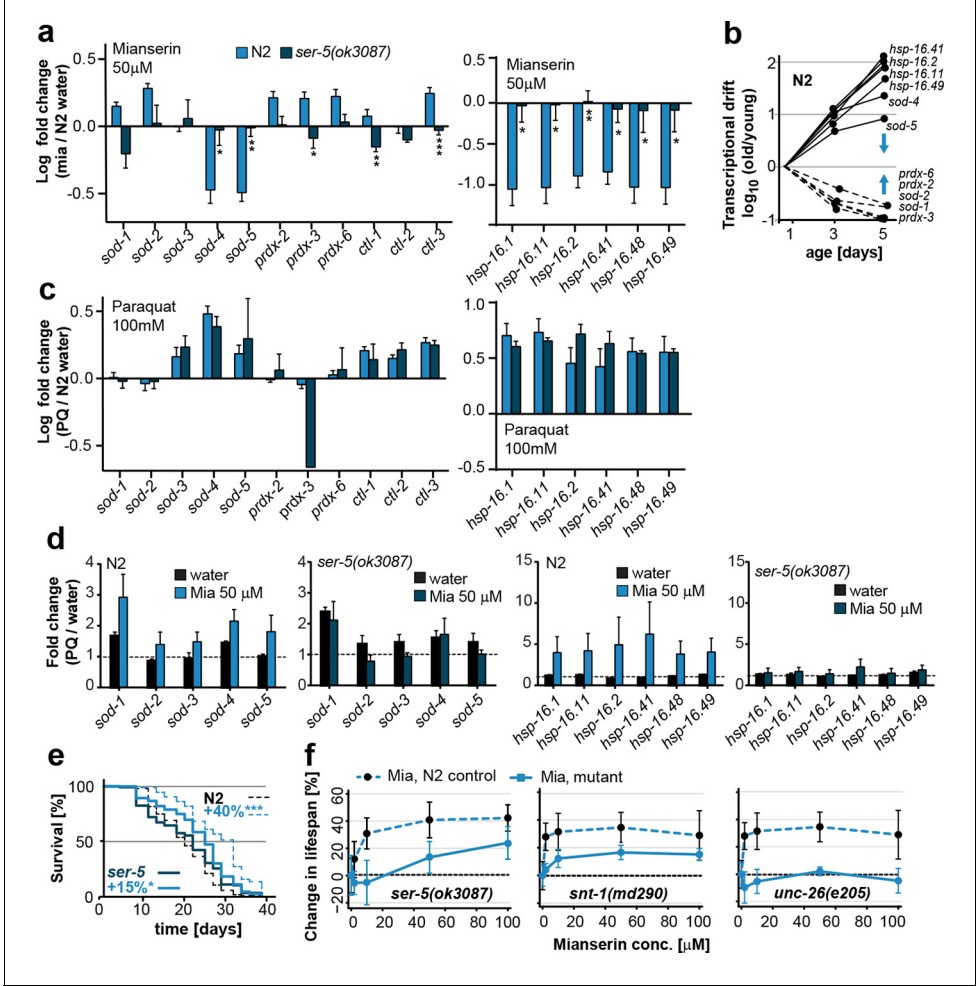

**Figure 5.** Mianserin attenuates drift-variance in peripheral tissues via SER-5. (a) Bar graphs quantifying transcriptional drift (log fold-changes in gene expression) as measured by qRT-PCR in 5-day-old N2 and *ser-5(ok3087)* animals treated with mianserin, relative to water-treated N2. Mianserin treatment increases expression of genes drifting down with age and decreases expression of genes drifting up with age in N2, but not in *ser-5(ok3087)* mutants. (See 5b). *P<0.05, **P<0.01, ***P<0.001,t-test; Error bars: S.E.M. (b) Log fold-change in gene expression as a function of age for stress response genes shown in a. Blue arrows indicate how mianserin treatment corrects age-associated changes in gene expression toward an expression pattern as seen in young adults. (c) Bar graphs quantifying log fold-changes in gene expression in 1-day-old N2 and *ser-5(ok3087)* animals treated with paraquat, relative to water-treated N2 animals. N2 and *ser-5(ok3087)* show an identical response to paraquat. (d) Mianserin treatment on day 1 of adulthood enhances transcription of *sod* and *hsp-16.x* genes in response to an 8h paraquat treatment on day 5 in wt (N2) animals compared to water treated controls. In contrast, mianserin treatment of *ser-5(ok3087)* animals did not enhance transcription of *sod* and *hsp-16.x* genes. mRNA levels of genes were evaluated by qRT-PCR and plotted as fold induction (PQ/water) (Y-axis) for each gene. (e) Survival plot of mianserin-treated and untreated N2 and *ser-5(ok3087)* animals. ***P<0.001, *P<0.05, Mantel–Haenszel version of the log-rank test. f) Percent increase in lifespan as a function of mianserin concentration. Mutations in *ser-5* or synaptic components rendered the animals partially or completely resistant to mianserin-induced lifespan extension. See *Figure 5—figure supplement 1* for additional data, and *Table 8* for detailed statistics.

The following figure supplement is available for figure 5:

**Figure supplement 1.** This figure relates to *Figure 5e* in main text.

Principle component analysis (PCA), a different statistical method to analyze differences between transcriptomes, confirmed this observation (*Figure 6a*). PCA showed that control samples aligned on the x-axis (dimension 1) according to age and that 10 day-old mianserin-treated animals aligned closer to 3-day-old than to 10-day-old control animals. These results suggested that the physiological shift that results in the 7–8 day lifespan extension observed in mianserin-treated animals at the end of a lifespan assay was already observable by day 10.

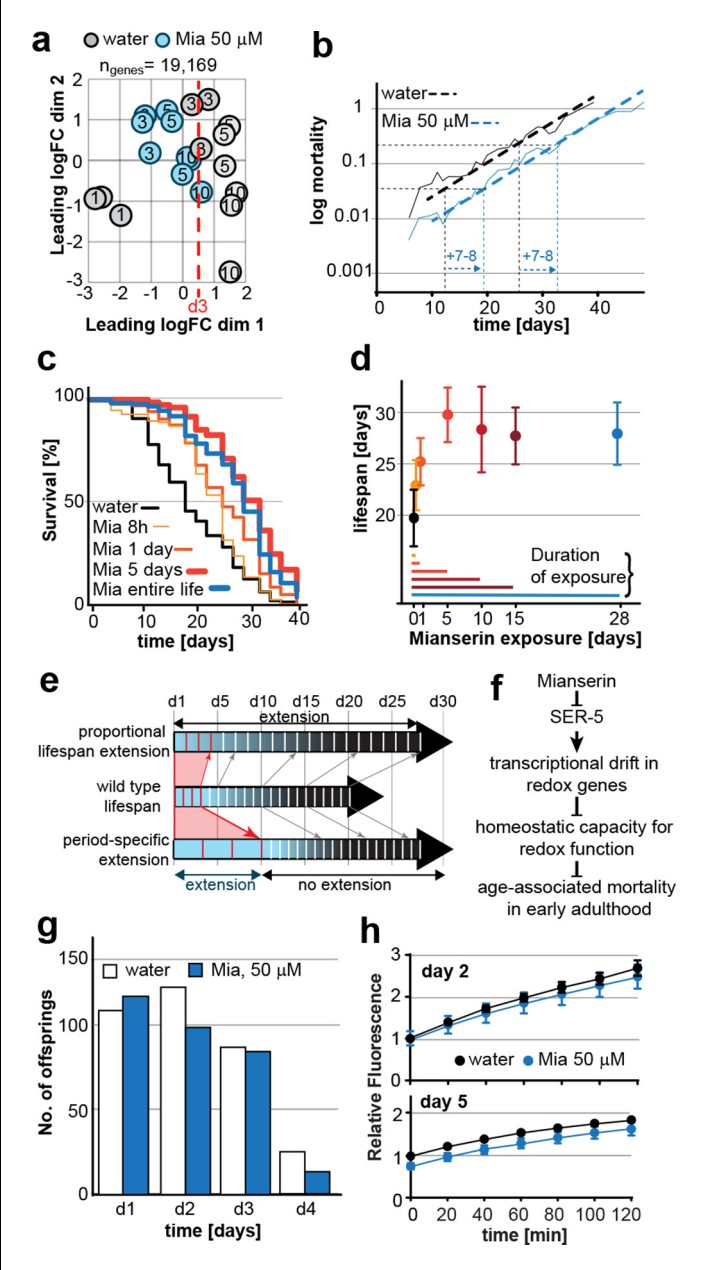

**Figure 6.** Mianserin extends lifespan by specifically slowing age-associated changes in early adulthood. (a) PCA plot of RNA-seq data. Each circle represents one RNA-seq sample with the age, in days, indicated. Mianserin-treated day 10 samples show the same transcriptional age as untreated day 3 animals, dotted red line. (b) Mortality curves (moving average) constructed using Gompertz equation for lifespan experiments from 15 independent experiments of ~100 animals each treated with water or mianserin 50 µM (n>1500 total for each condition). Mianserin treatment causes a 7–8 day parallel shift in log mortality as compared to the water-treated animals. (c) Survival of wt animals treated with mianserin for 8 hr, 1 day, 5 days or throughout life was determined and compared to water treated control animals. Removing mianserin after 8 hr or 1 day lessens its effect on lifespan, while removing mianserin on day 5 or maintaining treatment throughout life showed a comparable effect. (d) Mean survival of wt animals treated with water or mianserin for 8 hr, 1 day, 5, 10, 15 days or throughout life was plotted as a function of mianserin exposure in days. Mianserin treatment for 5 to 10 days was required and sufficient for an optimal lifespan extension. (e) Distinct modes of lifespan extension: Proportional lifespan extension leads to a proportional extension across life whereas period-specific lifespan extension leads to a reduced rate of age-associated degeneration during a specific period only. Mianserin reduces the rate of age-associated changes in early adulthood, thereby postponing mortality levels by 7–8 days causing a 'period-specific

*Figure 6 continued on next page*

*Figure 6 continued*
lifespan extension'. (f) Model for how mianserin modulates age-associated mortality in early adulthood. Blocking serotonergic signaling via SER-5 decreases transcriptional drift-variance with age in redox genes, leading to preserved homeostatic capacity in redox function, which subsequently delays age-associated mortality. (g) Mianserin does not affect reproductive longevity. Wt animals were treated with water or mianserin (50 µM) on day 1 followed by counting the number of viable eggs laid by them on day 1, day 2, day 3 and day 4. h) Chymotrypsin-like 26S proteasome activity measured from wt animals treated with water or mianserin (50 µM) on day 1 followed by proteasome activity assay on day 2 (upper panel) or day 5 (lower panel). Mianserin treatment does not lead to an increase in proteasome activity, unlike long lived germline-less animals. Error bars S.E.M. See *Figure 6—figure supplement 1* for additional data and detailed statistics.
The following figure supplement is available for figure 6:

**Figure supplement 1.** This figure relates to *figure 6* in main text.

We therefore asked whether mianserin slowed age-associated physiological change specifically in early adulthood causing a 7–8 day delay by day 10. If so, mianserin would be expected to specifically lower the mortality rate in young but not in old adults. However, the number of age-associated death events in young adults is too low to directly determine changes in age-associated mortality rates before the age of day 10. As we are comparing mortality in animals either treated with water or mianserin that is added to the same population of worms on day 1 of adulthood, we can confidently state that mortality levels are identical between mianserin-treated and untreated adults at the start of the experiment. Any difference in mortality levels observed from day 1 onwards must therefore be the result of a change in mortality rate by mianserin.

Plotting a mortality curve for over 3,000 mianserin-treated or untreated animals showed a significantly lower mortality level for mianserin-treated animals by day 12 (*Figure 6b*, *Figure 6—figure supplement 1a*). Therefore, mianserin treatment decelerated the rise in mortality levels between day 1 and 12 of adulthood. From then on, the mortality curves were parallel showing a 7–8 day shift in mortality across the remaining lifespan. The parallel nature suggested that mianserin did not affect mortality rates past day 12 and that its effect on lifespan was restricted to the period of early adulthood (*Figure 6b*, *Figure 6—figure supplement 1a*) (*Mair et al., 2003*; *Vaupel, 2010*). Power calculations confirmed that these mortality curves were sufficiently powered to detect a one day difference in lifespan in over 90% of the experiments ($\alpha$=0.01) (*Figure 6—figure supplement 1b*) (*Ye et al., 2014*). These results further supported a model in which mianserin treatment specifically lowered age-associated change in early adulthood, causing a shift in physiology and mortality that can be observed in transcriptomes by day 10.

We reasoned that if the effect of mianserin on lifespan precedes the onset of mortality and is completed by day 10, mianserin treatment beyond day 10 should be dispensable. Alternatively, if mianserin still influenced mortality later in life, shorter exposures would lead to a shorter lifespan extension compared to a lifelong exposure. We therefore limited mianserin exposure to 8 hr, 1, 5, 10 and 15 days and compared their lifespan with animals treated for the entire life (*Figure 6c,d*). Exposing the animals for 5 or 10 days was sufficient to extend lifespan to the same extent as lifelong exposure (*Figure 6c,d*). Shorter exposures (8 hr, 1 day) also extended lifespan, but not by as much, showing that removing mianserin from the culture is an effective means to restrict its action (*Figure 6c,d*). Taken together, these results are most consistent with a model in which mianserin specifically lowers the rate of age-associated change during the first few days of adulthood, thereby extending their longevity (*Figure 6e*) and postponing the onset of mortality. While the change in age-associated mortality rate during early adulthood is too small to be accurately determined, when we measured drift-variance, it allowed us to monitor the age-associated change in the transcriptome during early adulthood (*Figure 6e,f*).

Since the effect of mianserin in early adulthood overlapped with the reproductive period (first 5 days of adulthood), we asked whether mianserin treatment increased reproductive lifespan as has been observed in *tph-1(mg280)* mutants (*Sze et al., 2000*). Mianserin treatment blocks serotonin-induced egg-laying (*Petrascheck et al., 2007*), but had a minor effect on amount or timing of spontaneous egg-laying and brood size (*Figure 6g*). Most importantly, mianserin did not increase reproductive longevity (*Figure 6g*).

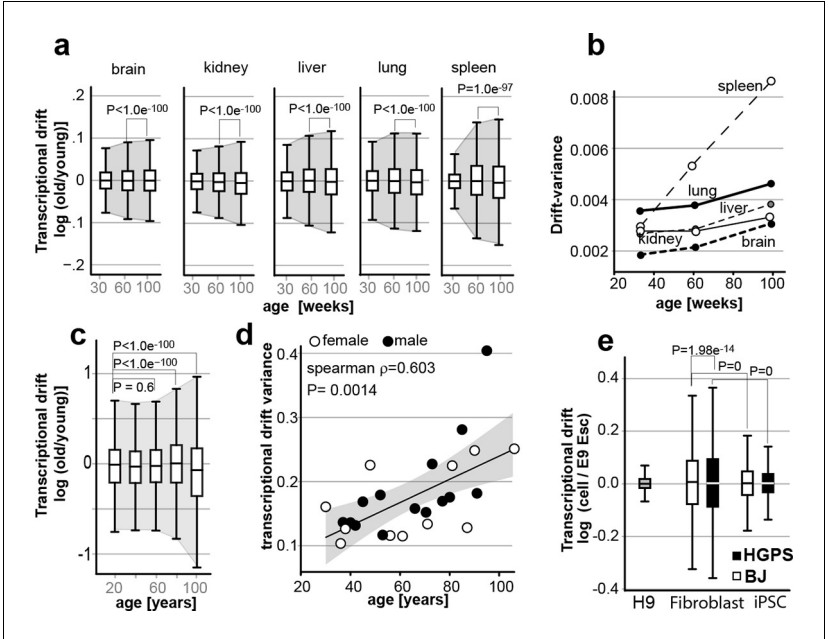

**Figure 7.** Transcriptional drift-variance increases with age in various species. (**a**) Transcriptional drift-variance in gene expression from different mouse tissues aged 13 to 130 weeks. Drift-plots show an increase in drift-variance with age in mouse brain, kidney, liver, lung and spleen (**b**) Drift-variance plotted as a function of age for different organs. To obtain drift-variance values for young animals, a single transcriptome was set aside and used a reference. (**c**) Drift-plot for gene expression from 32 human brains (frontal cortex) plotted as a function of age in years. Data binned in 20-year increments. (**d**) Drift-variance plotted as a function of age in years for individuals. Each dot corresponds to one brain sample (frontal cortex). Shading indicates 95% confidence interval ($\rho$=0.603, P=0.0014). (**e**) Drift plots show a higher transcriptional drift-variance in BJ fibroblasts (BJ) and fibroblasts from Hutchinson Gilford progeria syndrome (HGPS), when compared to H9 embryonic stem cells. Reprogramming the BJ and HGPS cells to induced pluripotent stem cells (iPSCs) leads to a partial reversal of the transcriptional drift-variance to a lower variance corresponding to the young phenotype of the iPSCs. See *Figure 2—figure supplement 1* for additional information on transcriptional drift calculation, and Methods section for transcriptional drift calculation in each figure panel.

We further considered the possibility that mianserin acted by a mechanism similar to lifespan extension by germline ablation (*Figure 6h*). Two previous findings suggested otherwise: i) Lifespan extension by germline ablation depends on *daf-16*, while mianserin does not (*Arantes-Oliveira et al., 2002*; *Petrascheck et al., 2007*); ii) germline ablation increases lifespan of *eat-2 (ad1116)* mutants while mianserin does not (*Crawford et al., 2007*). We measured whether mianserin treatment mimicked the increased proteasome activity observed in *glp-1* mutants (*Vilchez et al., 2012*) (*Figure 6h*). A 24 hr mianserin treatment did not increase the proteasome activity, as measured by a fluorescence-based assay for chymotrypsin-like activity. On day 5, mianserin slightly decreased proteasome activity, consistent with a slight increase in drift-variance in proteasome-related genes (*Figure 6h*; *Figure 6—figure supplement 1c*). We concluded that mianserin specifically lowers the rate of age-associated change in somatic tissues and does not involve a mechanism directly related to the germline.

## Transcriptional drift-variance increases with age in mice and humans

Our data demonstrate that changes in drift-variance provide a metric for aging that correlates with mortality in *C. elegans*. To test whether drift-variance also increases with age in mammals, we re-analyzed published gene expression data-sets obtained from aging mouse tissues, aging human brains, and from fibroblasts derived from Hutchinson-Gilford progeria syndrome patients (*Figure 7*) (*Lu et al., 2004*; *Liu et al., 2011*; *Jonker et al., 2013*). We calculated drift-variances from brain, kidney, liver, lung, and spleen based on gene expression data-sets from mice aged 13, 26, 52, 78, 104

and 130 weeks. We calculated drift-variances using 13-week-old mice as a young reference (see Methods) and pooled mice into age-bins of 30, 60 and 100 weeks to reduce variability. Drift-variance increased in all tissues with age (*Figure 7a*). Compared to the drift-variance changes observed in *C. elegans* (*Figure 2a*), these changes however were small.

Because the 13-week-old mice were used as reference for young age (see methods), the drift-variance in the 30-week-old group including the 13-week-old sample is artificially low (*Figure 6a*, see material and methods). To better reflect the actual variance of the 30-week-old group, we set aside the data of one 13-week-old mouse to use as a young reference and recalculated drift-variances for all samples (*Figure 7b*). This strategy has the advantage that we can observe the real drift-variance for the 30-week-old group by excluding the reference data-set, but has the disadvantage that the results are less robust as they all depend on a single reference sample. Plotting drift-variance for each organ as a function of age confirmed that as mice age, drift-variance increases in all organs (*Figure 7b*). It will be interesting to learn if the different rates by which drift-variance increases in different organs will also be observed in other data-sets.

We re-analyzed the data from Lu et al. that recorded gene expression profiles from 32 human brains aged 26 to 106 years of age (frontal cortex) (*Figure 7c*) (*Lu et al., 2004*). For the first plot, we binned the data into 20-year bins and calculated the overall drift-variance for each 20-year bin. As a young age reference, we used the mean gene expression of adults below 30 (26, 26, 27, and 29) (see Materials and methods). This analysis shows that over the entire population, drift-variance remains relatively stable until the age of sixty, and then starts to rise (*Figure 6c*). We also plotted the drift-variance of each individual as a function of age. This revealed a significant correlation (Spearman, rho=0.6, P=0.0014) between age and drift-variance in the human brain.

Irrespective of the age of the mother, the aging process starts afresh for each new generation. We therefore hypothesized that aging must be reversed with each new generation and asked whether it is possible to reverse increases in drift-variances. To address this question, we re-analyzed the data-set generated by Liu et al. who derived induced pluripotent stem cells (iPSCs) from fibroblasts of healthy controls (BJ) and patients suffering from Hutchinson-Gilford progeria syndrome (HPGS), an accelerated aging syndrome (*Figure 7e*) (*Liu et al., 2011*). As a young-reference to calculate drift-variance, we used human H9 embryonic stem cells (ESC). As expected for a premature aging syndrome, fibroblasts from HGPS patients showed increased drift-variance relative to BJ control fibroblasts (*Figure 7e*). Furthermore, nuclear reprogramming reduced drift-variance in iPSCs to levels closer to those seen in H9 embryonic stem cells. Thus, increases in drift-variance are reversed by nuclear reprogramming in vitro

## Discussion

In this study, we have analyzed the dynamics of aging *C. elegans* transcriptomes and how these dynamics are affected by mianserin treatment. We separate transcriptional changes across groups into those that characterize activation or inhibition of entire pathways (type I) and those that characterize the relative expression levels among genes (type II, transcriptional drift, *Figure 1h,i*). In *C. elegans,* transcriptional drift continuously increases with age across the transcriptome, substantially altering stoichiometric balances observed in young animals (*Figure 2a*). Longevity mechanisms induced by either pharmacologically blocking serotonergic signaling or by blocking insulin signaling by *daf-2* RNAi attenuate transcriptional drift (*Figure 2a,g*). Abolishing lifespan extension by these mechanisms by either blocking serotonergic signaling too late (mianserin, day 5) or by addition of *daf-16* RNAi (*daf-2*) abolished the attenuation of drift-variance (*Figure 2*).

Detailed analysis of redox-related pathways showed that mianserin-reduced drift-variances are associated with improved stress resistance in older age (*Figure 3*). Mutations in the serotonin receptor SER-5 that abolish the effect of mianserin on drift-variance also abolished its effect on stress resistance and lifespan (*Figure 4*, *5*).

Using transcriptome-wide drift-variance values as a metric for age showed that mianserin treatment attenuated the age-associated increase of drift-variance, thereby preserving the characteristics of a much younger (~3 days-old) transcriptome up to chronological day 10 (*Figure 2a*, *6a*). These results showed that mianserin caused a 7–8 days delay in age-associated transcriptional change and suggested that the physiological changes leading to a lifespan extension were already completed by day 10.

**Table 3.** Gene ontology (GO) pathways of relevance to this study that are differentially regulated by mianserin.

| KEGG / GO ID | KEGG / GO Term | Number of Genes observed | Levene's test for variance (Difference in transcriptional drift-variance) Water D1 vs. water $D_x$ | Levene's test for variance (Difference in transcriptional drift-variance) water $D_y$ vs. mianserin $D_y$ |
|---|---|---|---|---|
| Transcriptome | | 19,196 | D3 : P < 1.0E-100<br>D5 : P < 1.0E-100<br>D10: P < 1.0E-100 | D3 : P < 1.0E-100<br>D5 : P < 1.0E-100<br>D10: P < 1.0E-100 |
| KEGG: Cel00030 | Pentose phosphate pathway | 17 | D3 : P = 0.0096<br>D10: P <1.0E-5 | D3 : P <1.0E-4<br>D10: P = 0.01 |
| GO: 0006979 | Response to oxidative stress | 67 | D3 : P <1.0E-10<br>D10: P <1.0E-16 | D3 : P <1.0E-4<br>D10: P = 0.001 |
| GO: 0045454 | Cell redox homeostasis | 52 | D3 : P <1.0E-6<br>D10: P <1.0E-10 | D3 : P <1.0E-4<br>D10: P = 0.029 |
| GO: 006749 | Glutathione metabolism | 13 | D3 : P <1.0E-4<br>D10: P <1.0E-7 | D3 : P =0.041<br>D10: P <1.0E-4 |
| GO: 0007186 | G-protein coupled receptor signaling | 335 | D3 : P <1.0E-24<br>D10: P < 1.0E-100 | D3 : P <1.0E-4<br>D10: P <1.0E-4 |
| GO: 0016209 | Antioxidant activity | 34 | D3 : P <1.0E-8<br>D10: P <1.0E-10 | D3 : P = 0.002<br>D10: P = 0.06 |

Summary of gene changes with RNA-seq transcriptome analysis in *Figure 3b*.

GO ID is the Gene Ontology identification number.

GO Term is the Gene Ontology term for the biological process.

$D_x$ = age in days for the animals indicated, compared with D1 water-treated animals.

$D_y$ = age in days for water- and mianserin-treated animals, compared on the same day of age indicated.

Measuring mortality levels supported this conclusion. By day 12, the entire mortality curve was shifted parallel by 7–8 days (*Figure 6b*) showing that the physiological delay leading to a lifespan extension was already completed. Experiments in which animals were exposed to mianserin for limited periods of time confirmed that mianserin exposure for the first 5–10 days of adulthood was necessary and sufficient to fully extend lifespan (*Figure 6c,d*). The most parsimonious explanation that accounts for all these results is that mianserin treatment slows degenerative processes specifically

**Table 4.** Survival data for paraquat stress resistance assays.

| Strain | Treatment | Conc. [μM] | Treatment added [day] | PQ 100 mM, added [day] | Survival after PQ [%] (expt. 1) | Survival after PQ [%] (expt. 2) | Survival after PQ [%] (expt. 3) | Mean, Survival after PQ [%] | S.D., Survival after PQ [%] | P-value | No. of wells | Total no. of animals |
|---|---|---|---|---|---|---|---|---|---|---|---|---|
| N2 | Water | 0 | d1 | d3 | 70.0 | 43.1 | 62.2 | **58.4** | 13.9 | | 48 | 450 |
| N2 | Mia | 50 | d1 | d3 | 87.3 | 47.9 | 53.9 | **63.0** | 21.3 | 7.72E-01 | 48 | 390 |
| N2 | Water | 0 | d1 | d5 | 55.8 | 56.2 | 66.1 | **59.3** | 5.8 | | 48 | 436 |
| N2 | Mia | 50 | d1 | d5 | 95.5 | 96.1 | 92.0 | **94.5** | 2.2 | 4.24E-03 | 48 | 435 |
| N2 | Water | 0 | d1 | d10 | 63.3 | 37.4 | 41.7 | **47.5** | 13.9 | | 48 | 400 |
| N2 | Mia | 50 | d1 | d10 | 91.9 | 82.1 | 85.4 | **86.4** | 5.0 | 2.85E-02 | 48 | 390 |

Summary of all stress resistance assays performed in *Figure 3c*. The treatments, water or mianserin (Mia), at the indicated concentrations (conc.) were added on day 1 of adulthood. Paraquat (PQ) was added to a final conc. of 100 mM on day 3 (d3), day 5 (d5) or day 10 (d10) and survival after PQ [%] was calculated 24 hr after the respective PQ addition. Mean and standard deviation (S.D.) of survival after PQ [%] were calculated from 3 independent experiments (expts.). P-values were calculated between water and mianserin-treatments on the same day of PQ addition, using unpaired *t-test*. The total number of wells and animals from which data were collected are indicated.

**Table 5.** Survival data for paraquat stress resistance assays, mianserin added on different days.

| Strain | Treatment | Conc. [µM] | Treatment added day | PQ 100 mM, added day | Survival [%] (expt. 1) | Survival [%] (expt. 2) | Survival [%] (expt. 3) | Mean, Survival [%] | S.D., Survival [%] | P-value | No. of wells | Total no. of animals |
|---|---|---|---|---|---|---|---|---|---|---|---|---|
| N2 | Water | 0 | d1 | d10 | 63.30 | 37.44 | 41.72 | **47.48** | 13.86 | | 48 | 400 |
| N2 | Mia | 50 | d1 | d10 | 91.85 | 82.05 | 85.38 | **86.43** | 4.97 | 2.85E-02 | 48 | 390 |
| N2 | Water | 0 | d3 | d10 | 63.97 | 41.25 | 38.35 | **47.85** | 14.02 | | 48 | 403 |
| N2 | Mia | 50 | d3 | d10 | 78.52 | 66.22 | 73.62 | **72.79** | 6.19 | 0.074 | 48 | 378 |
| N2 | Water | 0 | d5 | d10 | 57.31 | 43.83 | 42.57 | **47.90** | 8.16 | | 48 | 387 |
| N2 | Mia | 50 | d5 | d10 | 68.63 | 50.58 | 58.62 | **59.28** | 9.04 | 0.18 | 48 | 398 |

Summary of all stress resistance assays performed in **Figure 3d**. The treatments, water or mianserin (Mia), at the indicated concentrations (conc.) were added on day 1 (D1), day 3 (D3) or day 5 (D5) of adulthood. 100mM Paraquat (PQ) was added on day 10 (D10) and survival [%] was calculated after 24 hr. Mean and standard deviation (S.D) of survival [%] were calculated from 3 independent experiments (expts.). P-value calculated between water and mianserin-treatments using *t-test*. The total number of wells and animals from which data were collected are indicated.

between day 1 and 10, extending the duration of the period of young adulthood thereby postponing the onset of major mortality around mid-life (**Figure 6e,f**).

## Biological interpretation of transcriptional drift-variance

Aging has been shown to cause DNA damage, degeneration of the nuclear architecture, loss of histones, loss of histone modification (**Kaeberlein et al., 1999**; **Scaffidi and Misteli, 2006**; **Burgess et al., 2012**). These changes contribute to the degenerative phenotypes observed with aging (**Mostoslavsky et al., 2006**; **Feser et al., 2010**; **Peleg et al., 2010**). In the present study, we used expression patterns of young adults as a reference to monitor the aging process across the transcriptome. We found that aging causes the expression of genes within functional groups to drift apart, causing a loss of co-expression patterns as observed in young adults. We quantified this phenomenon using drift-variance, defined as the variance in gene expression among genes. It is important to distinguish transcriptional noise, which measures the variance of the same genes among samples (**Bahar et al., 2006**), from transcriptional drift, which measures variance among genes within the same samples. At present it is unclear whether transcriptional drift is the consequence of a regulated program or of degenerative changes in the nucleus that lead to a loss of transcriptional control. Consistent with a regulated program are recent findings that the germline actively represses the activation of heat shock promoters via histone methylation, causing a decline in heat shock capacity (**Labbadia and Morimoto, 2015b**). Consistent with degenerative changes are recent findings that show the loss of histone methylation to cause aberrant gene expression that increases with age leading to a transcriptional drift-like effect (**Somel et al., 2006**; **Mercken et al., 2013**; **Pu et al., 2015**; **Sen et al., 2015**).

Irrespective of whether transcriptional drift is the consequence of a regulated program or a degenerative change, its effect on pathway function is likely to be detrimental. Many physiological processes depend on appropriate stoichiometry of their components. Large and persistent deviations in mRNA balance as measured by drift-variance are likely to result in stoichiometric imbalances in protein complexes, negatively affecting proteostasis as has been recently observed (**Houtkooper et al., 2013**; **Walther et al., 2015**). Our results modulating drift-variance for redox genes via mianserin and SER-5 certainly suggest that the age-associated increases in drift-variance are associated with regulatory decline (**Figures 3**, **5**). Attenuation of transcriptional drift in the redox system was associated with an improved homeostatic capacity, i.e. an improved ability of the redox system to appropriately respond to outward stimuli.

Transcriptional drift also provided a useful concept to analyze aging transcriptomes. Accounting for its effects dramatically simplified what was an initially excessively complex expression pattern (**Figure 1**). Excluding gene expression changes due to drift left a set of genes that changed expression in response to mianserin treatment that was enriched for genes related to stress, innate

Table 6. Survival data for paraquat stress resistance assays.

| Strain | Treatment | Conc. [µM] | PQ conc. [mM] | Survival after PQ [%] (expt. 1) | Survival after PQ [%] (expt. 2) | Survival after PQ [%] (expt. 3) | Survival after PQ [%] (expt. 4) | Survival after PQ [%] (expt. 5) | Survival after PQ [%] (expt. 6) | Mean, Survival after PQ [%] | S.D., Survival after PQ [%] | P-value | No. of wells | Total no. of animals |
|---|---|---|---|---|---|---|---|---|---|---|---|---|---|---|
| N2 | Water | 0 | 0 | 89.9 | 98.9 | 95.8 | 98.2 | 93.9 | 93.2 | 95.0 | 3.4 | | 48 | 548 |
| | Water | 0 | 15 | 76.4 | 88 | 82 | 95.3 | 95.5 | 91.7 | 88.2 | 7.7 | | 48 | 578 |
| | Water | 0 | 25 | 74.2 | 91.3 | 80 | 92.9 | 85.1 | 80.4 | 84.0 | 7.2 | | 48 | 531 |
| | Water | 0 | 50 | 66.2 | 67.8 | 63.8 | 81.9 | 61.6 | 67.8 | 68.2 | 7.1 | | 48 | 530 |
| | Water | 0 | 75 | 50.1 | 61.1 | 44.1 | 64.6 | 42.4 | 51.8 | 52.4 | 8.9 | | 48 | 545 |
| | Water | 0 | 100 | 36.2 | 34.4 | 35.5 | 53.5 | 23 | 54.7 | 39.6 | 12.3 | | 48 | 503 |
| | Mia | 50 | 0 | 100 | 100 | 99.5 | 100 | 100 | 99.2 | 99.8 | 0.3 | 1.71E-02 | 48 | 556 |
| | Mia | 50 | 15 | 100 | 98.2 | 87.6 | 100 | 98.8 | 100 | 97.4 | 4.9 | 3.52E-02 | 48 | 523 |
| | Mia | 50 | 25 | 96.2 | 98.8 | 95 | 98.4 | 100 | 98.2 | 97.8 | 1.8 | 4.54E-03 | 48 | 529 |
| | Mia | 50 | 50 | 95 | 95.9 | 94.5 | 95.3 | 99 | 98.2 | 96.3 | 1.8 | 1.19E-04 | 48 | 536 |
| | Mia | 50 | 75 | 98.9 | 89.3 | 89.4 | 92.6 | 97.5 | 98.1 | 94.3 | 4.4 | 1.29E-05 | 48 | 516 |
| | Mia | 50 | 100 | 97.6 | 90.8 | 90.7 | 69.8 | 93.9 | 95.6 | 89.7 | 10.1 | 1.95E-05 | 48 | 539 |
| ser-1 (ok345) | Water | 0 | 0 | 92 | 71.3 | 89.2 | | | | 84.2 | 11.2 | | 24 | 228 |
| | Water | 0 | 15 | 73.3 | 57.9 | 81.8 | | | | 71.0 | 12.1 | | 24 | 187 |
| | Water | 0 | 25 | 71.3 | 55.9 | 67.8 | | | | 65.0 | 8.1 | | 24 | 209 |
| | Water | 0 | 50 | 54.8 | 46.4 | 42.6 | | | | 47.9 | 6.2 | | 24 | 213 |
| | Water | 0 | 75 | 39.4 | 56.3 | 50.7 | | | | 48.8 | 8.6 | | 24 | 213 |
| | Water | 0 | 100 | 24.2 | 27.3 | 46.6 | | | | 32.7 | 12.1 | | 24 | 224 |
| | Mia | 50 | 0 | 100 | 100 | 100 | | | | 100 | 0.0 | 0.13 | 24 | 215 |
| | Mia | 50 | 15 | 98.8 | 97.7 | 97.6 | | | | 98.0 | 0.7 | 0.06 | 24 | 211 |
| | Mia | 50 | 25 | 97.9 | 94.2 | 98.4 | | | | 96.8 | 2.3 | 1.51E-02 | 24 | 194 |
| | Mia | 50 | 50 | 94.8 | 95.4 | 97 | | | | 95.7 | 1.1 | 4.52E-03 | 24 | 224 |
| | Mia | 50 | 75 | 93.9 | 89.9 | 92.4 | | | | 92.1 | 2.0 | 9.87E-03 | 24 | 232 |
| | Mia | 50 | 100 | 87.4 | 89.6 | 89.5 | | | | 88.8 | 1.2 | 1.45E-02 | 24 | 234 |

*Table 6 continued on next page*

*Table 6 continued*

| Strain | Treatment | Conc. [µM] | PQ conc. [mM] | Survival after PQ [%] (expt. 1) | Survival after PQ [%] (expt. 2) | Survival after PQ [%] (expt. 3) | Survival after PQ [%] (expt. 4) | Survival after PQ [%] (expt. 5) | Survival after PQ [%] (expt. 6) | Mean, Survival after PQ [%] | S.D., Survival after PQ [%] | P-value | No. of wells | Total no. of animals |
|---|---|---|---|---|---|---|---|---|---|---|---|---|---|---|
| ser-2 (pk1357) | Water | 0 | 0 | 100 | 100 | 95.5 | 100 | | | 98.9 | 2.3 | | 32 | 278 |
| | Water | 0 | 15 | 88 | 97.5 | 73.7 | 92.2 | | | 87.9 | 10.2 | | 32 | 239 |
| | Water | 0 | 25 | 90.3 | 100 | 83 | 83.2 | | | 89.1 | 8.0 | | 32 | 206 |
| | Water | 0 | 50 | 76.7 | 87.2 | 73.7 | 62.7 | | | 75.1 | 10.1 | | 32 | 254 |
| | Water | 0 | 75 | 73.9 | 73.2 | 65.2 | 53 | | | 66.3 | 9.7 | | 32 | 220 |
| | Water | 0 | 100 | 72 | 59.6 | 54.4 | 47.7 | | | 58.4 | 10.3 | | 32 | 220 |
| | Mia | 50 | 0 | 98.9 | 100 | 100 | 100 | | | 99.7 | 0.6 | 0.51 | 32 | 231 |
| | Mia | 50 | 15 | 100 | 100 | 100 | 100 | | | 100 | 0.0 | 0.10 | 32 | 255 |
| | Mia | 50 | 25 | 98.9 | 100 | 98.9 | 96.9 | | | 98.7 | 1.3 | 0.10 | 32 | 228 |
| | Mia | 50 | 50 | 100 | 100 | 95.5 | 96.9 | | | 98.1 | 2.3 | 1.71E-02 | 32 | 243 |
| | Mia | 50 | 75 | 98.9 | 95 | 96.8 | 91.8 | | | 95.6 | 3.0 | 6.35E-03 | 32 | 245 |
| | Mia | 50 | 100 | 97 | 88.7 | 92.3 | 95.4 | | | 93.4 | 3.7 | 3.80E-03 | 32 | 210 |
| ser-3 (ad1774) | Water | 0 | 0 | 100 | 100 | 92.6 | | | | 97.5 | 4.3 | | 24 | 176 |
| | Water | 0 | 15 | 89 | 88.5 | 86.8 | | | | 88.1 | 1.2 | | 24 | 174 |
| | Water | 0 | 25 | 90.5 | 85.4 | 85.2 | | | | 87.0 | 3.0 | | 24 | 216 |
| | Water | 0 | 50 | 81.3 | 74 | 72.1 | | | | 75.8 | 4.9 | | 24 | 176 |
| | Water | 0 | 75 | 70.8 | 48.6 | 58.9 | | | | 59.4 | 11.1 | | 24 | 169 |
| | Water | 0 | 100 | 43.7 | 46.5 | 30.1 | | | | 40.1 | 8.8 | | 24 | 140 |
| | Mia | 50 | 0 | 98.2 | 100 | 95.8 | | | | 98.0 | 2.1 | 0.88 | 24 | 176 |
| | Mia | 50 | 15 | 98.9 | 100 | 98.4 | | | | 99.1 | 0.8 | 3.25E-04 | 24 | 228 |
| | Mia | 50 | 25 | 93.8 | 100 | 90.2 | | | | 94.7 | 5.0 | 0.10 | 24 | 173 |
| | Mia | 50 | 50 | 98.1 | 100 | 93.8 | | | | 97.3 | 3.2 | 4.97E-03 | 24 | 174 |
| | Mia | 50 | 75 | 92.4 | 95 | 91.8 | | | | 93.1 | 1.7 | 3.20E-02 | 24 | 197 |
| | Mia | 50 | 100 | 93.4 | 65.6 | 82.8 | | | | 80.6 | 14.0 | 1.92E-02 | 24 | 180 |

*Table 6 continued on next page*

*Table 6 continued*

| Strain | Treatment | Conc. [μM] | PQ conc. [mM] | Survival after PQ [%] (expt. 1) | Survival after PQ [%] (expt. 2) | Survival after PQ [%] (expt. 3) | Survival after PQ [%] (expt. 4) | Survival after PQ [%] (expt. 5) | Survival after PQ [%] (expt. 6) | Mean, Survival after PQ [%] | S.D., Survival after PQ [%] | P-value | No. of wells | Total no. of animals |
|---|---|---|---|---|---|---|---|---|---|---|---|---|---|---|
| ser-4 (ok512) water | | 0 | 0 | 100 | 87.6 | 100 | 98.6 | | | 96.6 | 6.0 | | 32 | 249 |
| | Water | 0 | 15 | 100 | 72.6 | 91.3 | 84.4 | | | 87.1 | 11.6 | | 32 | 262 |
| | Water | 0 | 25 | 98.2 | 67.9 | 72.5 | 85.5 | | | 81.0 | 13.7 | | 32 | 224 |
| | Water | 0 | 50 | 88 | 67.1 | 83.3 | 63.5 | | | 75.5 | 12.0 | | 32 | 229 |
| | Water | 0 | 75 | 69 | 47.2 | 75.8 | 61.4 | | | 63.4 | 12.3 | | 32 | 225 |
| | Water | 0 | 100 | 56.3 | 48.3 | 60 | 43.2 | | | 52.0 | 7.6 | | 32 | 204 |
| | Mia | 50 | 0 | 100 | 95.9 | 100 | 100 | | | 99.0 | 2.1 | 0.49 | 32 | 212 |
| | Mia | 50 | 15 | 96.9 | 97.2 | 97.5 | 97.7 | | | 97.3 | 0.4 | 0.21 | 32 | 228 |
| | Mia | 50 | 25 | 97.5 | 100 | 91.7 | 95.5 | | | 96.2 | 3.5 | 0.11 | 32 | 230 |
| | Mia | 50 | 50 | 93.8 | 96.8 | 96.4 | 95.3 | | | 95.6 | 1.3 | 4.31E-02 | 32 | 261 |
| | Mia | 50 | 75 | 100 | 91.5 | 88.1 | 96.5 | | | 94.0 | 5.3 | 9.66E-03 | 32 | 227 |
| | Mia | 50 | 100 | 96.9 | 86.5 | 90.3 | 89 | | | 90.7 | 4.4 | 3.75E-04 | 32 | 252 |
| ser-5 (ok3087) | Water | 0 | 0 | 98.8 | 92.2 | 99 | | | | 96.7 | 3.9 | | 24 | 206 |
| | Water | 0 | 15 | 91.1 | 83.6 | 85.5 | | | | 86.7 | 3.9 | | 24 | 230 |
| | Water | 0 | 25 | 86.2 | 71.6 | 88.2 | | | | 82.0 | 9.1 | | 24 | 222 |
| | Water | 0 | 50 | 83.2 | 67.5 | 75.9 | | | | 75.5 | 7.9 | | 24 | 222 |
| | Water | 0 | 75 | 68.4 | 64 | 77 | | | | 69.8 | 6.6 | | 24 | 216 |
| | Water | 0 | 100 | 65.1 | 58.2 | 62.9 | | | | 62.1 | 3.5 | | 24 | 232 |
| | Mia | 50 | 0 | 98.6 | 93.9 | 99.2 | | | | 97.2 | 2.9 | 0.85 | 24 | 248 |
| | Mia | 50 | 15 | 96.2 | 92.9 | 97.4 | | | | 95.5 | 2.3 | 3.90E-02 | 24 | 221 |
| | Mia | 50 | 25 | 95 | 78.4 | 90.9 | | | | 88.1 | 8.6 | 0.45 | 24 | 184 |
| | Mia | 50 | 50 | 89.5 | 77.4 | 82.9 | | | | 83.3 | 6.1 | 0.25 | 24 | 219 |
| | Mia | 50 | 75 | 73.2 | 55.7 | 72.5 | | | | 67.1 | 9.9 | 0.72 | 24 | 213 |
| | Mia | 50 | 100 | 64 | 54.6 | 79.4 | | | | 66.0 | 12.5 | 0.65 | 24 | 200 |

*Table 6 continued*

| Strain | Treatment | Conc. [μM] | PQ conc. [mM] | Survival after PQ [%] (expt. 1) | Survival after PQ [%] (expt. 2) | Survival after PQ [%] (expt. 3) | Survival after PQ [%] (expt. 4) | Survival after PQ [%] (expt. 5) | Survival after PQ [%] (expt. 6) | Mean, Survival after PQ [%] | S.D., Survival after PQ [%] | P-value | No. of wells | Total no. of animals |
|---|---|---|---|---|---|---|---|---|---|---|---|---|---|---|
| ser-5 (tm2647) | Water | 0 | 0 | 97.2 | 97.3 | 96.9 | | | | 97.1 | 0.2 | | 24 | 248 |
| | Water | 0 | 15 | 88.8 | 91.2 | 87.3 | | | | 89.1 | 2.0 | | 24 | 230 |
| | Water | 0 | 25 | 94.4 | 89.9 | 85.8 | | | | 90.0 | 4.3 | | 24 | 227 |
| | Water | 0 | 50 | 79.5 | 84.6 | 81.7 | | | | 81.9 | 2.6 | | 24 | 228 |
| | Water | 0 | 75 | 79.9 | 73.5 | 60.2 | | | | 71.2 | 10.0 | | 24 | 248 |
| | Water | 0 | 100 | 51.6 | 59 | 44.1 | | | | 51.6 | 7.5 | | 24 | 224 |
| | Mia | 50 | 0 | 96.7 | 99.2 | 94.3 | | | | 96.7 | 2.5 | 0.80 | 24 | 233 |
| | Mia | 50 | 15 | 96.7 | 88.4 | 95 | | | | 93.4 | 4.4 | 0.23 | 24 | 246 |
| | Mia | 50 | 25 | 97.2 | 88.5 | 92.4 | | | | 92.7 | 4.4 | 0.49 | 24 | 187 |
| | Mia | 50 | 50 | 83.7 | 87.8 | 85.4 | | | | 85.6 | 2.1 | 0.13 | 24 | 234 |
| | Mia | 50 | 75 | 69.7 | 77.3 | 73.5 | | | | 73.5 | 3.8 | 0.74 | 24 | 203 |
| | Mia | 50 | 100 | 46.4 | 75.1 | 70.3 | | | | 63.9 | 15.4 | 0.30 | 24 | 196 |
| ser-5 (tm2654) | Water | 0 | 0 | 81.5 | 96.3 | 83.4 | | | | 87.1 | 8.1 | | 24 | 232 |
| | Water | 0 | 15 | 68.8 | 86.6 | 75.9 | | | | 77.1 | 9.0 | | 24 | 223 |
| | Water | 0 | 25 | 77.1 | 89.1 | 69.1 | | | | 78.4 | 10.1 | | 24 | 226 |
| | Water | 0 | 50 | 55.2 | 79.8 | 78.4 | | | | 71.1 | 13.8 | | 24 | 254 |
| | Water | 0 | 75 | 47.5 | 42.5 | 55.3 | | | | 48.4 | 6.5 | | 24 | 209 |
| | Water | 0 | 100 | 41.2 | 36 | 45.8 | | | | 41.0 | 4.9 | | 24 | 215 |
| | Mia | 50 | 0 | 83.7 | 96.3 | 90.4 | | | | 90.1 | 6.3 | 0.63 | 24 | 232 |
| | Mia | 50 | 15 | 73.7 | 70.3 | 82.6 | | | | 75.5 | 6.4 | 0.82 | 24 | 232 |
| | Mia | 50 | 25 | 66.9 | 73.7 | 88.2 | | | | 76.3 | 10.9 | 0.81 | 24 | 184 |
| | Mia | 50 | 50 | 54.5 | 68.8 | 54.6 | | | | 59.3 | 8.2 | 0.29 | 24 | 200 |
| | Mia | 50 | 75 | 34.9 | 41.9 | 66.5 | | | | 47.8 | 16.6 | 0.95 | 24 | 227 |
| | Mia | 50 | 100 | 18.2 | 30.6 | 40.7 | | | | 29.8 | 11.3 | 0.22 | 24 | 187 |

*Table 6 continued on next page*

*Table 6 continued*

| Strain | Treatment | Conc. [µM] | PQ conc. [mM] | Survival after PQ [%] (expt. 1) | Survival after PQ [%] (expt. 2) | Survival after PQ [%] (expt. 3) | Survival after PQ [%] (expt. 4) | Survival after PQ [%] (expt. 5) | Survival after PQ [%] (expt. 6) | Mean, Survival after PQ [%] | S.D., Survival after PQ [%] | P-value | No. of wells | Total no. of animals |
|---|---|---|---|---|---|---|---|---|---|---|---|---|---|---|
| ser-6 (tm2146) | Water | 0 | 0 | 98.9 | 96.9 | 98.6 | 100 | | | 98.6 | 1.3 | | 32 | 230 |
| | Water | 0 | 15 | 95.1 | | 89.6 | 96.5 | | | 93.7 | 3.6 | | 32 | 260 |
| | Water | 0 | 25 | 97.7 | 87.5 | 90.3 | 84.8 | | | 90.1 | 5.6 | | 32 | 221 |
| | Water | 0 | 50 | 95.3 | 97.5 | 84.8 | 78.1 | | | 88.9 | 9.1 | | 32 | 256 |
| | Water | 0 | 75 | 84.8 | 87.1 | 77 | 63.5 | | | 78.1 | 10.6 | | 32 | 265 |
| | Water | 0 | 100 | 82.4 | 78.1 | 77.9 | 53 | | | 72.9 | 13.4 | | 32 | 253 |
| | Mia | 50 | 0 | 100 | 93.3 | 100 | 96.9 | | | 97.6 | 3.2 | 0.57 | 32 | 278 |
| | Mia | 50 | 15 | 98.8 | | 96.4 | 92.4 | | | 95.9 | 3.2 | 0.49 | 32 | 230 |
| | Mia | 50 | 25 | 97.9 | 97.5 | 96.4 | 91.3 | | | 95.8 | 3.0 | 0.14 | 32 | 190 |
| | Mia | 50 | 50 | 100 | 100 | 88.6 | 92.5 | | | 95.3 | 5.7 | 0.29 | 32 | 252 |
| | Mia | 50 | 75 | 92.2 | 100 | 88.5 | 88.9 | | | 92.4 | 5.3 | 0.07 | 32 | 242 |
| | Mia | 50 | 100 | 95.6 | 91.3 | 93.4 | 95.7 | | | 94.0 | 2.1 | 4.92E-02 | 32 | 221 |
| ser-7 (tm1325) | Water | 0 | 0 | 68.1 | 73.3 | 94.5 | | | | 78.6 | 14.0 | | 24 | 200 |
| | Water | 0 | 15 | 48.1 | 49.6 | 32.4 | | | | 43.4 | 9.5 | | 24 | 142 |
| | Water | 0 | 25 | 45.7 | 42.9 | 30.9 | | | | 39.8 | 7.9 | | 24 | 152 |
| | Water | 0 | 50 | 38 | 37.8 | 36.5 | | | | 37.4 | 0.8 | | 24 | 152 |
| | Water | 0 | 75 | 16.4 | 20.2 | 41.8 | | | | 26.1 | 13.7 | | 24 | 160 |
| | Water | 0 | 100 | 25.1 | 23.2 | 31.6 | | | | 26.6 | 4.4 | | 24 | 134 |
| | Mia | 50 | 0 | 98.8 | 98.9 | 100 | | | | 99.2 | 0.7 | 0.13 | 24 | 217 |
| | Mia | 50 | 15 | 95.8 | 93.8 | 97.2 | | | | 95.6 | 1.7 | 9.18E-03 | 24 | 212 |
| | Mia | 50 | 25 | 100 | 93.4 | 97.4 | | | | 96.9 | 3.3 | 2.25E-03 | 24 | 193 |
| | Mia | 50 | 50 | 88.5 | 92 | 94.6 | | | | 91.7 | 3.1 | 5.30E-04 | 24 | 179 |
| | Mia | 50 | 75 | 91.3 | 92.4 | 89.4 | | | | 91.0 | 1.5 | 1.37E-02 | 24 | 189 |
| | Mia | 50 | 100 | 96.9 | 91.7 | 81.6 | | | | 90.1 | 7.8 | 8.94E-04 | 24 | 186 |

*Table 6 continued on next page*

*Table 6 continued*

| Strain | Treatment | Conc. [μM] | PQ conc. [mM] | Survival after PQ [%] (expt. 1) | Survival after PQ [%] (expt. 2) | Survival after PQ [%] (expt. 3) | Survival after PQ [%] (expt. 4) | Survival after PQ [%] (expt. 5) | Survival after PQ [%] (expt. 6) | Mean, Survival after PQ [%] | S.D., Survival after PQ [%] | P-value | No. of wells | Total no. of animals |
|---|---|---|---|---|---|---|---|---|---|---|---|---|---|---|
| tph-1 (mg280) | Water | 0 | 0 | 97.2 | 96.1 | 98.2 | | | | 97.2 | 1.1 | | 24 | 148 |
| | Water | 0 | 25 | 66.9 | 67.8 | 76 | | | | 70.2 | 5.0 | | 24 | 156 |
| | Water | 0 | 50 | 52 | 47.1 | 56.9 | | | | 52.0 | 4.9 | | 24 | 164 |
| | Water | 0 | 75 | 32.2 | 34.6 | 48 | | | | 38.3 | 8.5 | | 24 | 148 |
| | Water | 0 | 100 | 12.2 | 6.7 | 42.3 | | | | 20.4 | 19.2 | | 24 | 169 |
| | Mia | 50 | 0 | 94.3 | 100 | 96.9 | | | | 97.1 | 2.9 | 0.96 | 24 | 161 |
| | Mia | 50 | 25 | 90.4 | 58.7 | 78.6 | | | | 75.9 | 16.0 | 0.61 | 24 | 159 |
| | Mia | 50 | 50 | 64.8 | 61.7 | 69 | | | | 65.2 | 3.7 | 2.33E-02 | 24 | 158 |
| | Mia | 50 | 75 | 52.9 | 28.9 | 57.9 | | | | 46.6 | 15.5 | 0.47 | 24 | 143 |
| | Mia | 50 | 100 | 8.7 | 1.8 | 40.6 | | | | 17.0 | 20.7 | 0.85 | 24 | 150 |

Summary of all stress resistance assays performed in *Figure 4a*. The treatments, water or mianserin (50 μM), with their indicated concentrations (conc.) were added on day 1 of adulthood. Paraquat (PQ) was added in the concentration range of 0 to 100 mM on day 5 and survival after PQ [%] was calculated 24 hr later. Mean and standard deviation (S.D.) of survival after PQ [%] were calculated from 3 to 6 independent experiments (expts.). P-values were calculated between water and mianserin-treatments at the same PQ conc., using t-test. The total number of wells and animals from which data were collected are indicated.

**Table 7.** Summary of oxidative stress protection by serotonin antagonists.

| Strain name | Fold change in PQ survival [(Drug/DMSO) -1] Expt.1 | Fold change in PQ survival [(Drug/DMSO) -1] Expt.2 | Fold change in PQ survival [(Drug/DMSO) -1] Expt.3 | Fold change in PQ survival [(Drug/DMSO) -1] Expt.4 | Fold change in PQ survival [(Drug/DMSO) -1] Expt.5 | Fold change in PQ survival [(Drug/DMSO) -1] Expt.6 | Fold change in PQ survival [(Drug/DMSO) -1] Expt.7 | Mean, Fold change in survival after PQ | S.D., Fold change in survival after PQ | P-value |
|---|---|---|---|---|---|---|---|---|---|---|
| **Dihydroergotamine 88 µM** | | | | | | | | | | |
| N2 | 0.62 | 0.70 | 0.79 | 0.19 | 1.75 | 1.43 | | 0.91 | 0.57 | |
| ser-5 (ok3087) | 0.45 | 0.15 | | 0.10 | | | | 0.23 | 0.19 | 3.49E-02 |
| **Metergoline 33 µM** | | | | | | | | | | |
| N2 | 0.54 | 0.57 | 0.68 | 0.94 | 1.24 | 1.67 | | 0.94 | 0.44 | |
| ser-5 (ok3087) | -0.05 | -0.27 | -0.11 | | -0.12 | | | -0.13 | 0.09 | 1.50E-03 |
| **Amperozide 13 µM** | | | | | | | | | | |
| N2 | 0.93 | 0.74 | 0.99 | 0.92 | 2.49 | 0.89 | | 1.16 | 0.66 | |
| ser-5 (ok3087) | 0.30 | 0.03 | | -0.58 | | | | -0.09 | 0.45 | 1.63E-02 |
| **Methiothepin 10 µM** | | | | | | | | | | |
| N2 | 0.80 | 1.08 | 0.95 | 0.36 | 0.77 | 2.94 | 1.39 | 1.19 | 0.89 | |
| ser-5 (ok3087) | 0.07 | 0.10 | | 0.16 | -0.01 | | | 0.08 | 0.08 | 1.24E-02 |
| **Ketanserin 176 µM** | | | | | | | | | | |
| N2 | 0.63 | 0.59 | 1.13 | 1.38 | 0.42 | 1.71 | | 0.98 | 0.51 | |
| ser-5 (ok3087) | -0.41 | -0.14 | 0.01 | | -0.07 | | | -0.15 | 0.18 | 1.91E-03 |
| **Mirtazapine 50 µM** | | | | | | | | | | |
| N2 | 0.8 | 0.7 | 1.1 | 0.4 | 1.0 | 0.8 | 1.5 | 0.89 | 0.35 | |
| ser-5 (ok3087) | 0.0 | -0.1 | | -0.1 | -0.2 | | | -0.11 | 0.07 | 1.92E-04 |
| **LY-165,163 33/PAPP µM** | | | | | | | | | | |
| N2 | 0.48 | 0.49 | 1.00 | 0.94 | 0.53 | 1.40 | | 0.81 | 0.37 | |
| ser-5 (ok3087) | -0.03 | 0.35 | -0.07 | | -0.16 | | | 0.02 | 0.23 | 3.19E-03 |
| **Mianserin 50 µM** | | | | | | | | | | |
| N2 | 1.10 | 1.11 | 1.18 | 0.53 | 3.24 | 1.60 | | 1.46 | 0.94 | |
| ser-5 (ok3087) | 0.14 | -0.18 | | -0.26 | | | | -0.10 | 0.21 | 4.49E-02 |

Summary of all stress resistance assays performed in *Figure 4—figure supplement 1b*. The treatments, DMSO or serotonin antagonists, with their indicated concentrations (conc.) were added on day 1 of adulthood. Paraquat (PQ) (100 mM) was added on day 5 and survival after PQ [%] was calculated 24 hr later. Mean and standard deviation (S.D.) of survival after PQ [%] were calculated from 3 to 7 independent experiments (expts.). P-values were calculated between N2 and mutant strains for fold change values with indicated small molecule treatments using *t*-test.

**Table 8.** Summary of all lifespan data for mianserin.

| | | | Cumulative statistics | | | | Statistics of individual expts. | | | |
|---|---|---|---|---|---|---|---|---|---|---|
| Strain | Small molecule | No. of expts. | Mean lifespan [days] (+Mia/+water) | change in lifespan [%] | S.E.M. | No. of animals (+Mia/+water) | Mean lifespan (days) (+Mia/+water) | change in lifespan [%] | P-value | No. of animals (+Mia/+water) |
| N2 | Mia | 12 | 26.7/19.8 | +35 | ± 7 | 642/577 | 26.4/19.8 | +34 | 1.67E-08 | 77/59 |
| | | | | | | | 25.5/21.5 | +19 | 6.85E-07 | 113/94 |
| | | | | | | | 28.1/20.1 | +40 | 3.71E-14 | 95/104 |
| | | | | | | | 30.6/19.0 | +64 | 3.17E-15 | 57/50 |
| | | | | | | | 26.8/21.5 | +25 | 1.87E-11 | 149/145 |
| | | | | | | | 22.6/16.6 | +27 | 1.61E-23 | 151/125 |
| snt-1 (md290) | Mia | 3 | 20.9/18.2 | +15 | ± 2 | 236/231 | 23.3/19.9 | +17 | 1.84E-05 | 86/90 |
| | | | | | | | 17.3/15.4 | +12 | 1.18E-02 | 79/80 |
| | | | | | | | 22.1/19.3 | +15 | 2.4E-03 | 71/61 |
| unc-26 (e205) | Mia | 3 | 25.0/26.7 | -7 | ± 7 | 135/165 | 27.8/26.9 | +3 | 0.53 | 54/68 |
| | | | | | | | 22.2/26.5 | -16 | 4.52E-02 | 14/24 |
| | | | | | | | 26.5/25.3 | +5 | 0.52 | 67/73 |
| ser-5 (ok3087) | Mia | 3 | 23.4/22.2 | +5 | ± 5 | 496/458 | 23.6/20.6 | +15 | 4.19E-02 | 152/144 |
| | | | | | | | 26.4/26.2 | +1 | 0.85 | 174/144 |
| | | | | | | | 20.1/19.8 | -1 | 0.25 | 170/170 |

Summary of all lifespan experiments performed in **Figure 5e,f** and **Figure 5—figure supplement 1a**. N2 and mutant strains were treated with 50 µM mianserin (Mia) on day 1 and lifespan [days] was scored until 95% of animals were dead in all tested conditions. Cumulative statistics and statistics of individual experiments are shown. Mean lifespan [days], change in lifespan [%] and S.E.M. for mianserin-treated (+Mia) and water-treated (+water) animals from multiple, independent experiments (expts.) are shown. Change in lifespan [%] and P-values for individual experiments were calculated using the Mantel–Haenszel version of the log-rank test. Number of animals in individual experiments and all experiments combined are shown.

immunity, aging and the xenobiotic response. With the exception of the xenobiotic response, which is expected to be triggered by addition of a foreign substance such as mianserin (*Figure 2f*), all other functions have been linked to serotonin signaling (*Table 1*) (*Zahn et al., 2006*; *Petrascheck et al., 2007*; *Rangaraju et al., 2015a*).

Further, in accordance with the hypothesis that increases in drift-variance are a signature of aging in the transcriptome, we find that drift-variance is attenuated by two longevity mechanisms (mianserin and *daf-2* RNAi) across large sections of the transcriptome. Many of the age-associated changes that were reversed by mianserin were also reversed by *daf-2* RNAi (58%). This overlap is consistent with chemical epistasis experiments. Treating *daf-2(e1370)* mutants with mianserin causes only a partial extension of lifespan (11% instead of 31%) (*Petrascheck et al., 2007*) consistent with the idea that many of the genes attenuated by mianserin treatment are already attenuated in *daf-2 (e1370)* mutants and thus do not further contribute to a lifespan extension. It should be noted that age-associated increases in drift-variance do not contradict the idea that transcription factors regulate longevity. Activation of DAF-16 target genes by *daf-2* RNAi prevent age-associated drift of thousands of genes, thus resulting in a net decrease of drift, even though a transcriptional program has

been induced (*Figure 2g*). Our experiment did not address the questions whether increasing drift-variance beyond what occurs naturally with age accelerates aging and whether attenuation of transcriptional drift-variance is universal to all longevity mechanisms.

At this point, it is prudent to mention possible pitfalls associated with transcriptional drift analysis. Drift-variance calculations require data-sets that include multiple ages (3 or more) as direct statistical comparisons to the young-reference are not permissible. Furthermore, in the context of GO annotations, it is important to realize that if a given GO annotation contains significant numbers of mis-annotated genes, these genes may change expression in a different direction giving the erroneous impression of transcriptional drift. To account for these effects in our study, we i) used the experimentally determined oxidative stress signature derived from Olivera et al (*Figure 3e*), and ii) used a robust Levene's test to determine statistical differences. The robust Levene's test uses a 10% trimmed mean, which removes large outliers such as those that would be expected by mis-annotation. These safeguards, however, are only effective if the number of mis-annotated genes is small relative to the total number of genes.

Conceptually, transcriptional drift is not a biomarker for aging. It is a metric for aging similar to lifespan measurements that can be used to monitor age-associated physiological changes on the molecular level within groups of genes. Lifespan measurements record the fraction of organisms alive in different cohorts at any given time to compare rates of aging, while drift-variance allows a similar comparison based on transcriptional drift-variance. What made drift-variance measures essential for the present study was that it allowed us to monitor age-associated physiological changes in young animals, at a time when age-associated mortality levels are too low to be accurately determined (see below).

## Period-specific lifespan extension

Measuring lifespan of mianserin-treated and untreated *C. elegans* revealed a mean lifespan extension of 7–8 days (*Figure 2*). Lifespan measurements detect differences after the majority of the animals have died and make no statements about the period during which the relevant physiological events that lead to an increase in lifespan occur (*Figure 2c,e*) (*Mair et al., 2003*; *Partridge and Gems, 2007*). The finding that transcriptional drift values in mianserin-treated animals already showed a 7–8 day delay in physiological change as early as day 10 suggested a model in which the physiological events responsible for the 7–8 days lifespan extension take place (and conclude) prior to day 10 (*Figure 2a*, *6a,e*).

Determining mortality levels at different ages confirmed this model. Mianserin or water is added on day 1 of adulthood to the same preparation of N2 animals. The mortality levels of both cohorts (water, mianserin) are therefore identical at the start of the experiment. Thus, the lower mortality level observed on day 12 in mianserin-treated animals is the result of a lower mortality rate prior to day 12 (*Figure 6b*). Furthermore, mianserin ceases to affect mortality rates past day 12 as evident by highly parallel mortality curves (*Figure 6b*). As with the results obtained with drift measurements, the most plausible explanation is that mianserin treatment specifically decelerates the rise in mortality in young adults leading to a lower mortality level sometime between day 10 to day 12 that persists throughout life, ultimately revealing itself in a 7–8 day lifespan extension (~30–40% increase in lifespan) (*Figure 6b*).

Analysis of drift-variance, PCA, mortality and survivorship independently arrive at the same 7–8 days delay in physiology, either measured as a feature of transcriptomes or by recording death times. All methods suggest that the delay is completed before day 10 or 12 and therefore occurs during early adulthood. We further experimentally confirmed this suggestion by showing that treatment for the first five or ten days of life was necessary and sufficient to achieve the same lifespan extension observed with lifelong treatment (*Figure 6c,d*).

Even though this period exactly overlaps with the reproductive period, the effect of mianserin appears to be specific to somatic tissue (*Figure 6g,h*). In contrast to germline ablation, mianserin extends lifespan of *daf-16* mutants but not of *eat-2* mutants (*Crawford et al., 2007*; *Petrascheck et al., 2007*; *Vilchez et al., 2012*) and does not increase proteasome activity as observed in *glp-1* mutants (*Figure 6h*). It is still possible that the mianserin-induced lifespan extension interacts or depends on the germline, but if it does, the connection is more indirect potentially similar to what has been observed for dietary restriction (*Crawford et al., 2007*).

Lifespan extension mechanisms that decelerate the rate of mortality are generally interpreted as slowing the aging process, while a parallel shift as the one we observe with mianserin is interpreted as a constant risk factor that causes a proportional shift in the overall risk of death (*Mair et al., 2003*; *Harrison et al., 2009*; *Vaupel, 2010*; *Kirkwood, 2015*). Our data do not challenge any of these prior interpretations, but add a further possibility. Parallel shifts may also be brought about by a period extension in which the rate of age-associated physiological change is specifically lowered in young adults. Age-associated mortality in young adults is very low compared to extrinsic mortality factors and thus changes in age-associated mortality rates are difficult to reliably determine (*Partridge and Gems, 2007*; *Beltran-Sancheza et al., 2012*). Specific changes in mortality rates during early adulthood therefore can go unnoticed but manifest themselves later as parallel shifts at the time when age-associated mortality levels are sufficiently high to be reliably determined. Whether the attenuation of physiological changes specific to young adults that affects later mortality, as seen for mianserin, is the equivalent of slowing aging in young adults is a debate for the general aging community.

In summary, this work describes the phenomenon of transcriptional drift and how it can be used as a metric for aging. Using this metric, we show that blocking serotonergic signals by mianserin delays age-associated physiological changes such as transcriptional drift and mortality exclusively during early adulthood, thus extending the duration of this period and postponing the onset of age-associated mortality.

## Materials and methods

### Measurement of transcriptional drift and drift-variance

Analyzing the RNA-seq data in aging *C. elegans*, we observed dramatic changes in the transcriptome with age. We simply termed these changes 'transcriptional drift', to emphasize the ambiguity of these changes. These changes could either be the result of regulated changes as part of a biological program, or caused by a progressive loss of transcriptional control with age. Note that a progressive loss of transcriptional control does not necessarily have to result in random changes. A gene that is continuously activated in young animals may be less activated in older animals due to a progressive functional decline in the transcriptional machinery. Thus, a gradual loss of transcriptional control would cause an age-associated decline in expression of that gene in a non-random fashion. Conversely, repressive chromatin is lost with age leading to increases in transcription that are repressed in young animals. As most physiological processes depend at least to some degree on transcriptional regulation, we propose that expression changes of genes within the same pathway that go into opposing directions (*drift-variance increases*) are detrimental for the functionality of the pathway (as seen for redox pathways in *Figure 3b*). These changes may also allow us to indirectly track the functional decline by measuring transcriptional drift.

### Calculating transcriptional drift and drift-variance

Transcriptional drift (td) is the change in transcript level of a gene at a given age from its level in young animals ("young reference"). As all the subsequent calculations depend on the age chosen for "young reference" we made sure to indicate the age used as a "young reference" for each plot (see below). For all the *C. elegans* work, the "young reference" age was day 1, at the onset of reproductive maturity in adulthood.

For any gene x, transcriptional drift (td) is defined as (*Equation 1*).

$$td_{gene\,x} = \left( \frac{\text{No.of transcripts}_{age[t]}}{\text{No.of transcripts}_{young\,reference}} \right) \tag{1}$$

or, which is the same as

$$td_{gene\,x} = \left( \frac{\text{cpm}_{age[t]}}{\text{cpm}_{young\,reference}} \right) \tag{2}$$

where, 'cpm' stands for counts per million; 't' stands for time in days, weeks or years, dependent on the organism.

**Table 9.** List of small molecules and chemicals used in this study with information

| Molecule name | CAS number | Catalog number | Manufacturer |
|---|---|---|---|
| Mianserin HCl | 21535-47-7 | 0997 | Tocris |
| Mirtazapine | 85650-52-8 | M3368 | LKT Laboratories |
| Dihydroergotamine mesylate | 6190-39-2 | 0475 | Tocris/R&D systems |
| LY-165,163/PAPP | 1814-64-8 | S009 | Sigma |
| Mirtazapine | 61337-67-5 | M3368 | LKT labs |
| Metergoline | 17692-51-2 | M3668 | Sigma |
| Ketanserin tartarate | 83846-83-7 | S006 | Sigma |
| Methiothepin mesylate | 74611-28-2 | M149 | Sigma |
| Amperozide HCl | 86725-37-3 | 2746 | Tocris/R&D systems |
| Paraquat (Methyl viologen) | 1910-42-5 | AC227320010 | Acros Organics |
| FUDR | 50-91-9 | F0503 | Sigma-Aldrich |
| DMSO | 67-68-5 | 472301 | Sigma-Aldrich |

*Equation 1* normalizes the level of transcription for all genes to 0 for a young animal. Note: If several biological replicates are available for the age of the young reference, a variance for the young age can be calculated (see the section below titled 'Variance for "the young reference"').

To evaluate changes in co-expression, we calculated the *drift-variance (dv)* (*Equation 3*) over a group of *n* genes with transcriptional drift-values ranging from $td_{i=1}$ to $td_n$.

$$drift\ variance = \frac{1}{n-1}\sum\nolimits_{i=1}^{n}\left(td_i - \overline{td}\right)^2 \tag{3}$$

Thus, if genes maintain a youthful co-expression pattern, *drift-variance* stays relatively small. If large fractions of genes within a GO or an entire transcriptome change expression in opposing directions, the *drift-variance* increases, suggesting a loss of youthful co-expression patterns as shown in *Figure 1h,i*.

**Table 10.** List of mutant and fluorescent strains outcrossed and used in this study.

| Strain name | Genotype | No.of times outcrossed | Gene name | Transgene | Allele | Parent strain(s) |
|---|---|---|---|---|---|---|
| VV78 | *unc-26 (e205) IV* | 4 | *unc-26* | | *e205* | CB205 |
| VV80 | *snt-1 (md290) II* | 4 | *snt-1* | | *md290* | NM204 |
| MT15434 | *tph-1 (mg280) II* | 4 | *tph-1* | | *mg280* | MT15434 |
| DA1814 | *ser-1 (ok345) X* | 10 | *ser-1* | | *ok345* | DA1814 |
| OH313 | *ser-2 (pk1357) X* | 4 | *ser-2* | | *pk1357* | OH313 |
| DA1774 | *ser-3 (ad1774) I* | 3 | *ser-3* | | *ad1774* | DA1774 |
| AQ866 | *ser-4 (ok512) III* | 5 | *ser-4* | | *ok512* | AQ866 |
| VV130 | *ser-5(ok3087) I* | 4 | *ser-5* | | *ok3087* | RB2277 |
| FX2647 | *ser-5 (tm2647) I* | 0 | *ser-5* | | *tm2647* | FX2647 |
| FX2654 | *ser-5 (tm2654) I* | 0 | *ser-5* | | *tm2654* | FX2654 |
| FX2146 | *ser-6 (tm2146) IV* | 0 | *ser-6* | | *tm2146* | FX2146 |
| DA2100 | *ser-7 (tm1325) X* | 10 | *ser-7* | | *tm1325* | DA2100 |

**Table 11.** List of oligos used for qRT-PCR

| Gene name | qRT-PCR forward primer (5'-3') | qRT-PCR reverse primer (5'-3') |
| --- | --- | --- |
| sod-1 | CGTAGGCGATCTAGGAAATGTG | AACAACCATAGATCGGCCAACG |
| sod-2 | TTCAACCGATCACAGGAGTC | GCTCCAAATCAGCATAGTCG |
| sod-3 | ATGGACACTATTAAGCGCGA | GCCTTGAACCGCAATAGTG |
| sod-4 | ATGTGGAACTATCGGAATTGTG | GGTTGAGATTGTGTAACTGGA |
| sod-5 | ATGGAGACTCAACCGATGG | GACCACGGAATCTCTTCCT |
| ctl-1 | AATGGATACGGAGCGCATAC | AACCTTGAGCAGGCTTGAAA |
| ctl-2 | TGATTACCCACTGATCGAGG | GCGGATTGTTCAACCTCAG |
| ctl-3 | CAATCTAACGGTCAACGACAC | CATTGGATGTGGTGAGCAG |
| prdx-2 | CATTCCAGTTCTCGCTGAC | ATGATGAAGAGTCCACGGA |
| prdx-3 | GTTCCGTTCTCTTGGAGCTG | CTTGTTGAAATCAGCGAGCA |
| prdx-6 | GGAGAACAATGGCTGATGC | ATCTGAACATGGCGTTTGC |
| hsp-16.1 | ACCACTATTTCCGTCCAGCT | TGACGTTCCATCTGAGCCAT |
| hsp-16.11 | ACCACTATTTCCGTCCAGCT | TGACGTTCCATCTGAGCCAT |
| hsp-16.2 | TCGATTGAAGCGCCAAAGAA | TCTCTTCGACGATTGCCTGT |
| hsp-16.41 | TCTTGGACGAACTCACTGGA | TCTTGGACGAACTCACTGGA |
| hsp-16.48 | CTCATGCTCCGTTCTCCATT | GAGTTGTGATCAGCATTTCTCCA |
| hsp-16.49 | CTCATGCTCCGTTCTCCATT | GAGTTGTGATCAGCATTTCTCCA |
| crn-3 | GAATGCACTCATGAACAAAGTC | TAATGTTCGACTGATGAACCG |
| rcq-5 | GATGTTAGAGCTGTAATTCACTGG | ATCTCTTCCAGCTCTTCCG |
| rpl-6 | TTCACCAAGGACACTAGCG | GACAGTCTTGGAATGTCCGA |

## Variance for the "young reference"

If multiple replicate data-sets for the "young reference" age are available, it is possible to plot *drift-variance* for the young reference as well. There are two ways to incorporate multiple "young reference" data-sets, each of which has its advantages or disadvantages.

Method #1 uses all "young reference" samples to calculate a mean gene expression level for each individual gene to generate the "young reference" values for *Equation 1*. Method #1 will result in a drift-variance for the "young reference" age as well, but this drift-variance is too small and should not be used for statistical comparisons due to circular referencing. The advantage of method #1 is that the results for all subsequent ages are more robust as the inclusion of several "young reference" samples thereby reducing the overall noise (used in *Figures 2a,g*, *3b*, *7a,c,e*).

Method #2 allows calculating a real drift-variance value for young animals by setting aside one or several samples as the "young reference." These samples are only used as references and therefore do not contribute to the drift-variance in each plot. For the remaining experimental replicates of the same age, transcriptional drift is then calculated using *Equation 1* without including any of the "young reference" samples." This will result in a *drift-variance* greater than 0 for the youngest age and show how much drift varies between young animals. Method #2 has the disadvantage that if there are only few young reference samples are available, and only one is used as a young reference, all values of the graph depend on a single reference sample. We used this method #2 to calculate the variances for *Figure 7b,d*. The case of 7d was ideal as there were 4 samples less than 30 years of age which were set aside as reference and that allowed us to calculate the "young reference"-mean over all 4 samples. As drift-variances for these 4 samples are artificially low due to self referencing they were excluded from the plot. Ideally, an experiment would have 4–6 gene expression replicates for the "young reference" age, in which case, half of them could be used as references, the others as experimental samples.

How transcriptional drift and variance relate to measures like fold-changes in transcription is shown in Supplementary *Figure 2a–d*. To determine whether the differences in variance were

statistically different, we used the Brown-Forsythe version of the Levene's test, as implemented in STATA software.

## Calculations for drift-plots in Figures

*Figure: 1g:* Volcano plot used mean cpm values from all three biological replicates.

The 0 line (young reference, day 1 expression, yellow line) indicates the expected expression level for young day 1 adult animals.

Black: Each dot represents one of for the 3,367 genes that significantly change expression with age between day 1 and day 3. The $-\log_{10}$(P-value) of the P-value comparing day3 $_{water}$ vs day 1 $_{water}$ is shown as a function of the the $\log_2$(cmp day 3 $_{water}$ / cpm day1 $_{water}$).

Blue: Same 3,367 genes as above. However the $-\log_{10}$(P-value) comparing day3 $_{mianserin}$ vs day 1$_{water}$ is shown as a function of the the $\log_2$(cmps day 3 $_{mianserin}$ / cpm day1 $_{water}$). Note: both datasets (black and blue) use identical y- coordinates to demonstrate the reduction in age-associated changes upon mianserin-treatment. (cpm stands for: counts per million).

*Young Reference:* To obtain a 'young reference' value for each individual gene the mean expression level across all three biological replicates of young day 1 old water-treated *C. elegans* animals was calculated.

*Figure 1h, i:* Drift plots for genes involved in oxidative phosphorylation (KEGG pathway: cel 00190) and the lysosome (KEGG pathway: cel 04142). Only one out of three replicates was used to generate these plots. Transcriptional drift for oxidative phosphorylation and lysosomal genes (line graphs) was calculated using *Equation 1* and plotted as a function of *C. elegans* age (gray lines). At each age, the transcriptional drift-variance across all genes within the pathway was calculated using *Equation 2* and plotted as Tukey-style box plots omitting outliers. Tukey plots were superimposed over the line graphs. See *Equation 1, 3*. Outliers were only omitted for graphical purposes but not for statistical testing (robust Levene's test). The lines for each gene were included in these two plots, superimposed on the Tukey-style box plot to illustrate the significance and utility of the box plots in visualizing transcriptional drift.

*Young reference:* As a "young reference" value for each individual gene, the expression level of young day 1 old water-treated *C. elegans* animals was used. Only replicate #1 of our data-set was used.

*Figure 2a:* Drift plots for all 19,196 genes in our data-set of water-treated control and mianserin-treated animals. Tukey plots show drift-variance calculated for the entire transcriptome (*Equation 3*). See *Equation 1, 3*. Outliers were only omitted for graphical purposes, but not for statistical testing (robust Levene's test).

*Young reference:* To obtain a "young reference" value for each individual gene, the mean expression level across all three biological replicates of young day 1 old water-treated *C. elegans* animals was calculated.

*Figure 2b:* Drift plots show transcriptional drift on day 5 for 19,196 genes as a function of mianserin concentration. For each concentration, drift-variances were calculated for 5-day-old animals that were treated with increasing concentrations of mianserin on day 1, and plotted as Tukey-style box plots as a function of mianserin concentrations, excluding outliers. Outliers were only removed for graphical purposes but not for statistical testing (robust Levene's test).

*Young reference:* To obtain a "young reference" value for each individual gene, the mean expression level across all three biological replicates of young day 1 old water-treated *C. elegans* animals was calculated.

*Figure 2d:* Drift plots show transcriptional drift on day 10 of adulthood for 19,196 genes as a function of age when mianserin-treatment was started. Tukey plots show *drift-variance* calculated for the entire transcriptome on day 10 (*Equation 3*) as a function of age at which mianserin-treatment was initiated.

*Young reference:* To obtain a "young reference" value for each individual gene, the mean expression levels across all three biological replicates of young day 1 old water-treated *C. elegans* animals was calculated.

*Figure 2f:* $\log_2$ fold changes in expression for each gene shown in the y-axis were calculated by the formula: $y = \log_2$(cpm treatment $_{day\ 10}$/cpm water $_{day\ 1}$).

*Figure 2g:* The data from Murphy et al. were dowloaded from the Princeton Puma database. Expression values were calculated using the following variables in the data-set: expression value =

ch1netmean/ch2normalizednetmean. Drift plots for control- RNAi, *daf-2(RNAi)* treated and *daf-16 (RNAi); daf-2(RNAi)* treated animals were plotted as transcriptional *drift-variance* as a function of *C. elegans* age. To plot *drift-variance* for the entire transcriptome as function of age in days, we binned the data as follows. Day 0 (8 hr), day 1 (24 hr), day 2 (28 hr, 40 hr, 52 hr), day 4 (72 hr, 96 hr), day 6 (144 hr, 196 hr).

*Young reference:* As a "young reference" value for each individual gene we used the expression level at 8 hr of age. The young reference was determined for each RNAi treatement specifically (control RNAi, *daf-16(RNAi); daf-2(RNAi), daf-2(RNAi)*.

*Figure 3e:* The log fold gene expression with age was calculated for each of the 252 genes that are known to be upregulated in response to oxidative stress and for each of the 88 genes known to be downregulated in response to oxidative stress. We then performed a linear fit for each set of genes for water-treated (gray) and mianserin-treated (blue) samples. Shaded region shows the 95% confidence interval.

*Figure 7a, b:* 7a) Drift plots showing transcriptional drift and drift-variance in different tissues across different mouse ages. For each age, the drift-variance was calculated across the entire transcriptome (*Equation 3*) and plotted as Tukey-style box plots omitting outliers. As only three mice were available for each age, we pooled two ages for each age bin.

7b) Drift-variance for each tissue as a function of age.

*Young Reference: 7a:* To obtain a "young reference" value for each individual gene, the mean expression level across all three biological replicates of young 13-week-old mice was calculated for each tissue.

*Young Reference 7b:* To obtain "young reference" values for each individual gene, we used one single 13-week-old replicate as a "young reference" from each tissue. The data from the "young reference" did not contribute to the graph and thus show a real transcriptional *drift-variance.*

*Figure 7c, d:* 7c). Drift plots showing transcriptional *drift-variance* in human gene expression data from frontal cortices as a function of age. For 7c, the data were pooled into 20 year bins.

7d) Plots drift-variance calculated based on *Equation 3* as a function of age for each sample individually.

*Young Reference:* To obtain "young reference" values for each individual gene, the mean gene expression levels was calculated averaging expression levels from 4 samples aged 25 to 29 years and used as the "young reference" value in *Equation 1*.

*Figure 2—figure supplement 1:* e) The transcriptional drift plots were constructed by using the GEO data-sets GSE21784 and GSE46051, which are independent publicly available data-sets for aging *C. elegans*.

f) The transcriptional drift plots were constructed by sub-sampling the data from our RNA-seq. We randomly assigned half of all genes (out of 19,196) to one of 10 gene-sets each containing ~1000 genes (5%) and plotted the drift-variance for each set. All 10 sets look nearly indistinguishable to *Figure 2a*.

*Figure 2—figure supplement 2:* f) The drift plot was constructed by removing all the genes from our data-set that were not detected in the sterile CF512 strain, thereby removing genes likely resulting from eggs and germline.

g) The drift plot was constructed by removing all genes from our data-set that were detected by RNA-seq in isolated *C. elegans* eggs.

k) Gene-sets enriched in AFD neurons (left plot), ASE neurons (middle plot) and NSM neurons (right plot) were used to construct drift plots based on their expression in our data-set.

## Principle component analysis

Principal components analysis plot (*Figure 6a*) was generated from the counts table using multidimensional scaling as implemented by the plotMDS function in the edgeR package, which computes inter-sample distances as the root-mean-square of the 500 genes with the largest log2 fold-changes between each pair of sample (the 'leading log fold-change").

## Chemicals

Solvents used to prepare stock solutions: Paraquat was dissolved in water; mianserin was dissolved either in water or DMSO as mentioned; Mirtazapine, Dihydroergotamine, LY-165,163/PAPP,

Mirtazapine, Metergoline, Ketanserin, Methiothepin, and Amperozide were dissolved in DMSO; FUDR was dissolved in S-complete (**Table 9**).

## Strains

Detailed descriptions of all strains used in this study are tabulated below. All strains were back-crossed at least 4 times with the N2 Bristol strain. All strains were maintained as described in (**Brenner, 1974**). The strains with name starting with VV were generated by outcrossing to N2 Bristol strain in our lab (**Table 10**).

## Lifespan assay and analysis

Lifespan assays were conducted in 96-well plates as described in (**Solis and Petrascheck, 2011**; **Rangaraju et al., 2015b**). Briefly, age-synchronized animals were cultured in S-complete media containing *E. coli* OP50 as feeding bacteria (~2 × 10$^9$ bacteria mL$^{-1}$) in 96-well plates, such that 5–15 worms are in each well. At the L4 stage, FUDR was added to prevent animals from producing offspring. Solvent (water or DMSO) or small molecules were added on day 1 of adulthood, exposing the worms to control or compound treatment until the end of the assay. When used, DMSO was kept to a final concentration of 0.33% v/v. Live animals were scored visually, based on movement induced by shaking and application of light to each well. Animals were scored three times a week, until 95% of animals were dead in all the tested conditions. Statistical analysis was performed using the Mantel–Haenszel version of the log-rank test.

## Stress resistance assays

Resistance to oxidative stress was determined by measuring survival of mianserin-treated and untreated worms after a 24 hr exposure to the ROS-generator paraquat (Methyl viologen). Experimental worm cultures were set up as described in Lifespan assays. For dose response assays, paraquat was added to a final concentration of 0, 25, 50, 75, 100 mM on day 5 of adulthood. For paraquat time-course experiment (**Figure 3c**), paraquat was added 3 days, 5 days, or 10 days after addition of mianserin on day 1 of adulthood. For mianserin time-course experiment (**Figure 3d**), 50 µM mianserin was added on day 1, day 3 or 5 of adulthood, followed by 100 mM paraquat on day 10. For all experiments, survival of worms was assessed 24 hr after paraquat addition and expressed as the percentage of live versus total animals.

## RNA-sequencing (RNA-seq) transcriptional studies and data analysis

Mianserin-induced changes in transcription were determined by RNA-seq. A total of 12 conditions were tested each run in three biological replicates. N2 worms were cultured in 96-well plates as described in (**Solis and Petrascheck, 2011**). Animals in cohort #1 were treated on day 1 with water (solvent) or 50 µM mianserin, and harvested on day 3, 5, and 10 of adulthood. Animals in cohort #2 were treated with water (solvent control) or mianserin (2, 10, or 50 µM) on day 1 of adulthood and harvested on day 5. Animals in cohort #3 were treated with water (solvent) or 50 µM mianserin on day 1, day 3 and day 5 and harvested on day 10 (See **Figure 1a**). RNA was also harvested from untreated day 1 adults, to obtain the "young reference". Harvested animals were washed three times in ice cold Dulbecco's phosphate buffer saline and frozen in liquid nitrogen. A parallel lifespan assay was conducted for all cohorts to ensure mianserin action. Three biological replicates were harvested for every cohort. To extract RNA, frozen worms were re-suspended in ice-cold Trizol, zirconium beads, and glass beads (cat # 03961-1-103 and cat # 03961-1-104) in the ratio of 5:1:1 respectively, and disrupted in Precellys lysing system (6500 rpm, 3 x 10 s cycles) followed by chloroform extraction. For RNA-seq, the extracted RNA was precipitated and purified further using Qiagen RNAeasy Mini kit columns (cat # 74104). RNA was precipitated using isopropanol and washed once with 75% ethanol. Integrity of the RNA was confirmed with a Bioanalyzer (Agilent Technologies, Santa Clara, CA, USA). To prepare the library, 100 ng of total RNA per sample was processed using NuGEN Encore Complete DR RNA-seq Prep Kit (NuGEN; San Carlos; CA, USA), as per manufacturer's instructions. The libraries were sequenced using v2 sequencing chemistry in a HiSeq2000 platform (Illumina, San Diego, CA, USA). A single-read sequencing approach was used with 100 cycles, resulting in reads with a length of 100 nucleotides each. Libraries containing their own index sequences were sequenced in a multiplex manner by pooling six libraries per lane. Resulting sequences

were obtained after 20–30 million reads per sample. Sequence data were extracted in FASTQ format and used for data analysis.

## RNA-seq data analysis

RNA-seq data were analyzed by aligning the reads to the *C. elegans* reference genome and transcriptome from WormBase using Tophat 2 (*Kim et al., 2013*), and unambiguously mapped reads were counted for each annotated gene in each sample (*Lawrence et al., 2013*). Data were normalized for sequencing depths (counts per million, cpm) but not for gene length as no comparisons between genes within the same sample were made. The quasi-likelihood F-test from the edgeR package (*Robinson and Oshlack, 2010*; *Lund et al., 2012*) was used to test these counts for statistically significant differential gene expression between water- and mianserin-treated samples, while controlling for expression differences between the 3 biological replicates. We performed multiple testing correction by using the Benjamini-Hochberg procedure to compute a false discovery rate (FDR) value for each gene, and we considered an FDR less than 10% to be significant (*Benjamini and Hochberg, 1995*; *Zhang et al., 2009*).

## Quantitative real-time PCR (qRT-PCR) and data analysis

All qRT-PCR experiments were conducted according to the MIQE guidelines (*Bustin et al., 2009*), except that samples were not tested in a bio-analyzer, but photometrically quantified using a Nanodrop. All strains were cultured in 96-well plates as described in (*Solis and Petrascheck, 2011*). Water (solvent) or mianserin were added on day 1 of adulthood and worms were harvested on day 5. RNA was extracted as described above, followed by DNAse (Sigma, cat # AMPD1-1KT) treatment and reverse transcription using iScript RT-Supermix (BIO-RAD, cat # 170–8841) at 42°C for 30 min. Quantitative PCR reactions were set up in 384-well plates (BIO-RAD, cat # HSP3901), which included 2.5 µl Bio-Rad SsoAdvanced SYBR Green Supermix (cat # 172–5264) or Kapa SYBR Fast master mix (cat # KK4602), 1 µl cDNA template (2.5 ng/µl, to final of 0.5 ng/µl in 5 µl PCR reaction), 1 µl water, and 0.5 µl of forward and reverse primers (150 nM final concentration for BIO-RAD SYBR mix and 75 nM final for Kapa SYBR mix) (see Table below for oligo pairs used for qRT-PCR of genes tested). Quantitative PCR was carried out using a BIO-RAD CFX384 Real-Time thermocycler (95°C, 3 min; 40 cycles of 95°C 10 s, 60°C 30 s; Melting curve: 95°C 5 s, 60°C- 95°C at 0.5°C increment, 10 s). Gene expression was normalized to three reference genes, *rcq-5, crn-3* and *rpl-6,* using the BIO-RAD CFX Manager software. Statistical significance was determined using Student's *t*-test (*Table 11*).

## Measurement of 26S proteasome activity

Wild-type N2 worms were cultured as described (*Solis and Petrascheck, 2011*). Water or Mianserin 50 µM were added on day 1 and 26S proteasome activity was assayed on day 2 and day 5 using the Millipore Proteasome activity kit (cat# APT280), following manufacturer's protocol. Equal number of worms per condition were washed off culture media using ice cold Dulbecco's phosphate buffer saline and freshly lysed using Precellys system (6500 rpm, 3 x 10 s cycles) in assay buffer (25 mM HEPES, pH 7.5, 0.5mM EDTA, 0.05% NP-40, and 0.001% SDS (w/v)). Chymotrypsin-like proteasome activity in the lysates were assessed using the Suc-LLVY-AMC substrate and fluorogenic AMC substrate cleavage was measured in 20 min intervals for 120 min. A subset of lysates were pre-incubated with Lactacystin (12.5 µM final) to ensure specificity of AMC cleavage by 26S proteasome. The amount of cleaved AMC fragments were quantified using TECAN xfluor safire II system at excitation of 360 nm and emission of 480 nm. The resulting readings were normalized to the total protein content in the samples measured using Bradford assay.

## Mortality curve and probability of detection

Mortality curves were generated based on the life table provided in *Figure 6—figure supplement 1*, tabulating death times of 15 independent experiments performed over 5 years. Each experiment consisted of 2 cohorts (water or 50 µM mianserin) and each cohort consisted of ~100 worms each amounting to ~1500 worms per condition. Power of detection was determined by Monte-Carlo simulations using a parametric model with parameters derived from our survival data of a cohort of over 5,026 N2 animals. The power of detection plot (*Figure 6—figure supplement 1*) shows the

probability to detect a true lifespan extension with a significance level α=0.01 as a function of percent increase in lifespan for an experiment consisting of n animals. An accuracy of 1 day is the equivalent of a 5% increase in lifespan.

## Acknowledgements

This work was funded by grants to MP, from the NIH (DP2 OD008398), a grant from The Ellison Medical foundation (AG-NS-0928-12), an MDA Development Grant for SR, and an NSF GRFP Fellowship for GMS. SEE is supported by The Ellison Medical Foundation (AG-NS-0950-1), and by a Baxter Foundation Young Faculty Award. Some strains were provided by Shigen-Japan or the CGC, which is funded by NIH Office of Research Infrastructure Programs (P40 OD010440). We thank Jim Priess (U. Washington), and Bruce Bowerman (U. Oregon) for advice, and Dr. Veena Prahlad, Dr. Eros Lazzerini Denchi, Dr. Maria Carretero, Dr. Bruno Conti, Dr. Andrew Chisholm and Caroline Broaddus for critical reading of the manuscript.

## Additional information

### Funding

| Funder | Grant reference number | Author |
|---|---|---|
| NIH Office of the Director | New Innovator Award | Michael Petrascheck |
| Ellison Medical Foundation | New Scholar in Aging Award | Michael Petrascheck |
| Muscular Dystrophy Association | Development Grant/ Postdoctoral Fellowship | Sunitha Rangaraju |

The funders had no role in study design, data collection and interpretation, or the decision to submit the work for publication.

### Author contributions

SR, conceived, designed and planned the studies; outcrossed the mutant strains, and performed the RNA-seq experiments and lifespan experiments; performed RNA work and qRT-PCR; performed the stress resistance assays; performed data; interpreted the results, prepared the figures and tables, and wrote the paper analyses; Conception and design, Acquisition of data, Analysis and interpretation of data, Drafting or revising the article; GMS, performed RNA work and qRT-PCR;performed the stress resistance assays, Acquisition of data, Analysis and interpretation of data, Drafting or revising the article; RCT, performed data analyses; Conception and design, Analysis and interpretation of data, Drafting or revising the article; RLG-A, performed the stress resistance assays; performed data analyses; Acquisition of data, Analysis and interpretation of data, Drafting or revising the article; LK, contributed intellectually to the study; Conception and design, Analysis and interpretation of data, Drafting or revising the article; SEE, performed and analyzed FUDR embryo imaging experiments; Acquisition of data, Analysis and interpretation of data, Drafting or revising the article; ABN, conceived, designed and planned the studies; Conception and design, Drafting or revising the article, Contributed unpublished essential data or reagents; DRS, conceived, designed and planned the studies; Conception and design, Analysis and interpretation of data, Drafting or revising the article; MP, conceived, designed and planned the studies;performed data analyses; interpreted the results, prepared the figures and tables, and wrote the paper, Conception and design, Acquisition of data, Analysis and interpretation of data, Drafting or revising the article

### Author ORCIDs

Ryan C Thompson, http://orcid.org/0000-0002-0450-8181

## Additional files

### Major datasets

The following datasets were generated:

| Author(s) | Year | Dataset title | Dataset URL | Database, license, and accessibility information |
|---|---|---|---|---|
| Michael Petrascheck, Ryan C Thompson | 2016 | Suppression of Transcriptional Drift Extends C. elegans Lifespan by Postponing the Onset of Mortality | http://www.ncbi.nlm.nih.gov/geo/query/acc.cgi?acc=GSE63528 | Publicly available at the NCBI Gene Expression Omnibus (Accession no: GSE63528). |

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
