## [Decision Letter]

Thank you for submitting your work entitled "Suppression of Transcriptional Drift Extends Lifespan by Prolonging *C. elegans* Youth" for peer review at *eLife*. Your submission has been favorably evaluated by K VijayRaghavan (Senior editor, who also served as Reviewing editor) and three reviewers.

The reviewers have discussed the reviews with one another and the Reviewing editor has drafted this decision to help you prepare a revised submission. If the requirements specified in the comments below can be addressed within the next two months, we will be happy to consider a substantially revised version of this manuscript. However, given the concerns raised by the reviewers, you may wish to consider another venue for the publication of this work. If you are unable to comply with the recommendations of the reviewers, please let us know if you wish to withdraw the work from further consideration.

Summary:

In this manuscript the authors present a framework in which age-specific transcriptional variance is examined as a biomarker of aging that is susceptible to modulation through serotonergic signaling. The authors interpret the effects of the drug mianserin, which inhibits serotonergic signaling and extending lifespan, as a manipulation that prolongs a putative youthful state. They assert that such data provide the first evidence that lifespan extension can be achieved by prolonging a specific period of life. Overall, we appreciate the approach and the questions that have been asked here, particularly that this way of viewing aging gets out of the rut of thinking that specific single genes in are key to aging.

While the ideas in the manuscript are thoughtful and creative, there are several major considerations that, in our opinion, severely limit the impact of the manuscript and call into question the authors' main conclusions. In our view, these need to be clearly and satisfactorily addressed. A manuscript with these major revisions will be re-examined by the reviewers and editor.

Essential revisions:

1) The paper would be improved considerably if the speculations were toned down, and the data were allowed to speak for themselves. Most importantly, the ideas and hypotheses presented in the manuscript are not tested in a rigorous manner. The authors use predominantly descriptive data to support their own interpretation of the biology without asking whether they are also consistent with alternative interpretations. For example, the authors claim that if their measure of drift variance increases with age it is a sign of loss of homeostasis and of an increase in aging-related loss of transcriptional control. However, because the authors are examining among-gene variance (more on that below), this observation is also consistent with a model in which gene expression changes with age in a strongly regulated manner. Some genes may increase/decrease expression with age to promote reproduction, for example, or to modulate adult-specific behaviors. Even if such gene expression was regulated with 100% precision (i.e., reproducibly identical among individuals) then the variance in expression among genes will increase with age. Slowing the aging process (i.e., reducing the slope of that increase) would also reduce the rate of increase in drift-variance. This would occur without any specific youthful phase and completely in the absence of any sort of aging-related loss of transcriptional control. In other words, a model that is conceptually the opposite of the one proposed by the authors is consistent with many aspects of their observed data. There is no attempt or framework to distinguish this (and other) alternative hypotheses. For example, models of homeostasis can be separated from models of defined, aging-related changes by also examining variance within genes.

2) Consideration of alternative models is also lacking from other aspects of the manuscript. For example, the authors interpret the data in Figure 3 as implying that mianserin early in life is the key aspect of the manipulation that leads to enhanced stress resistance. While this interpretation is consistent with the observations, so is a more parsimonious one; that increased time on mianserin increases stress resistance regardless of the age at which it is given. (See also other concerns, below, on mianserin experiments).

3) Many studies in different species have described versions of the concept that parameters associated with fidelity of gene expression decline with age. These include evidence concerning proteostasis and stress response "collapse", as well as transcriptional changes, noise, and epigenetic alterations associated with age. Many of these changes occur during early adulthood in the worm. These findings are cited to an extent, but not comprehensively and not in a way that fully develops this idea. This is important because it is not clear whether transcriptional drift truly represents something new, or yet another parameter reflecting this regulatory decline. That doesn't mean the marker isn't useful, but it needs to be put into a more appropriate context in order to judge its significance.

4) The claim is that drift variance is a biomarker of aging, i.e., a measure that more accurately predicts "physiological age" or future lifespan better than chronological age. However, a population measure is used, which confounds several influences. True biomarkers of aging must be defined at the level of the individual, and they are ideally based on rates of change across ages (repeated measures) and are validated by establishing that they predict remaining lifespan better than chronological time itself. Even if we accept this as a biomarker, the human data are confusing, where transcriptional drift variance doesn't appear to exhibit significant differences across test subjects until the age of 60 (Figure 6). Does this imply that humans are in their youthful state until 60? These data appear to undermine the notion that this measure is an evolutionary conserved bio-marker of aging.

5) We do not understand the assertion that there exists a qualitatively distinct, youthful phase. If we accept drift variance as a biomarker of aging, all of the data appear to change relatively continuously throughout the measurement period. It is formally possible that if the authors' normalize gene expression to L2 data that age 1 day adults would have lots of "drift." In other words, the work is lacking specific criteria upon which to define this hypothetical new youth stage. The lack of a clear definition for "youth" leads to confusion and ambiguity. For example, the authors state: "Simply put, by chronological age of day 10, mianserin treated animals had yet to progress transcriptionally beyond the physiological age of day 3 [based on data in Figure 2]." However, they do not point out that mianserin treated individuals on day 10 have higher variance than they do on day 2 (even though they do have similar levels of variance seen in younger control populations), which implies that they have "gotten worse" over time. If we accept that they putatively remain in a youthful stage, then we must accept that they were hyper-youthful early on and that youth was subject to the same process of change in drift variance as aging. So how is this different from aging itself?

6) Furthermore, additional insight into the youthful state could have been provided by including data beyond age 10 days. At 10 days old, more than 95% of the worms are still alive. Have authors measured what is the transcriptional drift at 50% of the survival? Or 90% of the survival? Does transcriptional drift increase linearly, or does it plateau at a high level later in life, as the authors seem to suggest. If transcriptional drift mostly happens in early life, can this be reliably used as a biomarker of aging?

7) Importantly, it seems like a stretch to conclude that mianserin specifically slows aging during youth, at least based upon the evidence here. As noted above, many parameters degenerate during early adulthood, but aging biomarkers become manifest later. Isn't it more likely that drift and other markers of gene expression changes reflect an early degeneration in control that will play out in its biological effects later, rather than a specific marker of "aging" in the young? To address this idea definitively, we would need to know whether other interventions have the same effect on transcriptional drift, or act on downstream or different parameters. Is mianserin unique, or is protecting the animal from this regulatory decline the most effective way to delay aging, and one that is seen in most contexts of long life?

8) The worm is different in many ways biologically at day 1 compared to later days. Reproduction ceases day 4-6, and body size increases during adulthood. How might these affect "drift", or vice-versa? Does mianserin affect these parameters? How can claims be made about effects on aging early in life without considering this biology? It is either disingenuous or naïve to describe young adulthood in terms of a period of aging while ignoring the biological events that are occurring, particularly the cessation of reproduction.

Some additional important points:

9) What are the genes and GO terms that are changed between mianserin and untreated animals? Is this transcriptional signature like *daf-2*, or like CR treatment? This would be good to know and discuss *before* using transcriptional drift analysis, which looks at the rate of those changes, since later some of these targets (?) are discussed in the redox potential experiments. This information would also help us understand what the underlying mechanisms of mianserin's delay of aging are. And how many of these expression differences are attributed to eggs rather than soma?

10) Most of these experiments in worms, with the exception of the published data on *daf-2* and *daf-16* RNAi (which use worms without sperm), use animals that are still fertile. Since that is the case, how can the authors know which transcripts are due to changes in the soma, which presumably are the ones they are interested in because of aging, or due to changes in the eggs? It seems that the authors could do some comparisons with the control worms from the sterile animals to address this point; otherwise, one set of experiments that uses mianserin-treated sterile animals would be expected.

11) While the SER-5 data are clear, we did not fully understand why other receptors would not be needed for stress protection which they are still needed for the lifespan effect.

12) Does *daf-*16 suppress mianserin effects or act in parallel?

Conclusion and Reiterations:

At the cost of being repetitive we summarise our consultations on the major points of 'drift' and 'youth'.

13) Drift

If drift indeed is a marker of something useful, this needs to be stated very clearly for all the concerns given above. The idea of "drift" seems to be another manifestation of regulatory/homeostatic collapse, a phenomenon already described in various other ways". We do not see direct evidence that transcriptional "noise" indeed arises from dysregulation and that it is a general phenomenon across the genome. Even in a case that "drift" represents loss of homeostasis or failure of transcriptional regulation rather than "programmed" expression pattern, we still have reservations in using the transcriptional drift as a bio-marker of aging. As stated above, conceptually, a bio-marker should have a predictive power of an individual and it is not clear how a metric that "describes" variance in a population can be considered a good way to predict an individual's physiological age. We are concerned that the idea of drift as elaborated in the manuscript has the potential to really muddy the waters when it comes to an already confusing body of literature that examines gene dysregulation with aging. At the very least, a discussion of this matter should be expanded to include a broader survey of the existing literature and a more rigorous analysis of the data that includes consideration of alternative models. (For example, are there longevity mutants that do not suppress drift and instead use a different mechanism?)

14) Youth (and Mianserin)

In our view to the idea of an effect on "youth" is not established, given the concerns we have stated above, and particularly given the coincidence with reproductive cessation. In addition to the eggs, reproduction itself is probably the most influential program that would change gene expression in young adults. Changes in developing eggs would account for a huge fraction of the transcriptional changes in the "aging" worms that have nothing to do with aging. The mianserin experiments were done with WT worms (and even FUdR would not change this point), so the authors need to seriously address this concern. That would be distinct from drift. If reproduction were somehow accounted for, we might be more willing to approach the authors' interpretation of the variance data. Even if reproduction is addressed, the notion of a specific effect on youth is still tenuous unless the authors show that an intervention that can affect aging late in life can affect "drift" at that point. So, the 'youth as a developmental stage' idea is not tenable, and this aspect is best left out, unless the issue of reproduction is addressed.

---

## [Author Response]

*[…] While the ideas in the manuscript are thoughtful and creative, there are several major considerations that, in our opinion, severely limit the impact of the manuscript and call into question the authors' main conclusions. In our view, these need to be clearly and satisfactorily addressed. A manuscript with these major revisions will be re-examined by the reviewers and editors.*

Thank you for the comments, giving us an opportunity to revise the paper and taking the time to review our manuscript. We acknowledge that we made several omissions like alternative hypotheses, but only to shorten an already long and somewhat complicated story. We have re-written large sections of the manuscript to incorporate all the changes.

We certainly did not start this project with any preconceived notion or deviated from common practice to prove a point. What concerned us at the outset of this project was that standard gene-expression analysis resulted in a very complex gene expression pattern that was able to support almost any hypothesis for our data-set depending upon what genes are picked and analyzed (data included in the revised manuscript, Figure 1—figure supplement 1). If indeed there is a situation that “muddies” the waters, it is this.

Incorporating the concept of transcriptional drift into the analysis results in a very sensible and comprehensive picture and all the conclusions we present are the direct consequence of this change in view (drift-analysis). In the revised manuscript we included the most important alternative hypothesis that we considered as well as additional experiments to test them. We acknowledge that this manuscript at times is difficult to read due to new ways of addressing expression analysis using variance, non-parametric statistics, log mortality and so forth. We hope that we have done a better job in explaining all our ideas the second time around.

We strived to address all the reviewers concerns, substantially revised the manuscript and included additional experiments. As there were many concerns, and as we try to provide a substantial answer to all of them, this became a rather long response. To make this as easy as possible, we therefore start with answers to points 13 (usefulness of drift) and 14 (youth and eggs). We have started with point 14 which we split into the question of FUDR and RNA coming from egg contamination and the question of “youth” extension. After answering the general points 13 and 14, we will proceed and address all the other major and minor concerns point by point.

*14) Youth (and Mianserin) In our view to the idea of an effect on "youth" is not established, given the concerns we have stated above, and particularly given the coincidence with reproductive cessation. In addition to the eggs, reproduction itself is probably the most influential program that would change gene expression in young adults. Changes in developing eggs would account for a huge fraction of the transcriptional changes in the "aging" worms that have nothing to do with aging. The mianserin experiments were done with WT worms (and even FUdR would not change this point), so the authors need to seriously address this concern. That would be distinct from drift. If reproduction were somehow accounted for, we might be more willing to approach the authors' interpretation of the variance data. Even if reproduction is addressed, the notion of a specific effect on youth is still tenuous unless the authors show that an intervention that can affect aging late in life can affect "drift" at that point. So, the 'youth as a developmental stage' idea is not tenable, and this aspect is best left out, unless the issue of reproduction is addressed.*

13) Drift.

*If drift indeed is a marker of something useful, this needs to be stated very clearly for all the concerns given above. The idea of "drift" seems to be another manifestation of regulatory/homeostatic collapse, a phenomenon already described in various other ways". We do not see direct evidence that transcriptional "noise" indeed arises from dysregulation and that it is a general phenomenon across the genome. Even in a case that "drift" represents loss of homeostasis or failure of transcriptional regulation rather than "programmed" expression pattern, we still have reservations in using the transcriptional drift as a bio-marker of aging. As stated above, conceptually, a bio-marker should have a predictive power of an individual and it is not clear how a metric that "describes" variance in a population can be considered a good way to predict an individual's physiological age. We are concerned that the idea of drift as elaborated in the manuscript has the potential to really muddy the waters when it comes to an already confusing body of literature that examines gene dysregulation with aging. At the very least, a discussion of this matter should be expanded to include a broader survey of the existing literature and a more rigorous analysis of the data that includes consideration of alternative models. (For example, are there longevity mutants that do not suppress drift and instead use a different mechanism?)*

We have further distilled the points 14 and 13 into three main concerns:

Concern 1: (Answer to point 14) The technical issue of the presence of eggs and their possible confounding factor in the presented transcriptomes.

Concern 2: (Answer to point 14) Additional data that support the claim that mianserin extends lifespan by extending the period of young adulthood by modulating the rate of aging exclusively during young adulthood and not at any later age.

Concern 3: (Answer to point 13). The general usefulness of drift and what it can and cannot do.

Concern 1: The technical problem of the presence of eggs and their transcripts in our RNA-seq data

Reiteration of the concern:

Adult *C. elegans* animals contain eggs that are a source of “young RNA”. Extracting RNA from fertile animals containing eggs results in RNA samples containing eggs as well as adult somatic RNA. The reviewers argue that the amount of RNA from eggs is substantial and will potentially blur any gene expression signals across different ages.

Response:

During the design of this study, we have tested mianserin on CF512 animals and mianserin increases the lifespan of CF512 animals, showing that its lifespan extension is not dependent on the method of sterilization. However, because mianserin does not extend lifespan when added after day 5, at the beginning of the study, we suspected a link to reproduction (we now show that mianserin does not affect reproductive longevity (Figure 6). We therefore decided to use FUDR for our RNA-seq study instead of sterile animals, as FUDR-treated animals still have to invest resources and energy into making eggs.

We have included Figure 2—figure supplement 2 to address the concerns relating to the presence of egg RNA in our samples, raised in point 14. The following supplementary data included in the revised manuscript show that the RNA derived from FUDR-treated eggs is minor in our whole animal samples and that it does not affect any statements made about drift.

Figure 1—figure supplement 1 (A-I):

A) Eggs/embryos isolated from day 1 adults that were FUDR-treated at the L4 stage, uniformly arrest around the 400-500 cell stage, right at the ventral closure around “bean stage”. This arrest is permanent and the eggs are still at the stage 72 h later. FUDR-treated eggs show a much shrunken appearance consistent with a smaller cell mass.

B) All FUDR-treated animals have a germline and produce eggs 24 h after addition of FUDR, and these animals are indistinguishable from non-FUDR-treated animals. Of note, we observed many of the reported side-effects of FUDR only to occur when FUDR is used on NGM plates. Use of FUDR in our 96 well culture conditions does not result in side-effects such as developmental delays, lack of a germline or lifespan extension of *gas-1* mutants.

C) Consistent with the embryonic arrest and shrunken appearance, FUDR-treated eggs produce 5 to 6 times less RNA compared to untreated eggs. Extracting RNA from FUDR-treated adults or from their eggs (normalized by number of adults from which RNA were extracted) shows that the RNA content from FUDR-treated eggs makes only ~5% of total RNA from whole animals, while non-FUDR-treated animals contain 25% of RNA arising from eggs.

D) RNA extracted from whole worms (+eggs) treated with FUDR and eggs only, resolved in an agarose gel.

Given these results, the ~5% contamination of egg-RNA still falls within the detection limit of RNA-seq. To account for the contribution of this small percentage of egg RNA in our RNA-seq samples, the following measures were taken. In Figure 2—figure supplement 2, we will account for the contamination of egg RNA by calculating drift using only genes that were found to be expressed in sterile CF512 animals (Figure 2—figure supplement 2) or by excluding genes that are expressed in eggs (Figure 2—figure supplement 2).

Before we explain the recalculations of drift using restricted datasets to omit any influence of eggs, we would like to point out what information can be extracted from Tukey-style plots with respect to transcriptomes. The plot consists of the “whiskers” representing 1.5 x interquartile mean and the “box” which represents the interquartile mean itself (Figure 2—figure supplement 2). In other words, in a drift plot, the box represents the 50% of the transcriptome that changes least, while the whiskers represent the rest of the genes (50% without outliers) that change most (see PMID: 24645192 for details). Every drift plot in our manuscript in which drift changes with age shows that the change is present in the whiskers as well as in the box. Thus, drift is not driven by a few hundred genes that change extremely but is driven by thousands of changes distributed across the entire transcriptome. For the question of RNA from eggs, what this means is that even if we assume the worst possible scenario for our hypothesis that the eggs RNA is the RNA that changes more than any other RNA (in which case the whiskers would represent egg RNA), this would not be sufficient to alter the drift-variance across the entire sample. We chose the Tukey-style plots to avoid the situation where results are driven by extreme outliers or non-normal distributions (which are present in our datasets). Because drift is driven by thousands of gene changes, its results are extremely robust and don’t change even if we introduce massive changes to the subsets of genes we analyze (Figure 2—figure supplement 2). As we see a drift phenomenon even in the interquartile mean (50% of the transcriptome that changes least with age), drift is a global effect across the transcriptome (Figure 2—figure supplement 2) and is resilient to sub-sampling (Figure 2—figure supplement 2).

To test whether the above stated arguments are true, we calculated drift for different gene-sets that should exclude RNA resulting from eggs (Figure 2—figure supplement 2). As the Murphy et al., 2003 data were derived from CF512 animals (sterile), the genes detected in this sample are not of embryonic origin. We therefore excluded all genes not detected by Murphy et al., 2003 from our dataset and recalculated drift. The resulting drift plot confirms our previous results (Figure 2—figure supplement 2). The potential problem with the approach used in (Figure 2—figure supplement 2) is that it only removed eggs/germline genes that are specific for eggs but that it did not remove genes that were present in both, eggs and soma. For Figure 2—figure supplement 2, we removed all the genes that were identified in *C. elegans* eggs by RNA-seq (PMID:25875092), detecting ~7,700 transcripts. Most of the 7,700 transcripts identified in eggs were present in our dataset. Note that in this case, we remove all ubiquitously expressed genes like ribosomal, mitochondrial and similar genes that are present in both embryos and soma. Even though this operation removes only ~7,200 out of 19,196 individual genes, speaking in terms of total RNA, these ~7,200 genes account for 73% of the entire transcriptome in total counts. These genes belong to the class of mostly highly expressed genes. Despite this dramatic reduction in overall RNA, the remaining 11, 904 genes (mostly low expressing genes) again confirm that drift is increasing with age and that day 10 mianserin-treated animals show the same drift variance as day 3 untreated animals. The median expression increase seen in plot Figure 2—figure supplement 2 is an artifact because most of the high expressing genes have been removed and the remaining low expressing genes cannot really decrease any further. However, with respect to each other, the result remains unchanged.

To identify gene-sets that could not have originated from the FUDR-treated eggs, we exploited the fact that FUDR causes a relatively specific arrest in embryonic development before the birth of AFD, ASE and NSM neurons. To ascertain that FUDR arrests embryonic development before the birth of these neurons, we imaged eggs of *C. elegans* that carried a P*gcy-8*::GFP transgene (AFD marker) that were FUDR-treated or not (Figure 2—figure supplement 2). Eggs from untreated animals showed a clear expression of the marker, while FUDR-treated eggs did not (Figure 2—figure supplement 2 I, J (n>100)). FUDR did not repress the expression of the P*gcy-8*::GFP transgene in adults, showing that the lack of a signal in FUDR-treated eggs is due to a uniform arrest before the neurons are born and not due to inhibition of expression of the transgene by FUDR. As AFD neurons are born before ASE and NSM neurons, these results suggested that none of these three neurons are present in FUDR-treated eggs (Sulston 1983). Gene-sets for genes that are highly enriched in these three neuron types (AFD, ASE, NSM) have been published (PMID17606643, 25372608). We therefore constructed drift-plots for our datasets only using genes highly enriched in AFD, ASE or NSM neurons. Even for these highly restricted gene-sets, transcriptional drift dramatically increased with age and was repressed by mianserin (Figure 2—figure supplement 2).

In summary: FUDR-treated eggs arrest, produce little RNA and constitute only a minor fraction of total RNA. Drift plots using gene-sets excluding genes found in eggs show the same results. We hope that this series of experiments convince the reviewers that contamination from egg-derived RNA is minimal in FUDR-treated animals, and immaterial of the method of sterilization (the murphy data show drift too) and that our conclusions are not influenced by a small subset of eggs RNA genes (~5%), as drift is very robust and replicable across many adult-specific subsets of genes.

Concern 2: Period extension of young adulthood and the relation to the germline

In response to: “youth as a developmental stage' idea is not tenable, and this aspect is best left out, unless the issue of reproduction is addressed”. We agree that youth is not a developmental stage and removed the word “youth” as well as the word “stage” from the manuscript. We conducted two additional experiments, which support the period-specific extension and therefore, we have not removed the concept of a period extension. The results clearly came out in support of a model in which mianserin specifically slows age-associated changes in young adults before the onset of mortality in middle age (literally day 10 is the middle age of a 21 day lifespan).

For a longevity mechanism to act during a specific period of time (period extension), the idea of youth to be a specific stage, as we framed it before, is not necessary. The only requirement for a period extension model is the existence of processes that occurs for a limited window during the life of an organism and causes some kind of degeneration during that period. If such a process occurs in young adults, it will lead to a higher mortality level by the onset of major mortality around mid-life. Thus, reducing the degenerative changes will extend lifespan but the effect will occur only if initiated during the specific period and the effect will be restricted to that period. To frame it in colloquial terms for mianserin, mianserin specifically slows aging in young adults and thereby lowers the mortality level by the time the animals reach middle age.

What is conceptually new? What is not new is that aging starts in young animals. What is new is that certain mechanisms specifically slow the age-associated physiological decline during young adulthood and not throughout life. Aging in early adulthood can be specifically slowed down without slowing down aging in later in life. The effect of the intervention on longevity has a beginning and an end, and the end is long before the animal dies. In the case of mianserin, its effect on longevity ends even before the onset of major mortality (see below for implications on treating age-related disease).

The mentioning of Gompertz curves by one of the reviewers made us realize that mortality analysis provides an independent method other than drift to test the period extension model. In a mortality plot a period-specific lifespan extension would lead to a lower mortality rate in young adults followed by a parallel shift in mortality levels at later ages. If period specific lifespan extensions exist, as we claim, why do mortality curves thus far failed to detect such an extension mechanism? The technical problem is that mortality curves are “blind” for the period of young adulthood (before day 10 in worms). The uncertainties in mortality levels and rates for young adults are too large to detect a deceleration of the age-associated mortality rate, for the following technical reasons: First, mortality curves cannot reliably measure changes in age-associated mortality rates in young adults, because the age-associated contribution of mortality is too small in relation to non-aging-associated mortality, i.e. accidental deaths (see below, Technical note). However, as remarked by others, it is important but difficult to separate aging-associated deaths from non-aging-associated deaths (PMID: 17994065). Non-aging-associated deaths pose a serious problem because in most cases it is nearly impossible to distinguish an accidental death from a death due to aging and even small mistakes will have an enormous influence. Using very high number of animals will not solve this problem as the number of non-aging-associated deaths will increase proportionally. Second, since age-associated mortality levels are so low in young adults, it is generally not possible to ascertain that two populations started out at the same mortality level. Thus different mortality levels observed later in life, when mortality levels are high enough to be determined reliably, could always be the consequence of different mortality levels at the start, rather than a change in mortality rates.

Drift-variance suggested that the action of mianserin decelerated the age-associated changes before day 10, a time at which it is very difficult to measure mortality levels as well as mortality rates to the precision necessary to independently test the results obtained by drift. The following experiments circumvent the above mentioned problems and provide new evidence based on demographic mortality analysis, that suggest that mianserin acts by decelerating age-associated physiological changes exclusively during the first 5 to 10 days of life (Figure 6).

Evidence that mianserin extends lifespan by decelerating age-associated changes during young adults only: At the end of a lifespan assay, one can detect that mianserin treatment caused the animals to die 7-8 days later, which is to say that the physiological decline leading to death was delayed by 7-8 days. A lifespan assay makes no statement on when during the life of the animal this delay occurred. Analyzing drift-variance detected a physiological delay of 7-8 days by day 10 suggesting the possibility that the physiological delay happened and concluded before middle age (middle age= day 10, literally half of the mean lifespan of 21 days).

To test this possibility with an independent method, we used demographic mortality analysis as a way to test the period-extension model. By treating one half of a synchronized population of isogenic worms (N2) with water and the other with mianserin, we ascertain that the mortality levels were identical for both mianserin and water treated control animals at the start of the experiment (day 1). Any difference in mortality levels between the two populations at a later time must therefore be due to different mortality rates. Such a statement for example can never be made when comparing mutants as it is always possible that mutations cause a different mortality level form the start due to slight alterations in development. A period extension model predicts that mianserin treatment will lower the mortality rate during the first few days of adulthood but not later, exerting an anti-aging effect during a specific period of life (young adulthood). This experimental design allows us to distinguish differences in age-associated mortality rates during young adulthood even though age-associated mortality levels at that age are too small to be directly measured (see Technical note for estimates and details).

Combining all lifespan experiments testing mianserin in the past 5 years, in which mianserin was tested using identical conditions (n>3000), we find that by day 12 of adulthood, mianserin-treated animals clearly showed lower mortality levels than water-treated controls. As both cohorts started at an unknown, but identical mortality level, the most plausible explanation is that mianserin lowered the mortality rate during the first 12 days of life leading to a lower mortality level as soon as mortality levels can be reliably measured (day 12). Furthermore, the parallel curves show that after day 12, mianserin has no effect on the mortality rate. Thus, lowering the mortality rate in young adults, without effects later, leads to a 7-8 day parallel shift in the lifespan curve. Mortality analysis, similar to drift-variance, shows that the 7-8 day shift in physiology, that ultimately causes a lifespan extension, is already present before the onset of major mortality in middle age (day 10, literally half the lifespan of 21 days),supporting the period-specific extension model.

As one of the reviewers correctly pointed out, parallel shifts have been interpreted as “non-aging,” and we agree with this interpretation. However the “non-aging” statement is only true for the ages during which mortality analysis can reliably determine mortality levels. Our result show that period extensions can give rise to parallel shifts in lifespan curves if they slow down aging in young adults, a period during which age-associated mortality rates are very difficult to determine.

A second experiment further added evidence that the effect of mianserin on lifespan ends by day 10. We treated *C. elegans* with mianserin for restricted periods of time (8h, 1, 5 10 and 15 days) and show that exposure for 5 to 10 days is necessary and sufficient to fully extend lifespan and that longer treatments do not further extend lifespan. This finding is again consistent with the model that mianserin slows the rate of age-associated decline during a very restricted period in life ranging from day 1 to day 10.

Technical note: Detecting mortality in young adults: Our statement that mortality levels in humans before age 40 are mostly driven by extrinsic mortality factors that mask the age-associated increase are based on the paper of Beltram-Sanches et al (PMID:23626899) analyzing 650 human cohorts from several countries. In these studies, it becomes clear that mortality levels in humans before the age of 40 are not driven by age, but by extrinsic factors. This seems similar in worms. Our statement that *C. elegans* mortality levels before day 12 cannot be reliably quantified to detect changes in mortality rates is based on our data measuring lifespan for large cohorts (ranging 500 to 50,000 animals). From these data-sets, we estimate that there are generally 1: 1000 dead animals by day 1 of adulthood (e.g. 56 out of 54,688). Many of them, however, were unlikely to have died due to aging suggesting that the age-associated mortality rate on the first day of adulthood (day 1) is lower than 1:1000. Even around day 4, we find that mortality rates vary by several fold between experiments again suggesting that even the few deaths caused by extrinsic, non-aging factors (e.g. protruded vulvas) cause a dramatic variation. Our current best estimate for day 4 and 5 is a mortality rate of 0. 003 (3:1000). But again if only one of these 3 deaths is due to non-aging, the value is off by 33% and if two are, the value is off by a factor 3, making it impossible to distinguish different rates. Starting from day 12 the mortality levels between large cohorts of over 1000 animals become comparable. We found publications showing mortality curves that include values for young *C. elegans* adults, but none of these studies included confidence intervals nor any power of detection calculations. As these studies generally used lower number of animals compared to our larger cohorts, we have to assume that the errors for young adults inherent in published mortality curves are as substantial as the error in our curves.

In summary, we agree that youth is not a stage and youth was a poor choice of word. We argue that a period extension model doesn’t require a specific stage but only degenerative changes that occur during a restricted period throughout life. In the revised manuscript, we provide evidence that the mortality rate in early adulthood is decelerated in mianserin-treated animals as suggested by drift-analysis. We show that mianserin treatment is only required for a limited period of life. The conceptual novelty is that period extension is a class of lifespan-extending mechanisms that act within a defined beginning and an end and that mechanisms exist that specifically extend the duration of the period of early adulthood. Period extensions that extend the period of young adulthood have been missed, as age-associated mortality levels are very difficult to measure in young adults. We show that a deceleration of mortality rates early in adult life seizes before the onset of mortality. Period extensions provide an additional explanation on how parallel curve shifts in mortality curves arise and that the longevity effects do not have to act throughout life.

In the revised manuscript we added the two additional experiments mentioned above.

I) Figure 6: We show that the 7-8 day shift seen in the transcriptome by drift analysis and PCA (Figure 6) precedes a 7-8 day shift in the mortality curves. With 1500 animals in each condition, our mortality curve has the statistical power to detect differences of 1 day. These results show that mianserin lowers the mortality rate prior to day 12, but not after day 12.

II) Figure 6: We further expose worms to 8h, 1, 5, 10, 15 days of mianserin and compare the effect on lifespan to lifelong treatment. Mianserin treatment for 5 to 10 days is required and sufficient for a full lifespan extension. Shorter exposures give less of an effect and longer exposures don’t increase it further. As one reviewer eloquently put it, mianserin prevents the degenerative changes happening early in adulthood. This in turn results in lower mortality levels when the animals reach middle age (literally half the lifespan, day 10-12) and thus ultimately translates into a lifespan extension, even though the effect has seized long ago.

Reproduction: There is the concern that we ignored the possibility that mianserin acts by a mechanism related to lifespan extension by germline ablation or similar:

We completely agree with the reviewers that the period in which mianserin exerts its lifespan extending effect overlaps exactly with the reproductive period. Furthermore, *tph-1* mutant animals have been shown to have an extended reproductive lifespan in a manner dependent on *daf-16* (PMID: 10676966). All these examples, clearly suggest an involvement of the germline in the lifespan-extending effect in other lifespan-extending paradigms. We did not mention possible effects on the germline because prior experiments have convinced us otherwise. Mianserin extends the lifespan of *daf-16* mutants and germline ablation does not. Mianserin does not extend the lifespan of *eat-2* mutants but germline ablation does (PMID: 18033297, PMID: 17711560). We now mention this explicitly in the text.

We added two new experiments that are again inconsistent with an effect of mianserin on the germline.

I) Figure 6: We tested whether mianserin-treatment extends the reproductive period. It did not. The number of eggs that mianserin-treated animals lay is about the same as untreated controls.

II) Figure 6: We asked whether mianserin treatment increased proteasome activity as seen in *glp-1*(ts) mutants. It did not and on day 5 showed a slight reduction. Interestingly, the drift plots for proteasome genes show that mianserin treatment increases drift variance on day 5, consistent with a slight reduction in proteasome activity with Mianserin.

Concern 3: What is drift useful for?

Regulation vs. deregulation:

Two reviewers appear to argue opposing points whether drift is an addition to the concept of the regulatory/proteostatic collapse and thus a sign of deregulation or whether drift is due to regulated change. We have re-written the manuscript in a way that does not take a position on whether the changes observed with drift are due to regulation or deregulation (see also our answer to point 1 for more detail). We also stated this very clearly in the first version, but it seems that the word “drift” has suggested that we have taken a clear position on the issue. We used the word drift because of prior uses by others (e.g. “epigenetic drift” used by George Martin).

The point we are trying to make is that as animals age, co-expression patterns change causing dramatic changes in mRNA balance between genes cooperating in the same function. Sustained imbalances in mRNA relations that can reach 400 fold differences are unlikely to be beneficial for the pathways function as we show for the redox pathway. Irrespective of whether the changes in the transcriptome are due to regulation or deregulation, their effect on physiological function is degenerative.

Conceptually, drift measurements are similar to lifespan measurements but allow tracking the rate of physiological change within any type of -omics data. Stoichiometric balance changes with age not only in transcriptomes but also in proteomes and metabolomes. In collaboration with other labs, we are about to submit a paper describing metabolomic drift in aging mouse and human brains.

The concept of drift provides a measuring stick for age-associated change in omics data. Using drift as a measuring stick allows us to sort transcriptomes/proteomes/metabolomes into older and younger transcriptomes without the need for prior knowledge about the species and how long it lives. The only information necessary is a young reference. Thus, drift-analysis provides a measure to track the aging process within different molecular layers (RNA, Protein, Metabolome) and along the central dogma of molecular biology. As we show in the paper, drift-analysis further allowed us to detect age-associated change much earlier than that is possible by analyzing mortality.

Drift vs. biomarker:

We carefully used the word metric and avoided the word biomarker throughout the manuscript. We used the work “marker” once saying “it is *not* a mere marker.” The word “mere” was misplaced, as it implies it is something better rather than conceptually different. As we state above, drift is a metric similar to lifespan, which tracks physiological change with age, but instead of measuring at the level of the organism it measures aging at the molecular level (-omics data, transcriptome, proteome, metabolome). As rightly stated by the reviewers, a true biomarker should work on the basis of an individual. The only data we provide that drift is applicable to individuals are the human brain data (Figure 7). We have started to investigate the possibility of whether drift could be used as a biomarker for a single individual by analyzing several thousands of arrays of published datasets from worms, mice, rats, macaques, and humans. Many of these published data-sets use different platforms (microarrays, different sequencing platforms for RNA-seq) and thus prevent us from making definitive statements at the moment, but the emerging picture is this:

On the organismal level, including RNA of all tissues, drift precedes future mortality events. Drift is very easy to detect in worms. For specific tissues, drift is best detected in tissues that are as old as the organism itself like brain, hematopoietic stem cells or oocytes. At least in humans, short-lived cells such as blood (~3 days old) or skin fibroblasts do not show any drift. It may be that drift works particularly well in *C. elegans* because the animal is post-mitotic.

Furthermore, we find that transcriptional drift relative to regulated changes is greater in short than in long-lived animals. This would entirely make sense if drift is a sign of a degenerative process. At least, thus far, drift is of limited use as a biomarker, as it appears most pronounced in long-lived tissues, which are poor options for biopsies.

We have carefully amended the text to clearly reflect that drift is a metric and not a biomarker (subsection “Biological interpretation of transcriptional drift-variance”).

Usefulness of drift in analyzing transcriptomes: In the revised manuscript, we compare the results of standard gene expression analysis and how it is simplified dramatically by drift-analysis. There is a striking difference and a very convoluted and complicated picture of hundreds of functions that change with age and with mianserin. Subtracting all the changes caused by drift, leaves gene expression changes related to the xenobiotic response, innate immunity, stress and aging. Very consistently, these processes are regulated by serotonin and the likely primary response to mianserin. Murphy et.al, suggested a similar model, in that, *daf-16*, acts on aging by changing the expression of relatively few key physiological genes. The figure in 2G shows that changing the expression of a few key physiological genes by *daf-16* as shown in the original paper attenuates drift of thousands of age-associated changes (see Results section for Figure 2). Drift was suppressed in many of the same genes (58% overlap) between *daf-2* and mianserin, consistent with chemical epistasis, showing that these two mechanisms overlap but are not identical (11% lifespan extension in *daf-2* instead of 31%, N2). It will be interesting to see whether other genetic epistasis experiments can also be explained on the basis of overlapping and non-overlapping sets of genes, whose age-associated drift-variance is attenuated.

Some other uses for drift:

Drift can be used to compare tissue-specific aging between brains of different mouse strains. The senescence-accelerated mice (SAMP mice) for example show an accelerated senility and it is currently not clear what this is based on. As mentioned above, the concept allowed us also to detect aging in aging mouse brains based on proteomics and metabolomics. Thus drift, the loss of stoichiometry between molecular components is a phenomenon that describes age-associated changes along the molecular information paradigm (RNA-> Protein->metabolite). For example, in collaboration with the Shubert group at the Salk Institute, we were able to identify the pathway that a drug molecule affected to reduce the age-associated cognitive decline in SAMP mice. Biochemical analysis further confirmed the finding. Standard gene-expression analysis did not pick this up because of the thousands of changes that were ongoing due to aging. We have written R downloadable programs for everyone to conduct such analysis in the future.

We certainly acknowledge that drift is not perfect and that there are many outstanding questions. The conceptual insight is that by looking at changes *among* genes across thousands of cells and using a young reference, we can detect slow and continuous deviations from the young state. How often and to what degree these changes are erased by dramatic overall gene expression changes, and how often such changes occur is not known. In the presented manuscript, we have made a substantial effort to address many of the most basic questions including whether drift is attenuated by longevity, whether it is evolutionarily conserved and whether it compromises function. By revealing the period specific extension of lifespan extensions, which we now confirmed using other methods (mortality, restricted treatment, Figure 6), we show that drift allowed us to detect a rather surprising way to extend lifespan that was previously missed.

*Essential revisions: 1) The paper would be improved considerably if the speculations were toned down, and the data were allowed to speak for themselves. Most importantly, the ideas and hypotheses presented in the manuscript are not tested in a rigorous manner. The authors use predominantly descriptive data to support their own interpretation of the biology without asking whether they are also consistent with alternative interpretations.*

In the revision, we have each time mentioned the most important alternative models we considered. It is not that we did not consider them before but that we did not mention them in the interest of brevity. We acknowledge that this practice must have given the impression that we approached this project with preconceived notions. This was by no means the case. At the start of this study, we conducted a standard analysis of the gene expression data (now included in Figure 1, [Supplementary-material SD1-data]–[Supplementary-material SD5-data] and Figure 1—figure supplement 1). We were concerned by the fact that the very complex picture that emerged allowed us to support any hypothesis simply picking some genes over others. Our data could have supported the statement that mianserin directly activates the oxidative stress response or could have supported the exact opposite statement dependent on the set of genes, we would have chosen to mention. If any situation “muddies the waters” it is this.

We present the drift model that accounts for all changes and does not require us to choose genes. Not only did this model account for the changes induced by mianserin but also for thousands of changes with age. The age-associated decline of the transcriptional machinery (more to that later) leads to loss of stoichiometry, and thus loss of co-expression patterns as seen in young adults. This method of explanation accounts for a large fraction of the observed changes, not requiring us to choose specific genes dependent on our favorite hypotheses. Consider that this explanation leads to a quantitative dose response curve across the entire transcriptome that exactly correlates with the associated longevity. This solution was by no means obvious and required a fair bit of computational and statistical development. By any account, we have gone out of our way to avoid bias. To highlight this and to avoid the impression that must have triggered the above comment we now start the paper with a standard analysis (Figure 1 and Figure 1—figure supplement 1).

*For example, the authors claim that if their measure of drift variance increases with age it is a sign of loss of homeostasis and of an increase in aging-related loss of transcriptional control. However, because the authors are examining among-gene variance (more on that below), this observation is also consistent with a model in which gene expression changes with age in a strongly regulated manner.*

The point raised by the reviewers is important because regulation is a problematic term in the aging field. In the previous version we used the term transcriptional regulation in a very narrow sense: transcriptional changes that modulate a cellular or physiological function in a way that is productive and beneficial for the organism’s development, reproduction or survival. Changes that lead to functional decline, sterility and death, in short, those changes observed with aging, are according to this definition aberrant, detrimental and not regulatory.

We agree that this view of transcriptional regulation in the field of aging is problematic as it automatically implies that aging, a process that leads to functional decline is not regulated. For example, a recent paper by the Morimoto group showed that activation of hsf-1 is actively blocked by the germline, and therefore a regulated response. Blocking hsf-1 activation is regulated, but the consequences are degenerative leading to inappropriate folding of proteins, aggregation and ultimately degeneration. The initial event is regulated; however, the consequences are not. In analogy, transcriptional-drift is the consequence of de-regulation in transcriptional control. The activation of transcription factors such as *daf-16* regulate longevity by acting on subsets of genes to preserve transcriptional control. Consider Figure 2. Even though *daf-2* RNAi leads to the activation of *daf-16* and an entire anti-aging program as shown previously, drift becomes smaller. Compared to the number of genes that change expression due to activation of *daf-16* (drift should increase), the number of genes that change less due to age is much more. As with *hsf-1*, the activation of the *daf-16* mediated anti-aging is subject to regulation, however, the consequences (drift) of not activating these transcription factors are probably degenerative.

What is the evidence for the existence of degenerative changes in aging transcriptomes? Multiple studies show that aging causes changes to the nuclear architecture, loss of histones, loss of histone modification as well as DNA damage (Misteli, Taylor, Brunet, Morimoto, Puh). In many cases, reversing these effects was found to extend lifespan showing that these changes are detrimental to survival. Thus, by the definition stated above, these changes are non-regulated as they are detrimental to cellular function and survival. Two very recent papers have shown that loss of histone methylation causes aberrant expression that increases with age exactly as seen with drift (PMID:26159996, 25838541).

While age-associated changes around the nucleus and chromatin are acknowledged and frequently cited, when it comes to analyzing transcriptomes from aging organisms, these findings are mostly ignored. Transcriptional data of aging organisms have been analyzed using the same principles applied for data from developing animals, despite these two having very different underlying biology. Thus far we have analyzed over 1000 published gene expression arrays with respect to drift. The results are mostly consistent with wide-spread degenerative changes. Since we could not think of a way to experimentally decide the issue of regulation versus deregulation we have re-written the paper in a way that avoids statements about regulation or de-regulation. However, regulated or not, increases in drift-variance are a deviation/loss of gene-expression patterns as observed in young adults.

Changes in the text:

We now included a supplementary figure (Figure 1—figure supplement 1) that shows 50 representative pie charts (out of 249) for GO annotations in which significant numbers of genes change in opposing directions with age to illustrate the dramatic change. We hope that the reviewers agree that for many of these pie charts it is rather difficult to make any statement about how their function changes with age.

We further added an experiment showing that reduced transcriptional drift associated with mianserin treatment is associated with an increased homeostatic capacity in the redox system (Figure 5). ROS-mediated induction of redox genes in animals pretreated with mianserin at a young age and challenged with paraquat at older age is much higher than in untreated controls. These findings are consistent with the model that mianserin preserves homeostatic capacity by attenuating drift and increases in drift are a sign of impaired transcriptional control.

*Some genes may increase/decrease expression with age to promote reproduction, for example, or to modulate adult-specific behaviors. Even if such gene expression was regulated with 100% precision (i.e., reproducibly identical among individuals) then the variance in expression among genes will increase with age. Slowing the aging process (i.e., reducing the slope of that increase) would also reduce the rate of increase in drift-variance. This would occur without any specific youthful phase and completely in the absence of any sort of aging-related loss of transcriptional control. In other words, a model that is conceptually the opposite of the one proposed by the authors is consistent with many aspects of their observed data. There is no attempt or framework to distinguish this (and other) alternative hypotheses. For example, models of homeostasis can be separated from models of defined, aging-related changes by also examining variance within genes.*

We are not entirely sure we understand all the points the reviewers are making. We specifically acknowledged in the first version of the manuscript that we do not know whether drift is the consequence of regulated or unregulated changes right at the point when we introduce the concept: “To emphasize our limited understanding on whether these age-associated changes in transcription were caused by regulatory programs or a progressive loss of transcriptional control with age, or a combination of both, we will simply refer to these changes as transcriptional drift”.

This was a direct statement from the start that we considered both possibilities. As stated above we have now removed any reference to regulation from the revised manuscript and simply refer to them as changes and clearly lay out both possibilities in the discussion. As we explained above the term regulation is problematic because the active (and regulated) inhibition of protective programs such as DNA damage response or heat-shock response will lead to many non-regulated degenerative events. However, in the end, drift increases with age as co-expression and mRNA stoichiometry drift away from what is seen in young adults.

We also would like to emphasize that drift does not have to necessarily increase with the activation of a transcriptional program.

Example 1: Consider Figure 5 (N2) that shows the change in *sod* genes with age and upon induction by oxidative stress. In Figure 5, in response to oxidative stress by paraquat all five *sod*’s are induced and expression increases (at least if the animals were treated with mianserin (light blue)). As the changes in Figure 5 go into the same direction, the effect on drift (their relative ratio to each other) will be minor. Now consider Figure 5, showing how the same genes behave with age. *Sod-1, -2*, go down *sod-3* does not change (not shown) and *sod-4* and *sod-5* go up. This will cause a major effect on drift.

Example 2: Consider Figure 6—figure supplement 1 that shows how proteasome genes change with age. Proteasome genes show very little drift (type II), but a coordinated down regulation with age (type I). There is no effect on drift (in water-treated animals) but a clear effect on the overall level of transcription.

Transcriptomes that consist of RNA of hundreds of adult worms that age without major environmental changes will “iron out” changes that are not happening in the majority of cells. If transcriptomes of adult animals would be highly dynamic, none of the aging studies conducted thus far could have found any effects due to aging as any short term effect on gene expression would overshadow events caused due to age. Thus the changes we see are those that persist and are occurring in large fractions of the cells. Transcriptomes of aging worms, such as the one we present, seem to be rather stable. We have controlled for the example of reproduction by including the analysis of the Murphy et al., 2003 data which used a sterile strain. This shows that different ways of sterilizing result in the same drift-phenomenon irrespective whether the animals make eggs or not. As mentioned above, the notion that drift must increase across the trancriptome with the start of any transcriptional program is directly contradicted by Figure 2. Inhibition of *daf-2* by RNAi leads to the activation of the transcription factor DAF-16, which is a transcription factor that induces a transcriptional program. Yet, induction of the transcriptional program causes a very clear reduction in overall drift across the entire transcriptome. Intuitively, one would expect the opposite as the reviewer states above, and it is a strength of the drift model that it is clearly able to explain an unexpected outcome.

As we state in the summary for concerns for point 13 and 14, there is no requirement for a youthful state for a period extension.

We now directly tested the hypothesis (Figure 3) whether mianserin preserved homeostatic capacity (Figure 5). The results show that mianserin prevents changes in redox genes with age. Mianserin-treated animals show less expression of redox-related genes compared to age-matched controls (Figure 3). However, if we add paraquat, mianserin enhances redox genes expression and is higher than in age-matched control. Therefore, mianserin improves homeostatic capacity with age, as it preserves the ability of the animal to appropriately respond to outward stimuli.

The data are predominantly descriptive.

We agree that there are lots of descriptive data. All the data that drift increases with age in worms, mice and human brains are descriptive and show that drift is a wide-spread phenomenon. However, we have directly tested the hypothesis that the observed increases in drift are a sign of the physiological processes associated with aging. The following experiments are not descriptive:

Two different lifespan extending paradigms (mianserin and *daf-2*) attenuate drift (Figure 2).

Abrogating the lifespan-extending effect by adding mianserin too late (day 5) or adding *daf-16*RNAi to *daf-2*RNAi abrogate the effect on lifespan and on drift (Figure 2).

Mianserin dose response curves show a clear correlation between longevity and drift-variance across an entire transcriptome (Figure 2).

Removing *ser-5* abrogates the effect of mianserin on drift (redox genes), stress resistance, and lifespan (Figure 4, Figure 5).

Many of the experiments were repeated with 7 structurally different serotonin inhibitors all requiring *ser-5*. Therefore, the observed effects are not caused by some mysterious unknown side-effects of mianserin (Figure 4—figure supplement 1, Figure 5—figure supplement 1).

*2) Consideration of alternative models is also lacking from other aspects of the manuscript. For example, the authors interpret the data in Figure 3 as implying that mianserin early in life is the key aspect of the manipulation that leads to enhanced stress resistance. While this interpretation is consistent with the observations, so is a more parsimonious one; that increased time on mianserin increases stress resistance regardless of the age at which it is given. (See also other concerns, below, on mianserin experiments).*

Thank you for this comment. In the revised manuscript, we now lay out several alternative possibilities. If the data in the previous version were insufficient to discount one or the other possibility, we added additional data. The comment above points out that the duration of miaserin treatment could be the deciding factor. However, that this was not the case, as directly addressed in Figure 3. Adding mianserin on day 1 testing resistance on day 5 increases stress resistance (Figure 3, middle bar), adding Mianserin on day 5, and testing resistance on day 10 does not (Figure 3, bar to the right). In both cases, there is a 5 day incubation but when the compound is added to day 5 adults, there is no significant effect on stress resistance. From the Table 4, it can also be seen that survival in Figure 3 for day 5 and day 10 is nearly the same at 90% in both cases. The greater fold change on day 10 is due to the lower survival of the control animals due to aging.

*3) Many studies in different species have described versions of the concept that parameters associated with fidelity of gene expression decline with age. These include evidence concerning proteostasis and stress response "collapse", as well as transcriptional changes, noise, and epigenetic alterations associated with age. Many of these changes occur during early adulthood in the worm. These findings are cited to an extent, but not comprehensively and not in a way that fully develops this idea. This is important because it is not clear whether transcriptional drift truly represents something new, or yet another parameter reflecting this regulatory decline. That doesn't mean the marker isn't useful, but it needs to be put into a more appropriate context in order to judge its significance.*

We apologize for the fact that we did not cite some of the earlier work. We hope we have done a better job this time. It was by no means our intention to diminish anyone’s contribution. We agree that the general idea that aging causes a regulatory decline in transcription is not novel and has been shown many times over. The lifespan extension by overexpressing Sirt2 suggested very early on that aging impairs transcriptional regulation and chromatin. However, despite these early findings, transcriptomes of aging organisms have been mostly analyzed by the same principles as transcriptomes from developing animals, not taking into account the above mentioned findings that aging has a degenerative effect on transcriptional control. If gene expression fidelity is compromised in aging organisms, this will affect aging transcriptomes in ways different from development. While statistical methods improved to analyze transcriptomes, other than linking genetic and biochemical information to the names of genes, biologists have done hardly anything to account for differences in biology between development process and aging process when analyzing aging transcriptomes.

We have rewritten the entire discussion where we point out weaknesses and possible considerations.

*4) The claim is that drift variance is a biomarker of aging, i.e., a measure that more accurately predicts "physiological age" or future lifespan better than chronological age. However, a population measure is used, which confounds several influences. True biomarkers of aging must be defined at the level of the individual, and they are ideally based on rates of change across ages (repeated measures) and are validated by establishing that they predict remaining lifespan better than chronological time itself. Even if we accept this as a biomarker, the human data are confusing, where transcriptional drift variance doesn't appear to exhibit significant differences across test subjects until the age of 60 (Figure 6). Does this imply that humans are in their youthful state until 60? These data appear to undermine the notion that this measure is an evolutionary conserved bio-marker of aging.*

Again, as mentioned in the summary, we avoided the word biomarker. The concept of a biomarker for aging implies that one can measure a subset of genes/processes across age in an organism and thus predict remaining lifespan/age for the entire organism. In other words, it assumes that aging is homogenous within the same organism. This is an assumption and is not necessarily true, or may be only true for a subset of tissues or genes. What we claim is that using the entire transcriptome of a whole organism, drift seems to accurately predict later death events as we show for *C. elegans*.

However, this result does not imply that any chosen subsystem/organ ages at the same rate. Reproductive organs, for example, seize functioning much earlier than brain function and therefore “age” faster. While there is some decline in cognitive reasoning in humans past the age 45, this becomes worse after 60 and is more pronounced in men than in women. The drift-plots of human frontal cortex, a region of the brain responsible for reasoning, reflect the relationship between drift and function rather accurately. As we could not find a second study that was so rigorously conducted to test transcriptional changes in other human organs with age, it is yet to be seen whether these correlations are applicable to other human organs. Owing to this reason, we did not mention about functional decline and drift in humans, so as not to overstate the case. Again, as mentioned above, drift seems to be most pronounced in tissues that are as old as the organism itself and is not seen in tissue with a high turn-over such as blood, which curtails its usefulness as a biomarker right there. However, one interesting application would be to ask whether different parts of the brain age to the same extent, and comparing drift in core pathways that are expressed everywhere such as sugar metabolism, mitochondria and actin skeleton and ask how this is different in long-lived mouse strains.

*5) We do not understand the assertion that there exists a qualitatively distinct, youthful phase. If we accept drift variance as a biomarker of aging, all of the data appear to change relatively continuously throughout the measurement period. It is formally possible that if the authors' normalize gene expression to L2 data that age 1 day adults would have lots of "drift." In other words, the work is lacking specific criteria upon which to define this hypothetical new youth stage. The lack of a clear definition for "youth" leads to confusion and ambiguity. For example, the authors state: "Simply put, by chronological age of day 10, mianserin treated animals had yet to progress transcriptionally beyond the physiological age of day 3 [based on data in Figure 2]." However, they do not point out that mianserin treated individuals on day 10 have higher variance than they do on day 2 (even though they do have similar levels of variance seen in younger control populations), which implies that they have "gotten worse" over time. If we accept that they putatively remain in a youthful stage, then we must accept that they were hyper-youthful early on and that youth was subject to the same process of change in drift variance as aging. So how is this different from aging itself?*

“Youth” was a poor choice of word on our part, as it is ambiguous in a way that we were not aware of. Young or early adulthood is more precise. Again as mentioned above, a period extension only requires a process that contributes to aging that has a restricted duration. The reviewer is correct in observing that the mianserin-treated animals got worse from day 3 to day 10. What mianserin does is, it decelerated the rate of age-associated physiological change during the first few days of adulthood. Yes, the mianserin-treated animals “got worse” from day 1 to day 10, but at a slower rate compared to untreated animals. At least according to Gompertz-types of analysis a lowering of the rate of physiological change is the equivalent of slowing aging. What is interesting and novel is that mianserin does not decelerates the rate of age-associated physiological change throughout the entire life as has been shown before for age-1 and other interventions (Johnson 1990), but does only during a specific period before the animals enter middle age and mortality sets in (middle age in its literal sense, day 10, half the age of 21 day lifespan).

To make an analogy, let’s consider a train travelling from point A to C. A train can arrive 7 hours too late at point C because it is driven slower throughout the entire track or because it was very slow initially but then proceeded at the normal speed. Which of the two scenarios is true can be determined by looking at milestones across the track. If the train had already a 7-hour delay when reaching point B, it means its speed from A to B was so low that it caused a 7-hour delay, but also that its speed from B to C was normal. Drift provides such milestones and applying the idea, by day 10, the mianserin-treated animals reach the day 3 milestone 7 days later than the control animals. On day 28, they reach the 21 day milestone (mean lifespan) still 7 days later than the control. There is a 7-day delay in aging till the end of the lifespan experiment and this delay can already be observed on day 10 of adulthood.

We now added a new Figure 6 including 2 new experiments addressing these issues (6B, C, D). We show (indirectly) that the rise in mortality levels in young adults is lower for mianserin-treated animals before day 12, but not after. Drift shows a 7-8 delay by day 10, mortality by day 12 (mortality on day 10 is not high enough to detect 1 day differences). Adding mianserin for 5 days or 10 days only is necessary and sufficient to extend lifespan.

*6) Furthermore, additional insight into the youthful state could have been provided by including data beyond age 10 days. At 10 days old, more than 95% of the worms are still alive. Have authors measured what is the transcriptional drift at 50% of the survival? Or 90% of the survival? Does transcriptional drift increase linearly, or does it plateau at a high level later in life, as the authors seem to suggest. If transcriptional drift mostly happens in early life, can this be reliably used as a biomarker of aging?*

We are not sure how we gave the impression that we think that drift plateaus. We never intended to. We added the analysis of two additional published dataset that shows that drift increases from day 10 to 15 to day 20. The question of linearity is a bit tricky to answer, as the drift-variance is a logarithmic scale. At least, we can say it continually increases till day 20. We think, however, the best use for drift is the ability it provides to analyze aging before the animals die.

*7) Importantly, it seems like a stretch to conclude that mianserin specifically slows aging during youth, at least based upon the evidence here. As noted above, many parameters degenerate during early adulthood, but aging biomarkers become manifest later. Isn't it more likely that drift and other markers of gene expression changes reflect an early degeneration in control that will play out in its biological effects later, rather than a specific marker of "aging" in the young? To address this idea definitively, we would need to know whether other interventions have the same effect on transcriptional drift, or act on downstream or different parameters. Is mianserin unique, or is protecting the animal from this regulatory decline the most effective way to delay aging, and one that is seen in most contexts of long life?*

We politely disagree with the “too much of a stretch” but we agree with the rest of the statement (“many parameters degenerate during early adulthood”). We wished we had had these precise words when we wrote the initial manuscript. Mianserin treatment prevents early degeneration occurring in young animals, lowering the age-associated mortality rate when the animals reach middle age. Thus, it is almost how the reviewer says it is. There is early degeneration that becomes apparent in mortality later. But, the only reason it becomes apparent later is the inability of mortality analysis to measure aging in young animals. The mortality rate is rising in young adults and mianserin is preventing the rise but the number of deaths is simply too low to reliably determine differences in mortality rates at the early adulthood. If it were possible to measure mortality, one would see that mianserin decelerated the mortality rate in young animals up to around day 10 to 12 and then stops doing so. However, drift analysis and some of the experiments we added reveal that. Without monitoring transcriptional drift, a period extension mechanism decelerating aging in young animals is invisible until mortality is high enough to be measured. However, by that time the period extension process is over and the only signature left behind is a parallel shift in mortality curves, which has been previously interpreted as non-aging. That interpretation is correct regarding many experiments (e.g. experiments involving radiation for example (PMID25750242). However as our data show, a parallel shift can also indicate that the slowing of the aging rate occurred at a time before mortality can be reliably measured. Therefore, “mianserin specifically slows aging during early adulthood” seems like a stretch because we and others have been unable to detect it thus far, before the development of the drift metric.

In addition to the mortality rate experiments, we confirmed the specific action by mianserin during day 1 to 5 by limiting the exposure of mianserin to young animals, which shows that this is necessary and sufficient to extend lifespan. Thanks to this review, we realized the possibility that parallel shifts in lifespan curves cannot only be explained by a proportional lowering of risks throughout life, but also by a specific slowing of age-associated physiological decline, specifically in young adults. Since parallel shifts in lifespan are quite frequent we would venture to suggest that mianserin is by no means unique. At the moment, however, using compounds to extend lifespan is the only technical means that allows switching longevity mechanisms “on” and “off” to distinguish period extensions from other possibilities on how to explain parallel shifts. (RNAi has no “off” switch).

We agree that we should look into this in all kinds of lifespan extension mechanisms, as we only show drift to occur in two different contexts. We are trying to find a cheaper way to measure drift without conducting three RNA-seqs for each condition. We also can say that the drift effect is not unique to transcriptome but is also present in the proteome and the metabolome and we are about to submit a paper about this effect in the human and mouse brain with our collaborators.

In summary between i) analyzing ~1000 published arrays, ii) investigating drift in transcriptomes, proteomes and metabolomes in different longevity mutants, species and organs, and iii) analyzing the different periods in which life can be extended, we believe that the concept of drift is rather stimulating with implications on many aging-related topics and datasets.

*8) The worm is different in many ways biologically at day 1 compared to later days. Reproduction ceases day 4-6, and body size increases during adulthood. How might these affect "drift", or vice-versa? Does mianserin affect these parameters? How can claims be made about effects on aging early in life without considering this biology? It is either disingenuous or naïve to describe young adulthood in terms of a period of aging while ignoring the biological events that are occurring, particularly the cessation of reproduction.* Thank you for pointing this out. We have considered the aspects of germline biology scientifically while designing our study, but did not mention it because our previously published results seemed to exclude the involvement of the germline. One reason to analyze the Murphy et al., 2003 data as well as data from non-reproductive tissues in mice and humans was to ensure that fertility is not the driving factor behind drift. We now remedied our lapse and provide extensive data that show that FUDR sterilization or fertility does not affect drift (Figure 2—figure supplement 2) and address the germline further in Figure 6. We also mention in the text that the fact that mianserin extends lifespan of *daf-16* but not of *eat-2* (Petrascheck, 2007) made us discount a direct germline-related lifespan extension early on. To further establish this, we have now included data showing that mianserin does not increase proteasome activity as seen in *glp-1* mutants and that mianserin does not increase the reproductive lifespan in contrast to *tph-1* (Figure 6). We hope we have now amended what we failed to do the first time around.

*Some additional important points: 9) What are the genes and GO terms that are changed between mianserin and untreated animals? Is this transcriptional signature like* daf-2*, or like CR treatment? This would be good to know and discuss* before *using transcriptional drift analysis, which looks at the rate of those changes, since later some of these targets (?) are discussed in the redox potential experiments. This information would also help us understand what the underlying mechanisms of mianserin's delay of aging are. And how many of these expression differences are attributed to eggs rather than soma?*

Thank you for pointing this out. At the outset of the project, we conducted all these analysis as stated above. We have now included them in the revised manuscript (Figure 1—figure supplement 1 and [Supplementary-material SD1-data]–[Supplementary-material SD5-data]). This analysis however showed a very complex and convoluted picture that was dramatically simplified by accounting for drift. Because the initial analysis was rather disappointing to us, we did not include it in the first version. However, we see that it is necessary to justify the reasons why we dramatically deviated from standard methods of analyses. Using the Murphy et al., data we show a 58% overlap in genes whose age-associated changes are reversed in both, mianserin-treated as well as *daf-2* RNAi treated animals (See Results section for Figure 2). This overlap is consistent with the fact that mianserin extends lifespan of *daf-2* mutants but only by ~11% rather than the ~31% seen in the parallel N2 experiment. We were unable to find published dietary restriction transcriptome data that monitored differences across different ages to conduct a similar comparison.

*10) Most of these experiments in worms, with the exception of the published data on* daf-2 *and* daf-16 *RNAi (which use worms without sperm), use animals that are still fertile. Since that is the case, how can the authors know which transcripts are due to changes in the soma, which presumably are the ones they are interested in because of aging, or due to changes in the eggs? It seems that the authors could do some comparisons with the control worms from the sterile animals to address this point; otherwise, one set of experiments that uses mianserin-treated sterile animals would be expected.*

This is a good point. Please see response to point 14. We have addressed this point in Figure 2—figure supplement 2. FUDR-treated eggs actually contain very little RNA compared to normal untreated eggs and the contamination of egg RNA is minor. Using various published datasets of egg RNA, neurons, and sterile strains we show that drift is hardly influenced by the presence of FUDR-treated eggs.

*11) While the SER-5 data are clear, we did not fully understand why other receptors would not be needed for stress protection which they are still needed for the lifespan effect.*

Please consider our response to point 9. Mianserin and *daf-2* attenuate drift in overlapping sets of genes ~58%. Mianserin does extend lifespan of *daf-2* mutants but not as much as in N2. As 58% of the genes are already attenuated by *daf-2* alone mianserin only adds to the anti-aging affect by attenuating an additional 42%, thus causing an increase in lifespan in *daf-2* animals that is a bit less than half what it does in N2(See results section for Figure 2). The same scenario is true for serotonin receptors. Serotonin affects different responses by different receptors. Note that in both, *ser-4* and *ser-5* mutants, mianserin still has some residual effect on lifespan suggesting independent functions. We show that SER-5 is specifically required for mianserin to induce stress resistance and to attenuate drift in the redox system. In the revised manuscript, we further show that SER-3 nor SER-4 are neither required for stress resistance nor for attenuating drift in redox genes (Figure 4 and Figure 4—figure supplement 1). Our model suggests that this is because these receptors attenuate drift of different subsets of genes. Thus, we should be able to dissociate which receptor causes transcriptional drift for which cellular process. If our model in which repeated activation of a transcriptional program causes drift is correct, then each of these receptors required for lifespan effect will cause drift in specific subset of genes it activates, when stimulated by serotonin or octopamine. For now, we provide evidence for the correlation between drift attenuation and redox function preservation by Mianserin via SER-5.

*12) Does* daf-16 *suppress mianserin effects or act in parallel?*

Mianserin clearly extends lifespan of *daf-16(mu86)* animals as well as of *daf-2(e1370)* animals. The lifespan extension, however, is less than in wt animals. (+11% instead of 31%)(Petrascheck 2007). Interestingly, the set of genes whose drift is attenuated by mianserin and *daf-2* with age overlap by 58% (see Results section for Figure 2). Thus, the overlap of drift-attenuation fits exactly the partial lifespan extension we have reported previously. Similar to this, it will be interesting to learn whether genetic epistasis can be explained by the overlap of genes whose drift is reversed.